# Hepatic nutrient and hormone signaling to mTORC1 instructs the postnatal metabolic zonation of the liver

Ana Belén Plata-Gómez [1], Lucía de Prado-Rivas [1], Alba Sanz[1], Nerea Deleyto-Seldas [1], Fernando García [2], Celia de la Calle Arregui [1], Camila Silva[1], Eduardo Caleiras[3], Osvaldo Graña-Castro [4,5], Elena Piñeiro-Yáñez [4], Joseph Krebs[6], Luis Leiva-Vega[7], Javier Muñoz [2,8], Ajay Jain[6], Guadalupe Sabio [7] & Alejo Efeyan [1] ✉

The metabolic functions of the liver are spatially organized in a phenomenon called zonation, linked to the differential exposure of portal and central hepatocytes to nutrient-rich blood. The mTORC1 signaling pathway controls cellular metabolism in response to nutrients and insulin fluctuations. Here we show that simultaneous genetic activation of nutrient and hormone signaling to mTORC1 in hepatocytes results in impaired establishment of postnatal metabolic and zonal identity of hepatocytes. Mutant hepatocytes fail to upregulate postnatally the expression of Frizzled receptors 1 and 8, and show reduced Wnt/β-catenin activation. This defect, alongside diminished paracrine Wnt2 ligand expression by endothelial cells, underlies impaired postnatal maturation. Impaired zonation is recapitulated in a model of constant supply of nutrients by parenteral nutrition to piglets. Our work shows the role of hepatocyte sensing of fluctuations in nutrients and hormones for triggering a latent metabolic zonation program.

Mammals have evolved multiple homeostatic responses to maintain circulating levels of nutrients within a narrow range despite feeding intermittency and extended periods of fasting. The liver coordinates many of these adaptive responses, including the storage, mobilization and metabolism of glucose, amino acids and lipids[1]. Systemic control of metabolism by the liver is in part achieved through the spatial segregation of metabolic functions associated to the exposure of distinct pools of hepatocytes to portal or systemic circulation. Portal hepatocytes are directly exposed to incoming blood supply from the portal circulation afferent from the gastrointestinal system, while central hepatocytes are irrigated by blood supply poor in oxygen and nutrients as a result of their consumption by hepatocytes along the portal-to-central transit. This unique circulatory system posits the liver as a privileged organ to probe, sense and respond to fluctuations in nutrient levels and in systemic metabolic hormones such as insulin[2].

The mechanistic target of rapamycin complex 1 (mTORC1) is a key metabolic kinase that triggers cellular anabolism in response to signaling cues from cellular nutrients and metabolic hormones[3,4]. Hormones such as insulin and growth factors (GF) activate a signal transduction cascade at the plasma membrane that leads to PI3K/Akt

[1]Metabolism and Cell Signaling Laboratory, Spanish National Cancer Research Centre (CNIO), Melchor Fernandez Almagro 3, Madrid 28029, Spain. [2]Proteomics Unit. Spanish National Cancer Research Centre (CNIO), Madrid, Spain. [3]Histopathology Unit. Spanish National Cancer Research Centre (CNIO), Madrid, Spain. [4]Bioinformatics Unit. Spanish National Cancer Research Centre (CNIO), Madrid, Spain. [5]Department of Basic Medical Sciences, Institute of Applied Molecular Medicine (IMMA-Nemesio Díez), School of Medicine, San Pablo-CEU University, CEU Universities, Boadilla del Monte, Madrid, Spain. [6]Department of Pediatrics, Saint Louis University, Saint Louis, MO, USA. [7]Myocardial Pathophysiology, Centro Nacional de Investigaciones Cardiovasculares (CNIC), Madrid, Spain. [8]Present address: Cell Signalling and Clinical Proteomics Group, Biocruces Bizkaia Health Research Institute & Ikerbasque Basque Foundation for Science, Bilbao, Spain. ✉e-mail: aefeyan@cnio.es

activation, which in turn inhibits the tuberous sclerosis complex (TSC1/2), itself a GTPase-activating protein (GAP) for the Ras-homologous enriched in brain (Rheb). Rheb is located at the outer lysosomal surface, where it induces an activating conformational change on mTORC1 by direct interaction[5–7]. Independently to GF signaling, intracellular nutrient abundance results in the activation of the Rag family of GTPases. In the presence of cellular nutrients (amino acids, glucose and certain lipids) RagA loads GTP, while its heterodimeric partner RagC becomes loaded with GDP, and this nucleotide configuration is essential for the recruitment of mTORC1 to the outer lysosomal surface, allowing its activation via Rheb-mTORC1 interaction[8,9]. Thus, nutrient sufficiency and GF signaling are both required for robust activation of mTORC1[10]. Hepatic mTORC1 is deregulated in the obese state, and gain-of-function alleles affecting the nutrient and the GF signaling cascades in mouse hepatocytes have collectively shown that the control of mTORC1 activity by either limiting nutrients or GF signaling is critical for the transcriptional adaptation to fasting, and that deregulated GF signaling drives oxidative stress and hepatocyte transformation[11–18].

To understand how nutrient and insulin signaling cues are integrated and cooperate in the control of mTORC1 and liver metabolism, we have generated mice with constitutive nutrient and GF signaling in hepatocytes, by means of the endogenous expression of an active form of RagA (*Rraga*$^{Q66L}$, referred to as RagA$^{GTP}$) and the deletion of *Tsc1* in hepatocytes. Simultaneous activation of nutrient and hormone signaling resulted in strikingly more profound metabolic alterations in the liver than the activation of either input alone. Surprisingly, proteomic and transcriptomic analyses revealed a strong dissipation of the metabolic portal and central zonal identities of hepatocytes. Double-mutant hepatocytes failed to execute a transient raise in Fzd receptors, which was accompanied by a deficiency in the production of the morphogenic Wnt ligands by liver endothelial cells (EC). The inability to establish metabolic zonation after birth was dependent on postnatal constitutively high mTORC1 activity and recapitulated in a model of total parenteral nutrition (TPN) in neonatal pigs. This work supports the notion of the requirement of postnatal intermittent activity of the mTORC1 pathway to execute a metabolic zonal identity that contributes to the multifaceted control of systemic metabolism by the liver.

## Results
### Characterization of mice with constitutive nutrient and hormone signaling to mTORC1 in the liver
To understand how nutrient signaling and hormone signaling to mTORC1 are integrated in the control of liver and whole-body metabolism, we generated mice with constitutive Rag GTPase signaling in hepatocytes (*Rraga*$^{GTP/floxed}$; *Albumin-Cre*$^{Tg}$, herein referred to as Li-RagA$^{GTP}$) as in ref. 17, mice with constitutive insulin signaling in hepatocytes (*Tsc1*$^{floxed/floxed}$; *Albumin-Cre*$^{Tg}$, herein referred to as Li-Tsc1$^{KO}$) as in ref. 15, and compound *Rraga*$^{GTP/floxed}$; *Tsc1*$^{floxed/floxed}$; *Albumin-Cre*$^{Tg}$, herein referred to as Li-TSC1$^{KO}$RagA$^{GTP}$) mice. Compared to the reported phenotypes of single mutants, simultaneous deregulation of nutrient and hormone signaling resulted in synergic phenotypic consequences as evidenced by the almost two-fold increase in liver weight (Fig. 1A), and the increase in markers of liver dysfunction (alkaline phosphatase (ALP), alanine aminotransferase (ALT), bile acids, bilirubin and cholesterol levels in serum; Fig. 1B), necrosis, fibrosis and inflammation (Fig. 1C, D). Inflammation was not locally restricted to the liver parenchyma, as neutrophils and monocytes were elevated in peripheral blood (Supplementary Fig. 1a), and accompanied by significant decrease in red blood cell count (Supplementary Fig. 1b). As reported for single-mutant mice, both male and female Li-TSC1$^{KO}$RagA$^{GTP}$ mice showed defects in glucose homeostasis, and female Li-TSC1$^{KO}$RagA$^{GTP}$ had further compromised glucose tolerance (Fig. 1E and Supplementary Fig. 1c).

Female Li-TSC1$^{KO}$RagA$^{GTP}$ mice also exhibited a slight increase in insulin sensitivity (Supplementary Fig. 1d, e). Li-RagA$^{GTP}$ mice have normal lifespan[17] and Li-TSC1$^{KO}$ have shortened lifespan due to spontaneous development of hepatocellular carcinomas (HCC)[15], so we monitored the survival of Li-TSC1$^{KO}$RagA$^{GTP}$ and observed an impressive acceleration of HCC development in both males and females (Fig. 1F–H) and markers of liver dysfunction, inflammation and red blood cell defects were also evident at humane end point (Supplementary Fig. 1f–h).

We next assessed mTORC1 activity by western blot of whole protein extracts of livers from 24 h fasted and refed wild-type, Li-RagA$^{GTP}$, Li-TSC1$^{KO}$, and Li-TSC1$^{KO}$RagA$^{GTP}$ mice and observed the expected increase in phospho-S6, and phospho-4EBP1 in single- and double-mutant mice, as compared with wild-type mice in the fasted state (Fig. 2A–C). Consistently with the increased phenotypic alterations driven by concomitant genetic activation of nutrient and hormone signaling compared to single mutants, Li-TSC1$^{KO}$RagA$^{GTP}$ samples showed a further increase in the phosphorylation of mTORC1 targets under fasting. Increased phosphorylation of S6 and 4EBP1 was less evident following a 2 h refeeding in Li-TSC1$^{KO}$RagA$^{GTP}$ livers (Fig. 2A and Supplementary Fig. 2a, b). We analyzed the requirement of deregulated mTORC1 activity for the observed phenotype by chronically dosing rapamycin to Li-TSC1$^{KO}$RagA$^{GTP}$. Indeed, pharmacological inhibition of mTORC1 with rapamycin in vivo not only suppressed the phosphorylation of mTORC1 targets in Li-TSC1$^{KO}$RagA$^{GTP}$ livers (Supplementary Fig. 2c, d), but also decreased liver weight to that of untreated wild-type mice (Fig. 2D) and normalized the markers of liver dysfunction (ALP, ALT, bile acids, bilirubin, and cholesterol, Fig. 2E). Moreover, rapamycin-fed mice showed no detectable necrosis, fibrosis nor infiltration in the liver parenchyma at 20 weeks (Supplementary Fig. 2e) and exhibited normalized levels of lymphoid and myeloid populations in peripheral blood without anemia (Supplementary Fig. 2f, g). Rapamycin also normalized glucose tolerance of Li-TSC1$^{KO}$RagA$^{GTP}$ mice (Supplementary Fig. 2h). Finally, while Li-TSC1$^{KO}$RagA$^{GTP}$ mice had a median survival of 34 weeks with spontaneous development of HCC, rapamycin extended the median survival of Li-TSC1$^{KO}$RagA$^{GTP}$ mice to 82 weeks and delayed liver tumor development (Fig. 2F, G).

### Dissection of nutrient and hormone signaling to mTORC1 in hepatocytes
These three genetic tools allow for interrogating whether the signal transduction cascade upstream of nutrients and GF are convergent cues in the control of mTORC1 but independent, as proposed[4], or alternatively, nutrient and growth factor signaling cascades functionally control each other. Thus, we obtained primary hepatocytes from all four genotypes and assessed the effect of deprivation-replenishment of amino acids and insulin in vitro. Consistently with the concept of independent signal transduction cascades, acute starvation-replenishment of amino acids in cultured primary hepatocytes resulted in inhibition-activation of mTORC1 in wild-type and Li-TSC1$^{KO}$, but mTORC1 activity was high in cells expressing GTP-locked RagA regardless of the presence of amino acids (Fig. 3A, C). Conversely, serum withdrawal followed by acute insulin stimulation modulated mTORC1 activity only in primary cells expressing *Tsc1*, regardless of the status of RagA (i.e.,: wild-type and Li-RagA$^{GTP}$; Fig. 3B, C), and Li-TSC1$^{KO}$RagA$^{GTP}$ hepatocytes were insensitive to manipulation of the levels of either insulin or amino acids. Altogether these data support a large independence of nutrient and GF signaling cascades to mTORC1 in hepatocytes.

### Spatial analysis of hepatic nutrient and hormone signaling to mTORC1
Whole-protein extracts from liver samples pool together proteins from all cell types present in the sample (including the profuse inflammatory

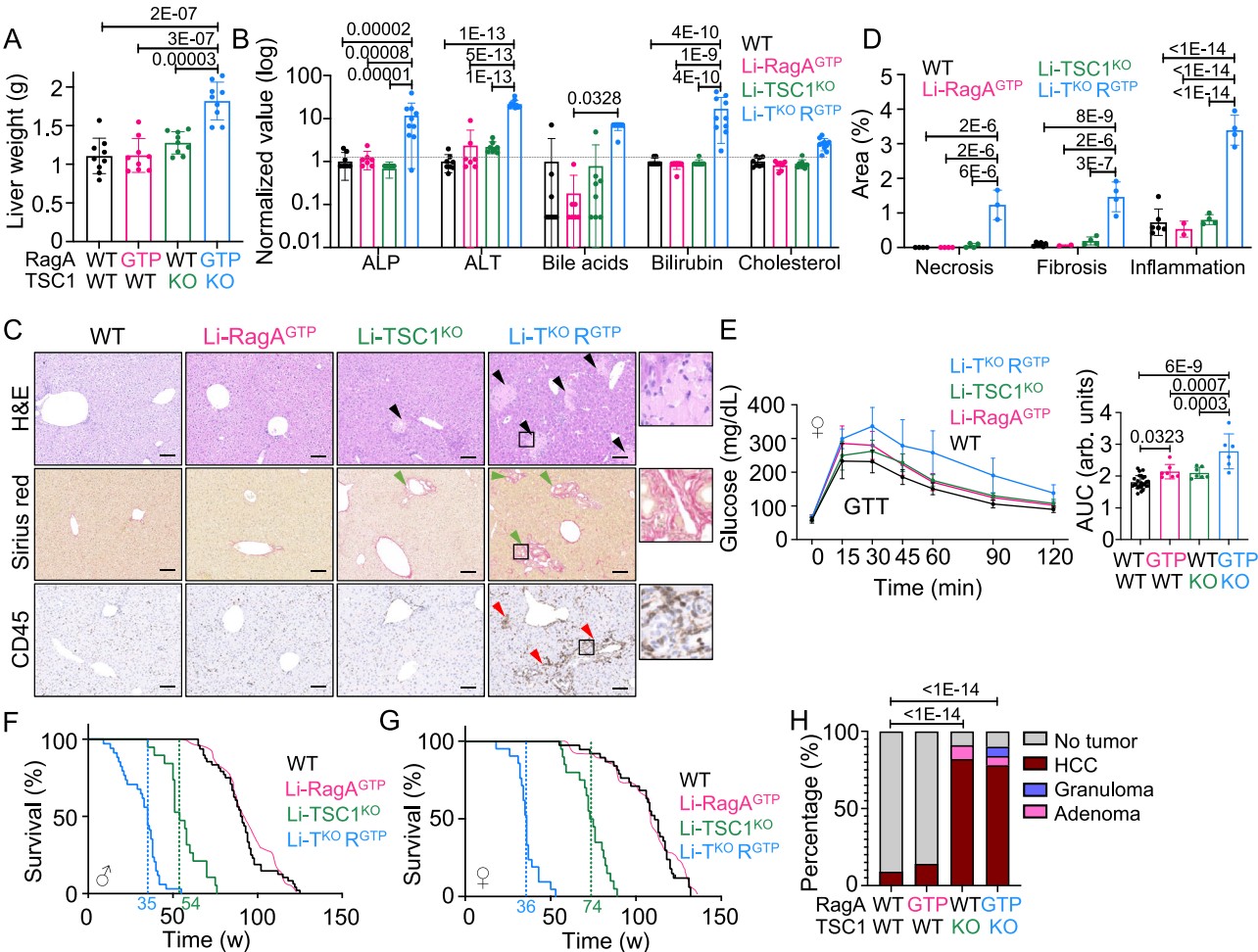

**Fig. 1 | Simultaneous deregulation of nutrient and hormone signaling to mTORC1 leads to synergic liver damage, glucose metabolism alteration and decreased lifespan. A** Liver weight of 10- to 26-week-old wild-type ($n = 9$), Li-RagA[GTP] ($n = 9$), Li-TSC1[KO] ($n = 9$) and Li-TSC1[KO]RagA[GTP] ($n = 9$) male and female mice. Statistical significance was calculated by using 1way ANOVA with Tukey's multiple comparisons test. **B** Levels of circulating alkaline phosphatase (ALP), alanine aminotransferase (ALT), bile acids, bilirubin and cholesterol were measured in 9- to 25-week-old wild-type ($n = 8$), Li-RagA[GTP] ($n = 7$), Li-TSC1[KO] ($n = 8$) and Li-TSC1[KO]RagA[GTP] ($n = 10$) male and female mice. Values were made relative to wild-type average level and presented in $\log_{10}$ scale. Statistical significance was calculated by using 2way ANOVA with Tukey's multiple comparisons test. **C** Representative hepatic H&E, Sirius red staining and CD45 IHC of 10- to 26-week-old wild-type, Li-RagA[GTP], Li-TSC1[KO] and Li-TSC1[KO]RagA[GTP] male and female mice. Black, green and red arrowheads indicate necrotic, fibrotic and inflammatory areas, respectively. Scale bar 100 μm. **D** Quantification of necrosis, fibrosis and inflammation in 10- to 26-week-old wild-type (necrosis $n = 4$, fibrosis/inflammation $n = 6$), Li-RagA[GTP] (necrosis

$n = 4$, fibrosis/inflammation $n = 2$), Li-TSC1[KO] (necrosis/fibrosis/inflammation $n = 4$) and Li-TSC1[KO]RagA[GTP] (necrosis $n = 3$, fibrosis/inflammation $n = 4$) male and female mice. Statistical significance was calculated by using 2way ANOVA with Tukey's multiple comparisons test. **E** Glucose tolerance test (GTT) of 8- to 16-week-old wild-type ($n = 25$), Li-RagA[GTP] ($n = 7$), Li-TSC1[KO] ($n = 7$) and Li-TSC1[KO]RagA[GTP] ($n = 6$) females and area under the curve (AUC) of glucose tolerance test. Statistical significance was calculated by using 1way ANOVA with Tukey's multiple comparisons test. **F, G** Kaplan–Meier survival curves of wild-type ($n = 48$ males, $n = 38$ females), Li-TSC1[KO] ($n = 19$ males, $n = 20$ females) and Li-TSC1[KO]RagA[GTP] ($n = 34$ males and $n = 21$ females) male and female mice, respectively. Mean survival depicted in green for Li-TSC1[KO] and blue for Li-TSC1[KO] RagA[GTP]. Mean survival of Li-RagA[GTP] mice from ref. 17 is depicted in pink. Statistical significance was calculated with the log-rank (Mantel–Cox) test. **H** Heterogeneous liver tumor development in wild-type ($n = 41$), Li-RagA[GTP] ($n = 33$), Li-TSC1[KO] ($n = 11$) and Li-TSC1[KO]RagA[GTP] ($n = 18$) male and female mice. Statistical significance was calculated with chi-squared test. In all panels data are presented as mean values ± standard deviation.

cells in Li-TSC1[KO]RagA[GTP] livers) and limit both the precise assessment of mTORC1 activity in hepatocytes and the dissection of potential differences in mTORC1 activity in different groups of hepatocytes, such as those from portal zone 1 (Z1) or central zone 3 (Z3) of the hepatic lobule. Thus, to capture potential signaling heterogeneity in different zones of the mammalian liver, we performed co-immunostainings of phospho-S6 together with Glutamine synthetase (GS), a canonical Z3 marker, to allow spatial recognition of Z3 and Z1, and we quantified phosphorylation of phospho-S6 by immuno-fluorescence in the different hepatic zones. A prominent increase in phospho-S6 in Z3 was evident by a 2 h refeeding of wild-type mice, but less so in Z1 and the Z2 in between (Fig. 3D, E). Consistently, genetic activation of either hormone or nutrient signaling alone was sufficient to significantly increase phospho-S6 in Z3 in the fasting state, but this

increase was less pronounced in Z1 and Z2 (Fig. 3D, F), suggesting that Z3 is comparably more sensitive to both genetic and physiological inputs that control the activation of mTORC1. In contrast, concomitant genetic activation of nutrient and hormone signaling to mTORC1 resulted in a significant increase in phospho-S6 in all liver zones, as compared to single mutants (Fig. 3D, F). In consistence with the western blot data (Fig. 2A), increased activation of mTORC1 signaling was found by IF on livers from Li-TSC1[KO]RagA[GTP] mice *versus* single mutants (Supplementary Fig. 3a) during fasting. Moreover, phospho-S6 was higher in all zones from Li-TSC1[KO]RagA[GTP] livers compared to the other genotypes in samples from 2 h refed mice (Supplementary Fig. 3b). To confirm the conclusions drawn by the IF staining of phospho-S6, we complemented this biochemical readout of mTORC1 activity with a cellular readout downstream of mTORC1 activation: increased cell

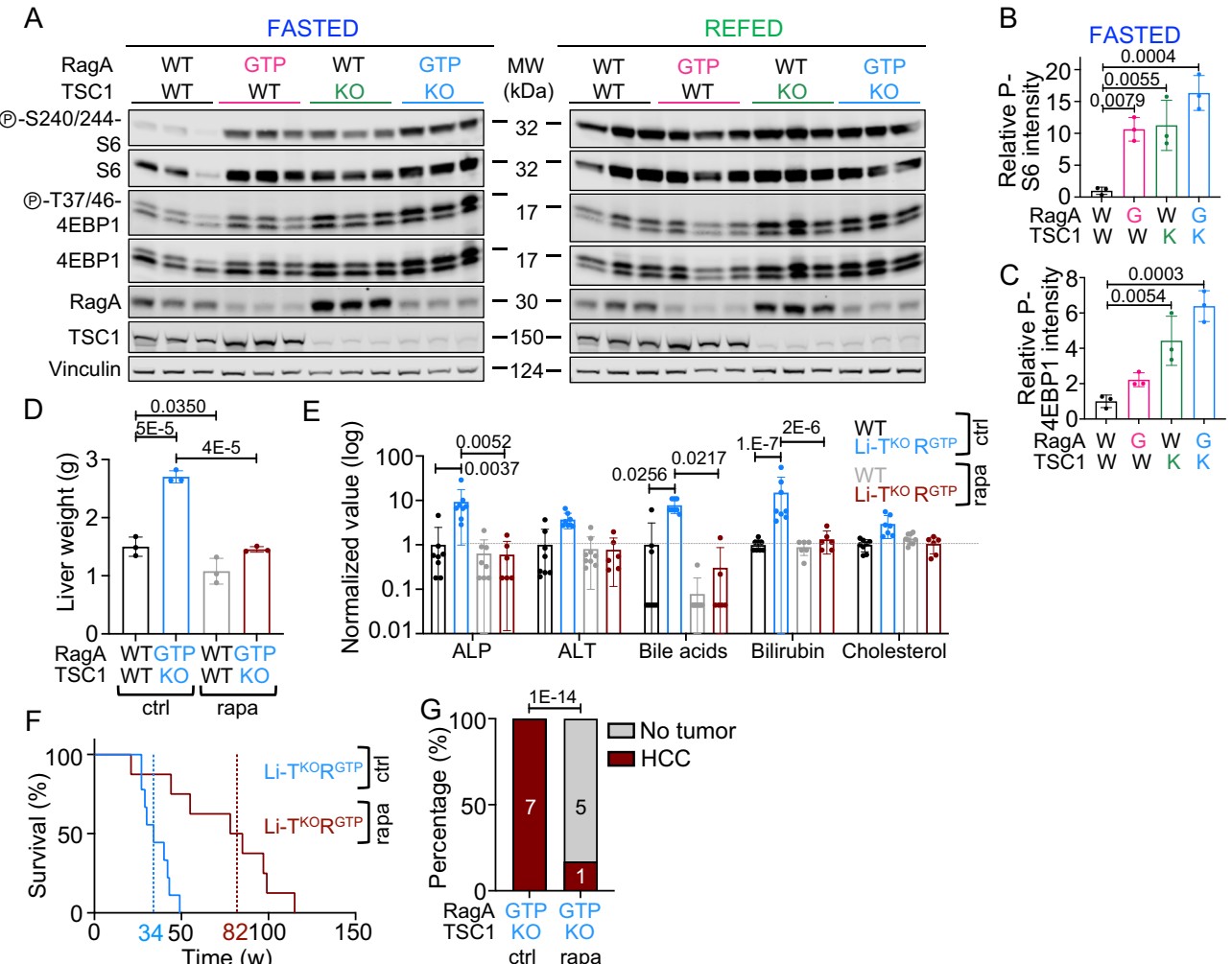

**Fig. 2 | Pharmacological inhibition of mTORC1 in Li-TSC1^KO RagA^GTP mice suppresses liver damage and extends lifespan. A** 10- to 26-week-old control, Li-RagA^GTP, Li-TSC1^KO and Li-TSC1^KO RagA^GTP females were deprived from food for 24 h, half of them following a 2 h refeeding, and then sacrificed. Protein lysates from the liver were immunoblotted for the indicated proteins. **B** Levels of Phospho-S240-244-S6 from fasted state on each lane from A are relative to vinculin levels and presented normalized to the average level of fasted wild-type mice (n = 3). Statistical significance was calculated by using 1way ANOVA with Tukey's multiple comparisons test. **C** Levels of Phospho-T37/46-4EBP1 from fasted state on each lane from A are relative to vinculin levels and presented normalized to the average level of fasted wild-type mice (n = 3). Statistical significance was calculated by using 1way ANOVA with Tukey's multiple comparisons test. **D** Liver weight of 18- to 20-week-old wild-type control (n = 3), wild-type rapamycin (n = 3), Li-TSC1^KO RagA^GTP control (n = 3) and Li-TSC1^KO RagA^GTP rapamycin (n = 3) male mice. Statistical significance was calculated by using 2way ANOVA with Sidák's multiple comparisons test.

**E** Levels of circulating alkaline phosphatase (ALP), alanine aminotransferase (ALT), bile acids, bilirubin and cholesterol were measured in aged WT control (n = 8), WT rapamycin (n = 9), Li-TSC1^KO RagA^GTP control (n = 8) and Li-TSC1^KO RagA^GTP rapamycin (n = 6) mice. Values were made relative to the average level in wild-type mice in control condition and presented in log₁₀ scale. Statistical significance was calculated by using 2way ANOVA with Tukey's multiple comparisons test.
**F** Kaplan–Meier survival curves of Li-TSC1^KO RagA^GTP control (n = 9) and Li-TSC1^KO RagA^GTP rapamycin (n = 8) mice. Mean survival depicted in blue for control Li-TSC1^KO RagA^GTP and garnet for rapamycin-treated Li-TSC1^KO RagA^GTP mice. Statistical significance was calculated with the log-rank (Mantel–Cox) test.
**G** Heterogeneous tumor development in the liver of control (n = 7) and rapamycin-treated (n = 6) Li-TSC1^KO RagA^GTP mice. Statistical significance was calculated with chi-squared test. In all panels, data are presented as mean values ± standard deviation.

size[19,20]. In agreement with an increased sensitivity of central Z3 hepatocytes to mTORC1-activating stimuli, hepatocytes within the central Z3 region were bigger in mice bearing either genetic activation upstream of mTORC1, namely Li-RagA^GTP, Li-TSC1^KO and Li-TSC1^KO RagA^GTP, compared to wild-type hepatocytes (Fig. 3G). An enlarged cell size in hepatocytes of Z1 and Z2 was significant for Li-TSC1^KO RagA^GTP hepatocytes, but less noticeable in the single mutants (Fig. 3G). Further supporting an increased sensitivity of Z3 hepatocytes to physiological fluctuations in nutrients and GF, changes in cell size in *ad libitum* fed *versus* fasted wild-type hepatocytes were more pronounced in Z3 (Supplementary Fig. 3c). The sensitivity of Z3 hepatocytes to fasting-feeding cycles and to genetic activation of mTORC1 correlated with increased expression of several (but not all[21])

components of the signal transduction cascade upstream of mTORC1 in Z3 hepatocytes, as assessed from published single-cell transcriptomics data[22] (Supplementary Fig. 3d). In summary, Z3 hepatocytes appear comparatively more sensitive to feeding cycles and genetic manipulation of regulators of mTORC1 (nutrient and GF signaling), a surprising observation considering that portal Z1 hepatocytes are directly irrigated by blood afferent from the gastrointestinal tract and the pancreas and thus, exposed to the fluctuations associated to feeding intermittency.

**Impairment of metabolic zonation in Li-TSC1^KO RagA^GTP mice**
To unveil molecular underpinnings of the metabolic phenotype of Li-TSC1^KO RagA^GTP mice, we conducted bulk RNA sequencing from fasted

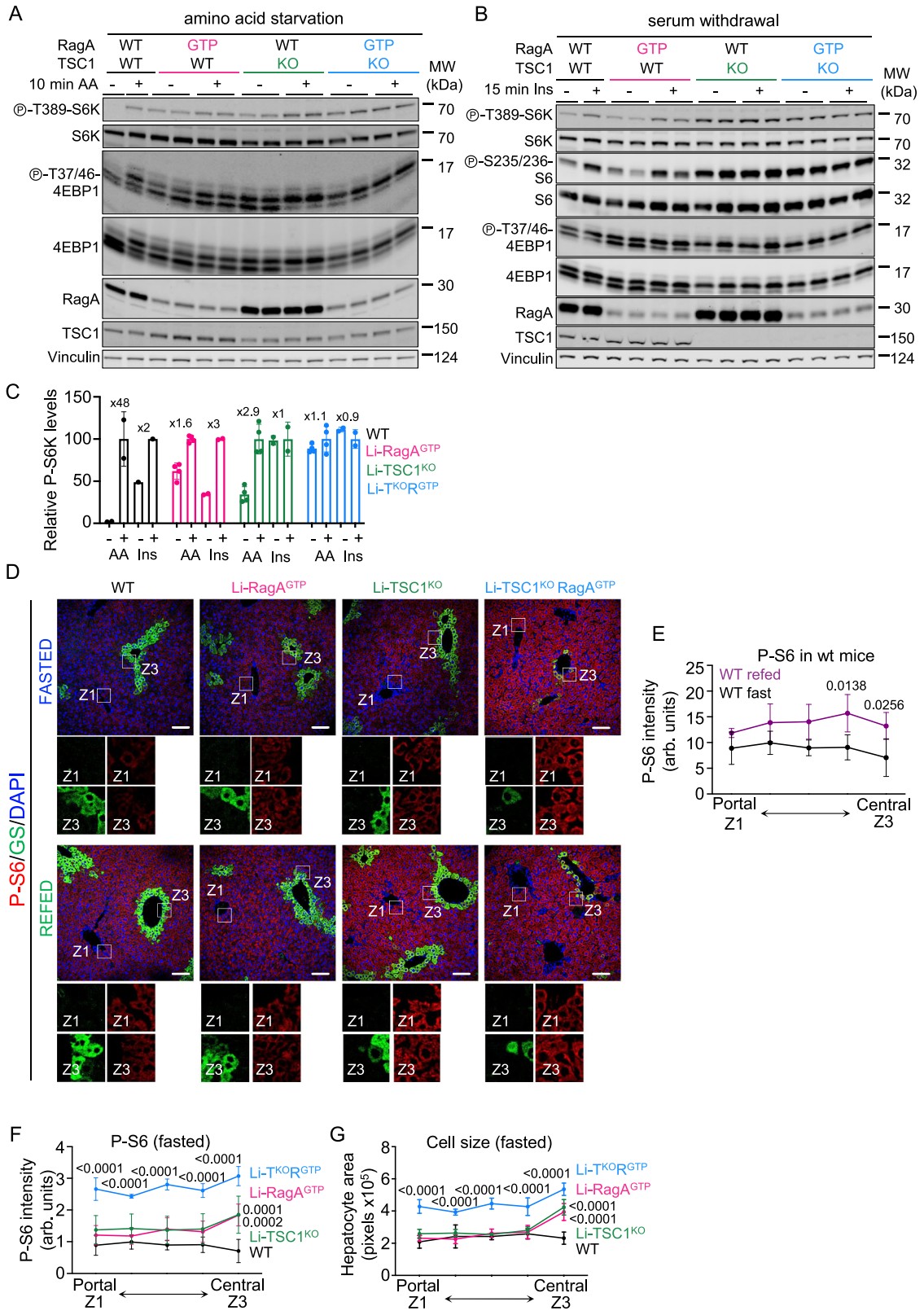

and refed wild-type, Li-TSC1^KO, and Li-TSC1^KORagA^GTP livers with and without rapamycin (Supplementary Fig. 4a). Principal component analysis (PCA) clustered samples based on genotype (Supplementary Fig. 4b). Interestingly, while fasting-refeeding resulted in a clear separation of wild-type samples in a two-dimensional PCA, such separation was absent in samples from Li-TSC1^KORagA^GTP mice,

suggesting profound transcriptomic changes largely insensitive to fasting-feeding (Supplementary Fig. 4b). Despite the protective effects of rapamycin against the aberrant phenotype of Li-TSC1^KORagA^GTP mice (Fig. 2D–G and Supplementary Fig. 2e-h), and a depletion of mTORC1 gene signature by a 3-day pharmacological inhibition of mTORC1 (Supplementary Fig. 4c), this acute treatment was insufficient to

**Fig. 3 | Nutrients and GF cascades are convergent but independent cues in the control of mTORC1 in the liver. A** Primary hepatocytes from 7- to 14-week-old control wild-type, Li-RagA$^{GTP}$, Li-TSC1$^{KO}$ and Li-TSC1$^{KO}$RagA$^{GTP}$ male and female mice were deprived from all amino acids in DMEM/F12 for 1 h and re-stimulated with amino acids for 10 min. **B** Primary hepatocytes from 7- to 9-week-old wild-type, Li-RagA$^{GTP}$, Li-TSC1$^{KO}$ and Li-TSC1$^{KO}$RagA$^{GTP}$ male and female mice were deprived of serum in DMEM/F12 for 16 h and re-stimulated with insulin for 15 min. **C** Levels of Phospho-T389-S6K on each lane from A and B are relative to vinculin levels and presented normalized to the maximum activation levels of each genotype ($n = 2$). Fold change increase after amino acids or insulin stimulation is presented for each genotype. **D** Representative pictures of immunofluorescence against Phospho-S240-244-S6 and Glutamine Synthetase in the liver of 10- to 26-week-old control wild-type, Li-RagA$^{GTP}$, Li-TSC1$^{KO}$ and Li-TSC1$^{KO}$RagA$^{GTP}$ male and female mice fasted during 24 h or fasted during 24 h followed by 2-h of refed. Zone 1 and Zone 3 are highlighted in each image. Scale bar 100 μm. **E** Quantification of the Phospho-S240/244-S6 intensity of 10- to 26-week-old wild-type male and female mice fasted during 24 h ($n = 4$) or fasted during 24 h followed by a 2-h of refed ($n = 4$) from portal to central zones. Statistical significance was calculated by using 2way ANOVA with Sidák's multiple comparisons test. **F, G** Quantification of the Phospho-S240/244-S6 intensity and hepatocyte area, respectively, of 10- to 26-week-old wild-type ($n = 4$), Li-RagA$^{GTP}$ ($n = 4$), Li-TSC1$^{KO}$ ($n = 4$) and Li-TSC1$^{KO}$RagA$^{GTP}$ ($n = 3$) male and female mice fasted during 24 h from portal to central zones. Statistical significance was calculated by using 2way ANOVA with Tukey's multiple comparisons test. $P < 0.0001$ indicate the statistical significance between the comparisons of L-TR vs L-R, L-T and WT. In all panels, data are presented as mean values ± standard deviation.

separate treated and untreated Li-TSC1$^{KO}$RagA$^{GTP}$ samples by PCA (Supplementary Fig. 4b), suggesting that chronic mTORC1 activation may be only rescuable by prolonged inhibition of the pathway that would reverse profound alterations such as inflammation, as seen in Fig. 2.

While 1506 genes were up- and down-regulated in wild-type livers in fasted *versus* refed states, only 71 genes were differentially expressed in fasted *versus* refed Li-TSC1$^{KO}$RagA$^{GTP}$ livers demonstrating the large effect of mTORC1 in the physiological response to fasting-feeding cycles. Rapamycin significantly changed the expression of 621 genes in fasted mutant livers. Finally, almost 3000 genes were significantly up- or down-regulated inLi-TSC1$^{KO}$RagA$^{GTP}$ *versus* wild-type livers, showing the large impact of concomitant activation of nutrient and hormone signaling to mTORC1 (Supplementary Data 1), as compared to the activation of single arms in the pathway (Supplementary Fig. 4d)[17,18]. Indeed, most changes in Li-RagA$^{GTP}$ and in Li-TSC1$^{KO}$ are contained within the changes observed in Li-TSC1$^{KO}$RagA$^{GTP}$, with additional changes present exclusively in Li-TSC1$^{KO}$RagA$^{GTP}$ livers (Supplementary Fig. 4d).

We next conducted paired Gene Set Enrichment Analysis (GSEA)[23] between fasted wild-type and Li-TSC1$^{KO}$RagA$^{GTP}$ samples. Li-TSC1$^{KO}$RagA$^{GTP}$ livers showed increased expression of genes belonging to signatures of mTORC1 activity, glycolysis, reactive oxygen species, and curated signatures of inflammation and fibrosis (Fig. 4A, Supplementary Fig. 4e, f and Supplementary Data 1). Several signatures of key metabolic functions of the liver were depleted from Li-TSC1$^{KO}$RagA$^{GTP}$ samples, including fatty-acid, bile-acid and xenobiotic metabolism, peroxisome and oxidative phosphorylation, among others, and PPARα targets (Fig. 4B and Supplementary Fig. 4g). These signatures were also significantly different when comparing Li-TSC1$^{KO}$RagA$^{GTP}$ *versus* Li-TSC1$^{KO}$ livers (Supplementary Fig. 4c–g). Most metabolic functions in the liver have a zonated pattern of expression, so we curated a central Z3 and a portal Z1 metabolic signature based on the gene expression patterns described on published work[22,24] (Supplementary Table 1). Interestingly, Li-TSC1$^{KO}$RagA$^{GTP}$ samples displayed a profound deregulation of zonated genes (Fig. 4C), showing a striking depletion of expression of central genes, and an enrichment of portal genes (Fig. 4D), suggesting a partial loss of central Z3 identity and a partial portalization of the Li-TSC1$^{KO}$RagA$^{GTP}$ livers.

We validated the existence of an aberrant pattern of zonated gene expression by qPCR, and observed decreased expression of the canonical Z3 genes (*Cyp2e1* and *Glul*), and partial changes in mRNAs from portal Z1 (Fig. 4E). The protein levels of Glutamine synthetase (GS, the product of *Glul* gene), normally restricted to the first layers of Z3 hepatocytes, were reduced and anomalously dispersed in Li-TSC1$^{KO}$RagA$^{GTP}$ livers, as revealed by immunofluorescence (Fig. 4F, G), and this reduction co-occurred with an expansion of the portal Z1 marker E-cadherin (Fig. 4F), corroborating a molecular periportalization of Li-TSC1$^{KO}$RagA$^{GTP}$ livers. Contraction of Z3 and expansion of Z1

were confirmed with by independent markers OAT (for Z3) and PCK 1 (for Z1) (Supplementary Fig. 4h). Such loss of zonation is unlikely to stem from zone-specific liver damage, as foci of cellular damage in Li-TSC1$^{KO}$RagA$^{GTP}$ mice were not restricted to portal or central zones (Supplementary Fig. 4i).

We further unbiasedly validated the transcriptomic results, and particularly the altered expression of zonated metabolic genes by means of whole-cell proteomics from wild-type, Li-RagA$^{GTP}$, Li-TSC1$^{KO}$ and Li-TSC1$^{KO}$RagA$^{GTP}$ livers (Supplementary Fig. 4j and Supplementary Data 1), extending our previous proteomics dataset from Li-RagA$^{GTP}$ and Li-TSC1$^{KO}$ from ref. 17 to double Li-TSC1$^{KO}$RagA$^{GTP}$ livers. Compared to 300+ differentially abundant proteins in Li-RagA$^{GTP}$ and 1000+ differentially abundant proteins in Li-TSC1$^{KO}$ livers *versus* wild-type livers[17], the levels of 3133 proteins were significantly up- or down-regulated in Li-TSC1$^{KO}$RagA$^{GTP}$ livers (Supplementary Fig. 4k), highlighting the differential impact that simultaneous activation of nutrient and hormone signaling exerts on the mouse liver. In sharp consistence with the transcriptomic analysis, GSEA analysis from proteomic data showed enrichment in signatures related to mTORC1, interferon response and other inflammatory signatures in Li-TSC1$^{KO}$RagA$^{GTP}$, and depletion of signatures of fatty-acid and bile-acid metabolism, oxidative phosphorylation, and peroxisome (Supplementary Fig. 4l). Moreover, the levels of proteins expressed in central Z3 were significantly depleted and those in Z1 significantly enriched in Li-TSC1$^{KO}$RagA$^{GTP}$ livers in paired comparisons with wild-type samples (Fig. 4H). Despite a mild depletion of Z3 gene signature in single mutant mice (Supplementary Fig. 4m), the abnormal zonated pattern of expression was not evident in single Li-RagA$^{GTP}$ nor in Li-TSC1$^{KO}$ mice (Supplementary Fig. 4n), neither single-mutant mice exhibited loss of zonated expression of GS and E-cadherin in immunostaining (Supplementary Fig. 4o, p). Rapamycin restored metabolic zonation (Supplementary Fig. 4q, r), arguing that this difference was caused by deregulated mTORC1 activity in Li-TSC1$^{KO}$RagA$^{GTP}$ livers.

## Down-regulation of Wnt/β-catenin signaling in Li-TSC1$^{KO}$RagA$^{GTP}$ mice

The Wnt/β-catenin signaling pathway is critical for the specification of the zonated metabolism in the liver. Wnt ligands and β-catenin activity dissipate in a central-to-portal gradient and this gradient establishes and maintains zonated gene expression, as many central genes are under transcriptional control of this pathway[24–27]. We mined our bulk transcriptomic data for expression of canonical Wnt targets and Wnt ligands, and both a curated "Wnt/β-catenin related genes" signature and the Hallmark signature "Wnt beta catenin signaling" were depleted in samples from adult Li-TSC1$^{KO}$RagA$^{GTP}$ mice compared to wild-type livers (Fig. 5A, Supplementary Fig. 5a, and Supplementary Table 2). Our proteomics analysis confirmed the deregulation of a β-catenin signature in Li-TSC1$^{KO}$RagA$^{GTP}$ mice, but not in single mutant animals (Supplementary Fig. 5b). The altered expression of Wnt/β-catenin targets in Li-TSC1$^{KO}$RagA$^{GTP}$ livers was confirmed by RT-qPCR (Fig. 5B and Supplementary Fig. 5c for females and males, respectively).

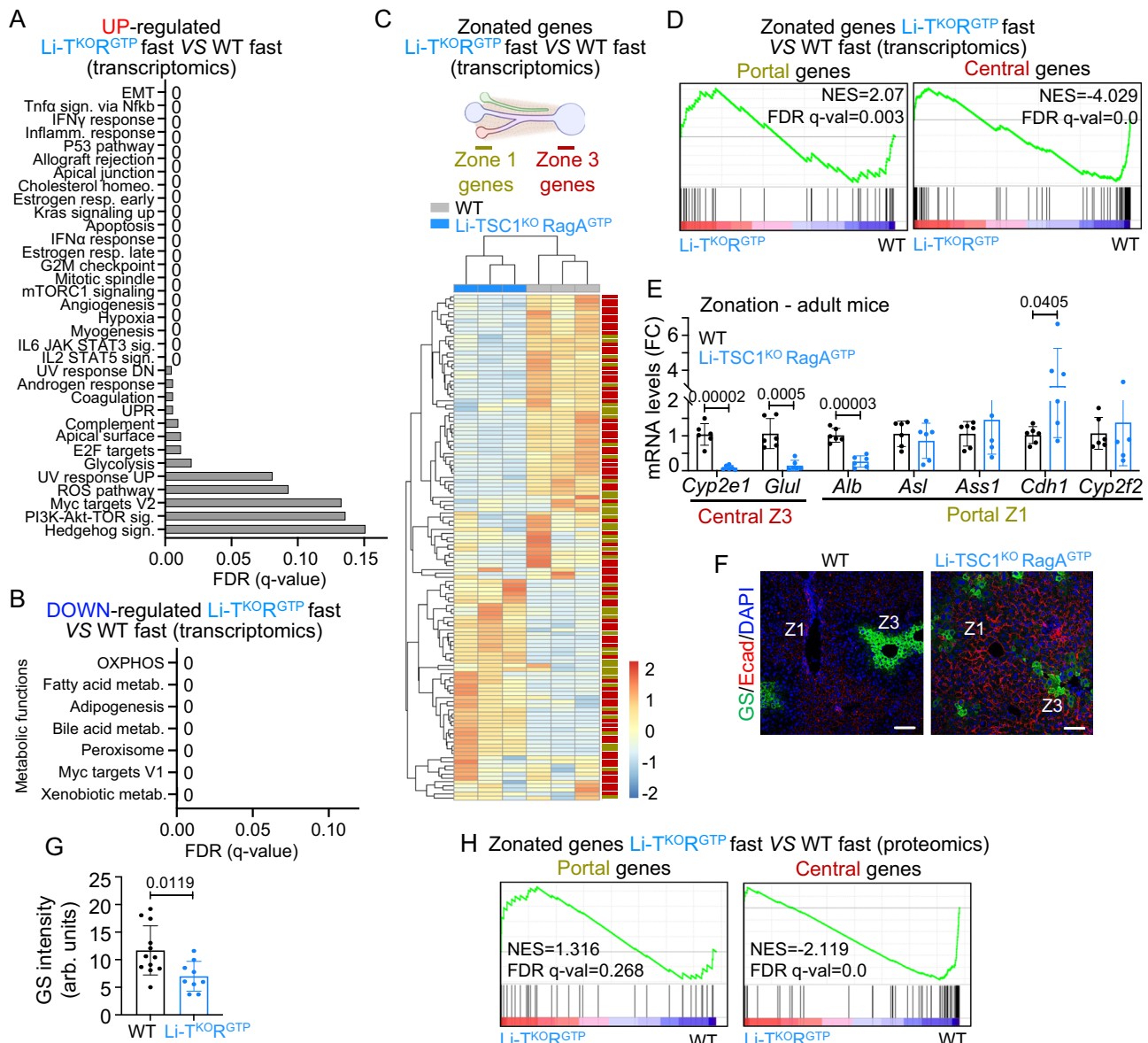

**Fig. 4 | Metabolic zonation is impaired upon concomitant activation of nutrient and hormonal arms of mTORC1 in the liver. A** Representation of the false discovery rates (FDRs) from the top Hallmark gene sets enriched in livers from 10- to 26-week-old Li-TSC1^KORagA^GTP (*n* = 3) *versus* wild-type (*n* = 3) female mice in 24 h fasting. **B** Representation of the false discovery rates (FDRs) from the top Hallmark gene sets downregulated in livers from 10- to 26-week-old Li-TSC1^KORagA^GTP (*n* = 3) *versus* wild-type (*n* = 3) female mice in 24 h fasting. **C** Hierarchical clustering Heatmap diagram representing mRNA expression patterns of Li-TSC1^KORagA^GTP (*n* = 3) and wild-type (*n* = 3) livers normalized by z-score. Belonging to Z3 or Z1 is indicated at the right of each gene. Created with BioRender.com. **D** Enrichment of gene sets related to central and portal signatures in transcriptomics from wild-type (*n* = 3) and Li-TSC1^KORagA^GTP (*n* = 3) livers. NES normalized enrichment score, FDR: false discovery rate. **E** RT-qPCR of livers from 8- to 21-week-old wild-type (*n* = 6) and

Li-TSC1^KORagA^GTP (*n* = 6) male and female mice. Expression levels of the indicated genes involved in zonation relative to the average level in wild-type mice. β-actin was used as housekeeping gene. Statistical significance was calculated by using multiple unpaired two-tailed *t*-test. **F** Representative pictures of immuno-fluorescence against Glutamine synthetase together with E-cadherin in the liver of 10- to 26-week-old wild-type and Li-TSC1^KORagA^GTP female mice. Zone 1 or Zone 3 are highlighted in each image. Scale bar 100 µm. **G** Quantification of the Glutamine synthetase intensity of wild-type (*n* = 12) and Li-TSC1^KORagA^GTP (*n* = 9) livers specifically in the central zone. Statistical significance was calculated by using unpaired two-tailed *t*-test. **H** Enrichment of gene sets related to central and portal signatures in proteomics from wild-type (*n* = 4) and Li-TSC1^KORagA^GTP (*n* = 4) livers. NES normalized enrichment score, FDR false discovery rate. In all panels, data are presented as mean values ± standard deviation.

Among the Wnt ligands quantified (*Wnt2*, *Wnt3a*, *Wnt4*, *Wnt5a*, *Wnt5b*, *Wnt7a* and *Wnt9b*), only *Wnt2* levels were significantly decreased in Li-TSC1^KORagA^GTP livers (Fig. 5C and Supplementary Fig. 5d for females and males, respectively). Wnt2 was not detected in the proteomics analysis (Supplementary Data 1), but its decrease was validated by western blot (Supplementary Fig. 5e, f). Collectively, these data point to deregulated Wnt/β-catenin signaling with diminished secretion of Wnt2 ligand in Li-TSC1^KORagA^GTP liver and partial loss of metabolic spatial identity.

The Wnt/β-catenin signaling pathway is not autonomously established by a gradient self-imposed by hepatocytes. Instead, Wnt ligands are synthesized mainly by liver endothelial cells (EC) and other non-parenchymal cells[28–30] in paracrine crosstalk with hepatocytes. To deconvolute a multicellular communication potentially relevant for the loss of zonation in Li-TSC1^KORagA^GTP mice, we conducted single-cell RNA sequencing (scRNAseq) from two wild-type and two Li-TSC1^KORagA^GTP liver samples (Supplementary Fig. 5g and Supplementary Data 1) utilizing a cell purification protocol that enriches for cells

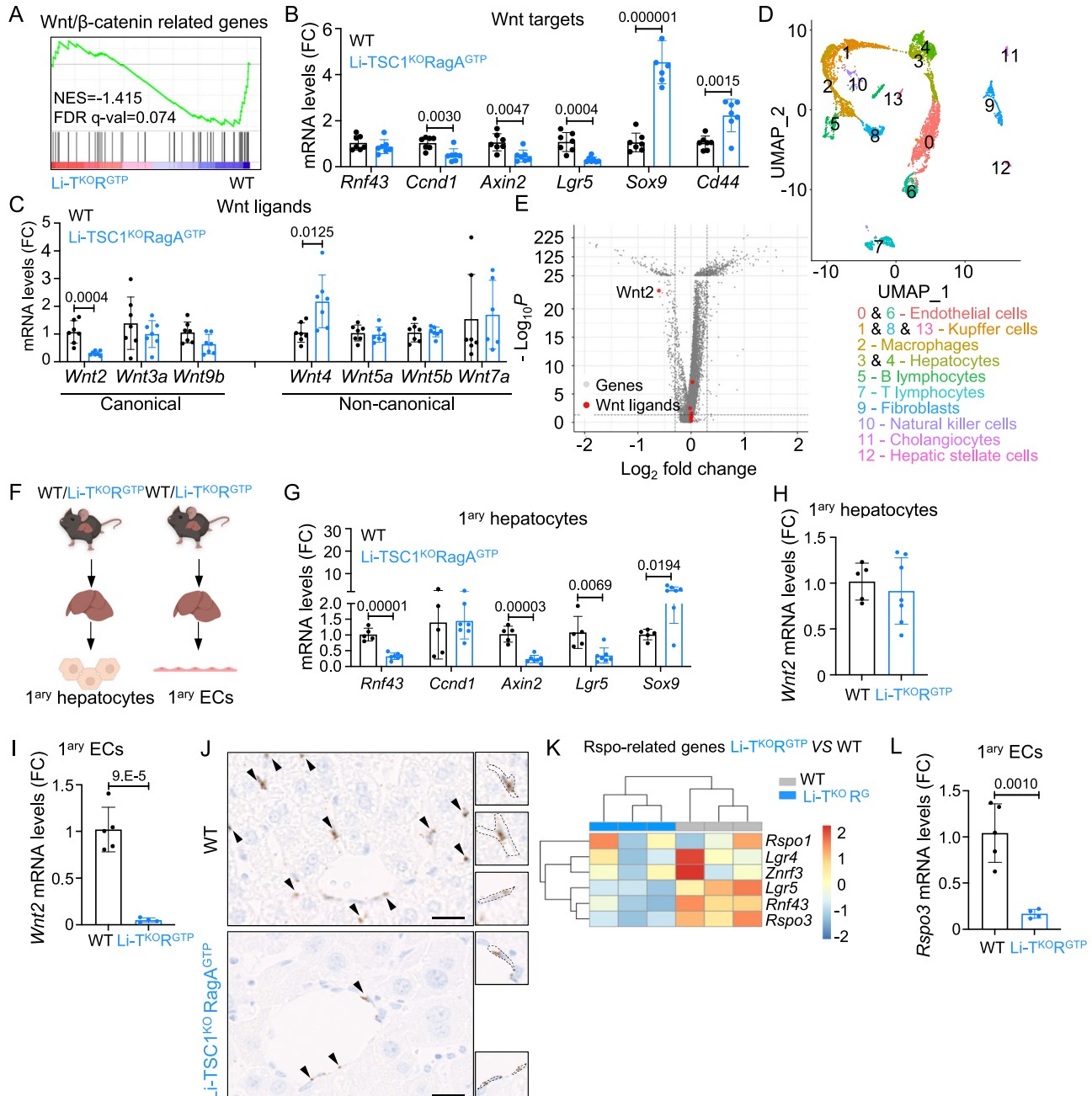

**Fig. 5 | Wnt/β-catenin pathway is down-regulated in Li-TSC1^KORagA^GTP livers.**
**A** Enrichment plot for "Wnt/β-catenin related genes" signature for 10- to 26-week-old Li-TSC1^KORagA^GTP *versus* wild-type livers from female mice fasted for 24 h followed by 2 h of refeeding at the mRNA level. NES normalized enrichment score, FDR false discovery rate. **B**, **C** RT-qPCR of livers from 17- to 25-week-old wild-type ($n = 7$) and Li-TSC1^KORagA^GTP ($n = 7$) female mice. Expression levels of the indicated genes relative to the average level in wild-type mice. β-actin was used as housekeeping gene. Statistical significance was calculated by using multiple unpaired two-tailed *t*-test. **D** Uniform Maniform Approximation and Projection for dimension reduction (UMAP) plot of liver cells showing 14 clusters. **E** Volcano plot highlighting Wnt ligands detected in EC clusters (0 + 6). *Y*-axis denotes - log10 *P*-values while *X*-axis shows Log2 FC values. Dotted lines represent filtering cut-off of log2 fold change above 0.3 and *p*-value below 0.05. **F** Primary hepatocytes/EC were isolated (Created with BioRender.com). **G**, **H** RT-qPCR of primary hepatocytes from 7-week-old wild-type ($n = 5$) and Li-TSC1^KORagA^GTP ($n = 7$) male/female mice. Expression levels of the indicated

genes relative to the average level in wild-type mice. β-actin was used as housekeeping gene. Statistical significance was calculated by using multiple unpaired two-tailed *t*-test. **I** RT-qPCR of primary EC from 8- to 15-week-old wild-type ($n = 5$) and Li-TSC1^KORagA^GTP ($n = 4$) mice. Expression levels of Wnt2 relative to the average level in control mice. β-actin was used as housekeeping gene. Statistical significance was calculated by using unpaired two-tailed *t*-test. **J** Representative pictures of RNAscope against mWnt2 in the liver of 4-week-old wild-type and Li-TSC1^KORagA^GTP male/female mice. Scale bar 20 μm. Insets highlight mWnt2 positive EC. **K** Hierarchical clustering Heatmap diagram representing mRNA expression of Rspo module genes in EC from Li-TSC1^KORagA^GTP ($n = 3$) and wild-type ($n = 3$) livers normalized by z-score. **L** RT-qPCR of primary EC from 8- to 15-week-old wild-type ($n = 5$) and Li-TSC1^KORagA^GTP ($n = 4$) male/female mice. Expression levels of Rspo3 relative to the average level in control mice. β-actin was used as housekeeping gene. Statistical significance was calculated by using unpaired two-tailed t-test. In all panels, data are presented as mean values ± standard deviation.

other than hepatocytes[31,32]. Two small clusters of hepatocytes, plus two clusters composed by EC, a cluster of cholangiocytes and several clusters of inflammatory cells were identified (Fig. 5D). Analysis of the EC clusters 0 and 6 revealed depletion of a Wnt ligand signature in Li-TSC1^{KO}RagA^{GTP} samples (Supplementary Fig. 5h), with *Wnt2* ligand remarkably depleted (Fig. 5E). *Wnt2* ligand was not detected in the rest of the clusters identified by scRNAseq analysis, and the Wnt ligands observed in other clusters did not significantly change their expression levels in samples from Li-TSC1^{KO}RagA^{GTP} *versus* wild-type livers (Supplementary Fig. 5i). Based on the transcriptomic data from bulk and scRNAseq, we reasoned that the dissipation of the molecular pattern of central hepatocyte identity could be consequence of a paracrine communication between Li-TSC1^{KO}RagA^{GTP} hepatocytes and EC. Indeed, ex vivo isolated hepatocytes from wild-type and Li-TSC1^{KO}RagA^{GTP} mice showed, as expected, a suppressed Wnt/β-catenin transcriptional program (Fig. 5F, G), but equally low mRNA levels of *Wnt2* ligand regardless of the genotype (Fig. 5H). A reduction in Wnt pathway targets with similar levels of Wnt ligands in hepatocytes was consistent with the scRNAseq data supporting reduced production and secretion of Wnt2 ligand from EC. Thus, we next isolated liver EC from wild-type and Li-TSC1^{KO}RagA^{GTP} mice (Fig. 5F), and observed that the expression of *Wnt2* was 100x higher in EC compared to hepatocytes (Supplementary Fig. 5j). More importantly, the expression of *Wnt2* ligand in purified EC from Li-TSC1^{KO}RagA^{GTP} mice was remarkably suppressed in comparison to their levels in EC from wild-type mice (Fig. 5I). The enrichment of cell populations of purified primary hepatocytes and EC was validated by RT-qPCR (Supplementary Fig. 5k). RNAscope imaging confirmed decreased *Wnt2* ligand transcript levels in EC from Li-TSC1^{KO}RagA^{GTP} livers (Fig. 5J). In addition, the RSPO signaling module can also paracrinally amplify Wnt signaling pathway activation in hepatocytes[33,34], and Rspo3 was also decreased in the EC cluster of Li-TSC1^{KO}RagA^{GTP} mice (Fig. 5K) and in purified EC (Fig. 5L).

To ascertain whether TSC1^{KO}RagA^{GTP} hepatocytes had an intrinsic inability to transduce paracrine Wnt signals, we supplemented cultured primary hepatocytes with recombinant Wnt2, recombinant Rspo1 and both (Supplementary Fig. 5l). Strikingly, supplementation of Rspo1 induced approximate 5-fold transactivation of the Wnt/β-catenin pathway in mutant primary hepatocytes, with a minimal effect on wild-type counterparts. The effect of Wnt2 supplementation was around 200x, similar to the effect of concomitant addition of Wnt2 plus Rspo1. Thus, we conclude that, if exogenously supplemented to isolated hepatocytes, Wnt ligands can boost the decreased Wnt pathway activation of TSC1^{KO}RagA^{GTP} hepatocytes (Supplementary Fig. 5m–r). These results strongly imply a hepatocyte-endothelium-hepatocyte communication in the impaired metabolic zonation of Li-TSC1^{KO}RagA^{GTP} mice.

## Abrogation of postnatal establishment of the spatial segregation of liver by constitutive hepatic mTORC1 signaling

Liver development, including the morphogenesis of portal vein and hepatic artery double vascular bedding that irrigates the liver, occurs entirely *in utero*. The Albumin-Cre strain[35] used herein is strongly expressed in hepatocytes after birth and with only mosaic, minimal expression *in utero* starting at E15.5[36]. Thus, *in utero* genetic activation of mTORC1 is very unlikely to impair the liver morphogenesis starting at mid-embryonic development. Moreover, nutrient and hormone signaling cascades upstream of mTORC1 started to be conspicuously deregulated in TSC1^{KO}RagA^{GTP} hepatocytes after birth, but mTORC1 activity was not increased in Li-TSC1^{KO}RagA^{GTP} at E19.5 (Supplementary Fig. 6a). Indeed, liver shape and the macroscopic and microscopic histoarchitecture of hepatic lobules at birth were identical in Li-TSC1^{KO}RagA^{GTP} and wild-type mice, including morphological central and portal zones and the portal triad (Supplementary Fig. 6b). In contrast, at the molecular level, a zonated pattern of gene expression is

not yet present at birth, and starts to become evident only days or weeks after birth in mammals[37]. In particular, restriction of GS expression occurs at E18 and becomes increasingly limited to the terminal perivenous hepatocytes only postnatally[38,39]. To distinguish between the dissipation of a zonated pattern of expression in adult mice *versus* an impairment in the establishment of the early postnatal zonation in Li-TSC1^{KO}RagA^{GTP} mice, we obtained livers from neonates, and serially from postnatal day p5, p10, p14 and p28 in wild-type and Li-TSC1^{KO}RagA^{GTP} mice. Both wild-type and Li-TSC1^{KO}RagA^{GTP} mice were born without zonated pattern of expression of GS, OAT, E-Cadherin and PCK1 (Fig. 6A and Supplementary Fig. 6c). In addition, the levels of expression of portal genes were not increased, neither were the central markers decreased, in livers from E19.5 in wild-type and Li-TSC1^{KO}RagA^{GTP} mice (Fig. 6A–C and Supplementary Fig. 6d–f). Moreover, levels of the *Wnt2* ligand and Wnt target gene *Axin2* were also similar in livers of both genotypes at E19.5 (Fig. 6D, E). Livers from wild-type mice started to show portal segregation of E-Cadherin and PCK1 levels at p10, and central segregation of GS and OAT staining at p5, and central and portal zones were fully defined molecularly at p28 (Fig. 6A and Supplementary Fig. 6c). In contrast, this gradual establishment of zonation as well as the Wnt pathway activation were markedly impaired in livers from Li-TSC1^{KO}RagA^{GTP} mice (Fig. 6A–E and Supplementary Fig. 6c–h). Expression of *Wnt2* ligand was also diminished in EC from Li-TSC1^{KO}RagA^{GTP} mice already by one week after birth, as revealed by RNAscope hybridization (Supplementary Fig. 6i), and by quantification of *Wnt2* mRNA levels from EC isolated from p14 wild type and Li-TSC1^{KO}RagA^{GTP} mice (Supplementary Fig. 6j). Thus, to capture early transcriptional differences in EC from wild type and Li-TSC1^{KO}RagA^{GTP} mice, we conducted RNA sequencing analysis from purified p14 EC (Fig. 6F). Samples clustered by genotype on PCA (Fig. 6G), and morphogenic pathways (Wnt, Hedgehog) appeared downregulated (Fig. 6H), while metabolic pathways (fatty acids, peroxisome, glycolysis) and signaling (mTORC1, Myc, KRas) were enriched in EC from Li-TSC1^{KO}RagA^{GTP} mice. A partial loss of EC polarization temporarily coincides with that of hepatocytes, with the increase in mTORC1 activity in Li-TSC1^{KO}RagA^{GTP} livers (Supplementary Fig. 6k), and with increase in oxidative stress, albeit minimal compared to that observed in adult Li-TSC1^{KO}RagA^{GTP} livers (Supplementary Fig. 6l–o). Consistently, administration of rapamycin from p1 to p14 partially restored zonation and Wnt targets of mutant livers (Supplementary Fig. 7a–d).

Genetic manipulation based on *Albumin-Cre* expression in Li-TSC1^{KO}RagA^{GTP} mice is restricted to hepatocytes, and we discarded spurious Cre activity in liver EC by means of the mT/mG lineage tracing approach, by assessing mTORC1 activation in EC in Li-TSC1^{KO}RagA^{GTP} by IF, and by quantifying the expression of *Tsc1* in primary hepatocytes and primary EC from wild-type and Li-TSC1^{KO}RagA^{GTP} mice (Supplementary Fig. 7e–h). Thus, changes in EC must occur secondarily to an alteration in hepatocytes by deregulated mTORC1 activity. We reasoned that a hepatocyte-intrinsic process in triggering the establishment of metabolic zonation should take place within the very early weeks, coincidentally or preceding the changes in EC. The expression of Frizzled (Fzd) receptors is essential to transduce the signal from Wnt ligands, and Fzd function has been linked to mTORC1 activity in the liver, as genetic deletion of *Tsc1* suppresses Wnt signaling pathway through the down-regulation of Fzd proteins in a Dvl-dependent manner[40,41]. Among the Fzd receptors, *Fzd1*, *Fzd7* and *Fzd8* are expressed at high levels in the mouse liver according to the *Tabula muris* dataset[42]. Thus, we quantified the levels of *Fzd1*, *7* and *8* by RT-qPCR in the early postnatal period of establishment of hepatic zonation. Strikingly, the levels of *Fzd1* and *Fzd8*, but not *Fzd7*, are largely upregulated in wild type livers between p5 and p10, and subsequently decrease thereafter (Fig. 6I–K). Interestingly, this transient, sharp raise during the early phase of establishment of zonation does not occur in livers from Li-TSC1^{KO}RagA^{GTP} mice. The downregulation of *Fzd*

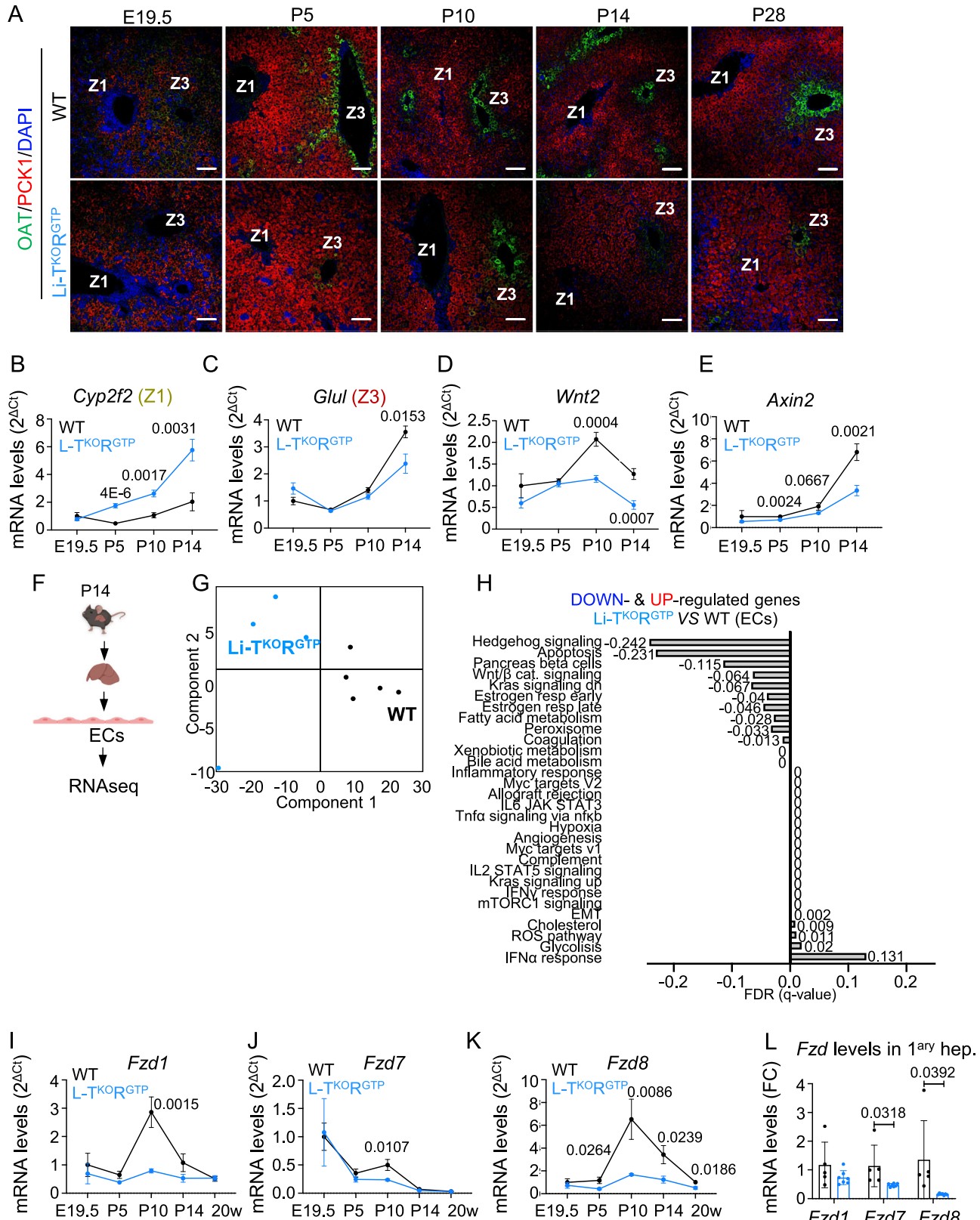

receptors in Li-TSC1$^{KO}$RagA$^{GTP}$ samples was confirmed in our RNA sequencing data (Supplementary Data 1), and in purified wild-type and TSC1$^{KO}$RagA$^{GTP}$ hepatocytes (Fig. 6L). High levels of *Fzd8* mRNA levels in the first two weeks after birth were evident by RNAscope hybridization in the whole parenchyma of wild-type livers, compared to Li-TSC1$^{KO}$RagA$^{GTP}$ samples, including hepatocytes located in Zone 3 (Supplementary Fig. 7i, j). These results strongly support the existence

of an early wave of suppressed Fzd-Wnt/β-catenin signaling in hepatocytes autonomously driven by increased mTORC1 activity that precludes the establishment of the central metabolic program. This suppressed signal transduction may then be reinforced in a paracrine, EC-dependent manner via the downregulation of expression and secretion of Wnt2 and Rspo3. Collectively, these results support that constitutive nutrient and hormone signaling to mTORC1 in

**Fig. 6 | Hepatic zonation in Li-TSC1$^{KO}$RagA$^{GTP}$ livers is not segregated during postnatal maturation. A** Representative pictures of immunofluorescence against Ornithine aminotransferase (Zone 3) together with Phosphoenolpyruvate Carboxykinase 1 (Zone 1) in the liver of wild-type and Li-TSC1$^{KO}$ RagA$^{GTP}$ male and female mice at E19.5, P5, P10, P14 and P28. Zone 1 and Zone 3 are highlighted in each image. Scale bar 100 μm. **B–E** RT-qPCR of livers from E19.5 wild-type ($n = 6$) and Li-TSC1$^{KO}$RagA$^{GTP}$ ($n = 5$) male and female mice, p5 wild-type ($n = 7$) and Li-TSC1$^{KO}$RagA$^{GTP}$($n = 7$) male and female mice, p10 wild-type ($n = 4$) and Li-TSC1$^{KO}$RagA$^{GTP}$($n = 6$) male and female mice, and p14 wild-type ($n = 7$) and Li-TSC1$^{KO}$RagA$^{GTP}$($n = 7$) male and female mice relative to levels in E19.5 wild-type animals. β-actin was used as housekeeping gene. Data are shown as mean with SEM. Statistical significance was calculated by using multiple unpaired two-tailed *t*-test. **F** Primary EC were isolated from the liver of wild-type and Li-TSC1$^{KO}$RagA$^{GTP}$ male and female mice at p14 postnatal stage. Created with BioRender.com. **G** PCA of the transcriptomic profiles of the samples. Each dot represents individual biological replicates. **H** Representation of the false discovery rates (FDR) from the top Hallmark gene sets enriched and depleted in EC from Li-TSC1$^{KO}$RagA$^{GTP}$ ($n = 4$) *versus* wild-type ($n = 5$) livers. **I–K** RT-qPCR of livers from E19.5 wild-type ($n = 6$) and Li-TSC1$^{KO}$RagA$^{GTP}$ ($n = 5$) male and female mice, p5 wild-type ($n = 7$) and Li-TSC1$^{KO}$RagA$^{GTP}$($n = 7$) male and female mice, p10 wild-type ($n = 4$) and Li-TSC1$^{KO}$RagA$^{GTP}$($n = 6$) male and female mice, and p14 wild-type ($n = 7$) and Li-TSC1$^{KO}$RagA$^{GTP}$($n = 7$) male and female mice relative to levels in E19.5 wild-type animals. β-actin was used as housekeeping gene. Data are shown as mean with SEM. Statistical significance was calculated by using multiple unpaired two-tailed *t*-test. **L** RT-qPCR of primary hepatocytes from 7-week-old wild-type ($n = 5$) and Li-TSC1$^{KO}$RagA$^{GTP}$ ($n = 7$) male and female mice. Expression levels of the indicated genes relative to the average level in wild-type mice. β-actin was used as housekeeping gene. Statistical significance was calculated by using multiple unpaired two-tailed *t*-test. In all panels data are presented as mean values ± standard deviation.

hepatocytes abrogate the postnatal establishment of the spatial segregation of liver functions via both intrinsic and paracrine signals after birth, impairing the metabolic specification and maturation of hepatocytes.

## Disruption of metabolic zonation under total parenteral nutrition in neonatal pigs

The establishment of molecularly distinct segregated functions of hepatocytes occurs only after the interruption of steady transplacental supply of nutrients at birth. Fluctuations imposed by intermittent feeding via the enteral-portal influx of nutrients after birth may operate as a trigger for the portal *versus* central metabolic specification of hepatocytes. Our data is consistent with constitutive nutrient and hormone signaling to mTORC1 after birth impairing such segregation of portal and central metabolic functions. In other words, the absence of zonated gene expression in Li-TSC1$^{KO}$RagA$^{GTP}$ mice supports that postnatal fluctuations in mTORC1 signaling in hepatocytes may unleash the execution of a transcriptional program for the spatial segregation of hepatocyte functions. In seeking for an independent support to this model, we decided to interrogate the licensing of early postnatal zonation in a system in which the intermittency of enteral-portal influx of nutrients was affected by non-genetic means. Thus, we monitored the establishment of hepatic zonation in neonatal pigs subjected to a total parenteral nutrition (TPN) regime[43]. In this experimental system, 2-week-old piglets, age at which zonation is only partially established (Supplementary Fig. 8a), are either orally fed with milk, or the jugular vein is catheterized and all nutrients are supplied parenterally and under a constant rate for two additional weeks[43] (Fig. 7A). Hepatocytes of TPN-fed piglets neither receive nutrients from the enteral-portal route of circulation, nor do the levels of nutrients and hormones fluctuate because of the absence of fasting-feeding cycles. Immunohistochemical and Immunofluorescent stainings of GS (Z3) in livers from piglets on TPN exhibited a more diffused pattern, as compared to orally-fed piglets (Fig. 7B, C). Moreover, RT-qPCR of zonated genes revealed altered mRNA expression of central and portal markers (Fig. 7D) and the Wnt/β-catenin target gene *Lgr5* was decreased in TPN-fed piglets (Fig. 7E), partially mirroring the results from Li-TSC1$^{KO}$RagA$^{GTP}$ mice. We conducted bulk RNA sequencing analysis from livers from orally- and TPN-fed piglets (Supplementary Data 1), which clustered by PCA as a function of feeding regime (Fig. 7F). mTOR signaling was upregulated under TPN (Supplementary Fig. 8b, c). Strikingly, the transcriptome of livers from TPN-fed piglets was enriched in mRNAs significantly upregulated in Li-TSC1$^{KO}$RagA$^{GTP}$ livers and significantly depleted from those significantly downregulated in Li-TSC1$^{KO}$RagA$^{GTP}$ livers, and zonation-related genes and portal and central gene signatures were also significantly affected (Fig. 7G, H and Supplementary Fig. 8d–f). Overall, this overlap argues that genetically abrogating nutrient and hormone signaling fluctuations in mouse liver has a degree of molecular similarity to the suppression of the intermittent enteral-portal supply in neonatal piglets under TPN. More importantly, transcriptomic analysis of neonatal pig livers confirmed a global deregulation of zonated genes and down-regulation of Wnt/β-catenin targets when subjected to TPN (Fig. 7I). Finally, the mRNA levels of *Wnt2*, the Wnt ligand downregulated in mouse Li-TSC1$^{KO}$RagA$^{GTP}$ livers were also significantly down-regulated in TPN-fed piglets (Fig. 7J), and the same occurred with *Rspo3* (Supplementary Fig. 8g). Altogether, the analysis of hepatic zonation in piglets subjected to TPN shows that early postnatal abrogation of the intermittent supply of nutrients through the portal circulation correlates with impaired specification of the molecular zonation of hepatocytes.

Collectively, based on genetic activation of nutrient and hormone signaling to mTORC1 in mice, and on the TPN-feeding regime in piglets, we propose the existence of a postnatal trigger of metabolic specification of zonation downstream of mTORC1 in hepatocytes. This program is unleashed by the sensing of fluctuations of enteral influx of nutrients via enteral-portal circulation, and in paracrine interaction with Wnt ligand secretion from EC (Fig. 7K).

## Discussion

We have generated mice with constitutive hormone and nutrient signaling to mTORC1, and observed an exacerbation of the pathologies and phenotypes seen in Li-TSC1$^{KO}$ livers (Fig. 1)[11–18]. Activation of RagA per se in hepatocytes does not compromise longevity, neither does it induce overt hepatic damage, inflammation, or carcinomas[17], but it has a synergic effect together with deletion of *Tsc1*. This synergism, rescued by rapamycin (Fig. 2), occurs with evidence of qualitative differences in the extent of activation of mTORC1 in different pools of hepatocytes. Central Z3 hepatocytes resulted sensitive to activation of either hormone signaling, nutrient signaling, and both inputs together (Fig. 3). This exquisite sensitivity of Z3 hepatocytes to either nutrient or GF signaling was also evident in wild-type liver in fasting-refeeding cycles, and as analyzed by both a biochemical readout of mTORC1 activity (phospho-S6) and a cellular consequence of mTORC1 activation (increase in cell size). This result may be surprising because Z1, and not Z3 hepatocytes, are exposed to fluctuating nutrient levels from the afferent enteral-portal circulation. Because oxygen, nutrients and hormones are relatively scarce in the central zone, hepatocytes located in this region may be more primed to respond to subtle changes in growth factors and nutrient levels. Nevertheless, it is conceivable that hepatocytes from Z1 and Z3 may be differentially primed to respond to specific nutritional and hormonal inputs, depending on both the fluctuations and the differential expression of sensors or transducers, as reported[21].

Whether the nutrient and GF signaling cues are independent cascades that orthogonally control mTORC1, or alternatively have functional interactions upstream of mTORC1 is not clear. Extensive research sustains that full mTORC1 activation only occurs if both

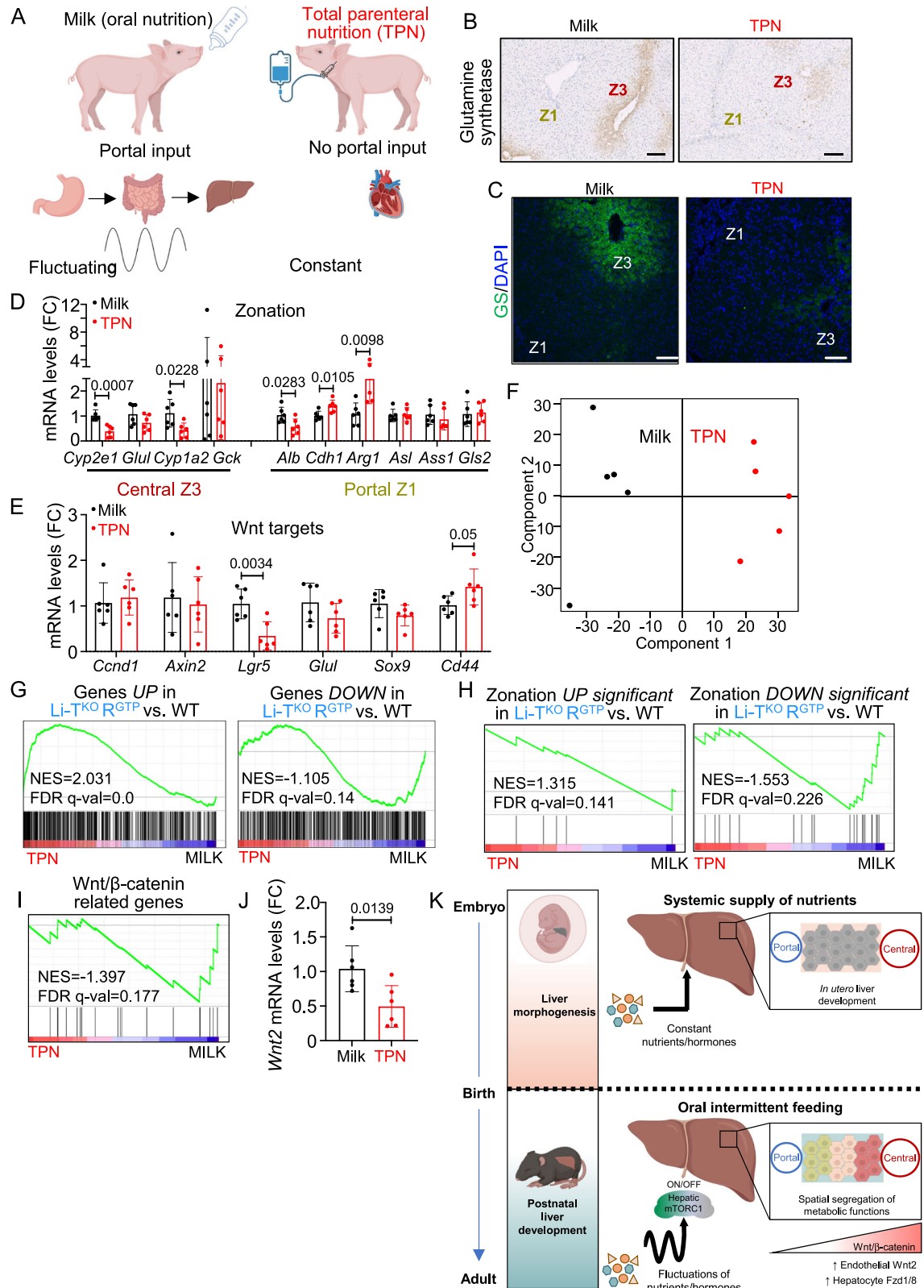

intracellular nutrient sufficiency and extracellular hormone signals recruit and activate mTORC1, respectively, thus operating as a coincidence detection mechanism[4]. However, biochemical data in cultured cells has also generated support for direct cross-talk between nutrient and hormone signaling[44,45]. Moreover, expression and localization of nutrients receptors can be indirectly modulated

by GF signaling[46], and decreased GF signaling lowers the expression of amino acid transporters[47]. Finally, mouse work unequivocally shows a relevant functional interaction between Tsc1/2 and RagC, but not RagA[48]. Unfortunately, manipulating the levels of nutrients and GF independently is technically impossible in mice. Our signal transduction experiments in primary hepatocytes (Fig. 3) show that

**Fig. 7 | Total parenteral nutrition impairs hepatic metabolic zonation in neonatal pigs. A** Schematic representation of key aspects of orally-fed (MILK) and TPN-fed (TPN) piglets. Created with BioRender.com. **B** Representative pictures of immunohistochemistry against Glutamine synthetase in the liver of 2-week-old orally-fed and TPN-fed piglets. Zone 1 and Zone 3 are highlighted in each image. Scale bar 100 μm. **C** Representative pictures of immunofluorescence against Glutamine synthetase in the liver of 2-week-old orally-fed and TPN-fed piglets. Zone 1 and Zone 3 are highlighted in each image. Scale bar 100 μm. **D**, **E** RT-qPCR of livers from 2-week-old orally-fed ($n = 6$) and TPN-fed ($n = 6$) piglets. Expression levels of the indicated genes relative to the average level in control milk piglets. Atp5f1 was used as housekeeping gene. Statistical significance was calculated by using multiple unpaired two-tailed *t*-test. **F** PCA of the transcriptomic profiles of the samples. Each dot represents individual biological replicates. **G** GSEA in TPN-fed compared to orally-fed piglets on the list of significantly upregulated genes or downregulated genes in Li-TSC1[KO]RagA[GTP] liver samples. NES normalized enrichment score, FDR false discovery rate. **H** GSEA in TPN-fed compared to orally-fed piglets on the list of significantly upregulated or downregulated genes related with zonation in Li-TSC1[KO]RagA[GTP] liver samples. NES normalized enrichment score; FDR false discovery rate. **I** Enrichment plot for the "Wnt/β-catenin related genes" signature for TPN-fed *versus* control livers from piglets at the mRNA level. NES normalized enrichment score, FDR false discovery rate. **J** RT-qPCR of livers from 2-week-old orally-fed ($n = 6$) and TPN-fed ($n = 6$) piglets. Expression levels of Wnt2 relative to the average level in control milk piglets. Atp5f1 was used as housekeeping gene. Statistical significance was calculated by using unpaired two-tailed *t*-test. **K** Nutrient and hormone signaling to mTORC1 as a metabolic trigger of hepatic zonation. Created with BioRender.com. In all panels data are presented as mean values ± standard deviation.

phosphorylation of S6K1 in cells endogenously expressing constitutive GTP-locked RagA is almost entirely insensitive to changes in amino acid levels, but sensitive to GF deprivation-stimulation. Reciprocally, mTORC1 in *Tsc1*[KO] hepatocytes is unresponsive to GF deprivation-stimulation but responds to starvation and replenishment of amino acids. Consistently, TSC1[KO]RagA[GTP] hepatocytes are totally insensitive to either perturbation. These data, generated in primary cells devoid of oncogenic mutations and without ectopic overexpressing active or inactive variants in the signaling cascades, favor the notion that under the maximal availability of nutrients and GF in culture conditions, both inputs signal to mTORC1 in a largely independent manner.

The transcriptomic and proteomic analyses revealed overlapping molecular changes among single- and double-mutant livers, and also changes unique of the Li-TSC1[KO]RagA[GTP] livers (Fig. 4 and Supplementary Fig. 4). Among these, the levels of zonated genes were modestly affected in RagA[GTP] and in TSC1[KO] livers as compared to wild-type livers (Supplementary Fig. 4), but largely altered in TSC1[KO]RagA[GTP] livers, and in particular, centrally-expressed genes (Fig. 4). These results go in agreement with a previous study showing that increased mTORC1 activation (by genetic deletion of *Tsc1* in mice) decreases the expression of Wnt target genes in the murine liver through a Dvl-Fzd axis[40]. Previous elegant mouse work has shown that the establishment and maintenance of zonated metabolism in not driven exclusively by hepatocytes, but rather the consequence of a cross-communication between endothelial cells that paracrinally induce the Wnt/β-catenin program in central Z3 hepatocytes by endothelial-cell secretion of Wnt ligands[49]. The hepatic zonation defect observed in Li-TSC1[KO]RagA[GTP] mice occurred with a large decrease in Wnt/β-catenin transcriptional program in hepatocytes and with decreased synthesis of *Wnt2* ligand from endothelial cells (Fig. 5). While exogenous supply of Wnt ligands and RSPO in culture (Supplementary Fig. 5) can transactivate Wnt/β-catenin signaling in hepatocytes, we observed that an early increase in Fzd receptor during the establishment of metabolic zonation after birth is strikingly impaired when mTORC1 activity is constitutive. These results strongly argue that activation of nutrient and hormone signaling to mTORC1 in hepatocytes impairs the establishment of the central program autonomously, and secondarily impairs a hepatocyte-to-EC message critical for the reinforcement and maintenance of postnatal metabolic zonation. It also reinforces the idea that hepatocytes are not merely recipients of spatial information from endothelial cells in the form of Wnt ligands, but rather an actor in the metabolic segregation of hepatocyte functions. The Dvl-Fzd turnover axis[40] may certainly be at work in postnatal liver maturation, and may act together with the transient raise in Fzd expression as part of the mechanism behind the loss of Zone 3 hepatocyte identity.

Impaired Wnt/β-catenin activity, deregulation of other morphogenic cues, or either portal or central liver damage, which typically results in compensatory expansion of the non-damaged zone[50–53], can result in loss of hepatic zonation. Thus, the defect in metabolic zonation in Li-TSC1[KO]RagA[GTP] livers might be secondary to liver damage, which is also present in these mice (Fig. 1), and as reported for mice with deregulated nutrient and hormone signaling[18]. However, we believe this is an unlikely explanation as inferred by several results. Damage in Li-TSC1[KO]RagA[GTP] mice is not restricted to portal or central zones (Supplementary Fig. 4), so it is unclear whether compensatory expansion of a non-damaged zone (in this case, portal Z1) may occur. Moreover, hepatocyte damage and compensatory proliferation should result in increased, rather than decreased, Wnt/β-catenin transcriptional program[49]. In addition, while oxidative damage is seen, it only mildly increases during the early days after birth, as compared by the massive increase that occurs in adult individuals, as reported[18]. Lastly, a strikingly similar decrease in central zonation and decrease in Wnt2 ligand is observed in TPN-fed piglets, with no detectable liver damage (Fig. 7).

At birth, mammalian liver has already entirely formed its lobes, the portal vein and hepatic artery circulation, cholangiocytes with the bile ducts, fenestrated endothelial cells, and is almost entirely mature[54,55]. In contrast to this normal macroscopic appearance, at the molecular level the metabolic differences of centrally and portally irrigated hepatocytes are yet to be established in a second, postnatal maturation phase. Spatially segregated functions of hepatocytes respond to the portal-central gradient of nutrients and oxygen, with portal hepatocytes executing energy-consuming functions of the liver. The lack of segregated functions at birth is coherent with the absence of such gradient of nutrients and oxygen due to the continuous transplacental irrigation *in utero*. Only after birth are central Z3 and portal Z1 hepatocytes exposed to fluctuations and different concentrations of nutrients, oxygen and energy, and subjected to the intermittency that occurs with the feeding cycles. Thus, only after birth is the metabolic specification of hepatocytes meaningful for an effective maintenance of metabolic homeostasis in response to gradients and intermittency of intake. In the light of our work, the detection of fluctuations of nutrients and hormones through mTORC1 signaling after birth may operate as a trigger for execution of a segregation program of the distinct molecular functions optimal for the gradients in energy, nutrients and oxygen between portal Z1 and central Z3 (Fig. 7).

In brief, our work proposes a postnatal maturation cue instructed by the mTORC1 pathway and operating as a trigger for the execution of latent metabolic programs of hepatocytes that adjust functions to portal-to-central gradient of metabolites as defined by blood supply.

## Methods
### Ethical approvals
All mice procedures carried out at the CNIO were performed according to protocols approved by the CNIO-ISCIII Ethics Committee for Research and Animal Welfare (CEIyBA) and the Autonomous

Community of Madrid (CAM). Protocol numbers PROEX285/15, PROEX15/18 and PROEX225.7/22. The pig study was approved by and conducted in accordance with the Institutional Animal Care and Use Committee (IACUC) of SLU (SLU No. 2657, US Department of Agriculture registration 43-R 011).

## Generation of Li-TSC1^KO^RagA^GTP^ mice

For hepatocyte-specific activation of RagA (Li-RagA$^{GTP}$), $RragA^{GTP/flox}$ mice[17] were bred with mice carrying Albumin-Cre (Alb-Cre) recombinase[56] (JAX stock #003574). For hepatocyte-specific deletion of $Tsc1$ (Li-TSC1$^{KO}$), $Tsc1^{flox/flox}$ mice[57] (JAX stock #005680) were bred with mice carrying Albumin-Cre (Alb-Cre) recombinase[56] (JAX stock #003574). For the generation of mice with constitutive nutrient and growth factor signaling to mTORC1 (Li-TSC1$^{KO}$RagA$^{GTP}$) we crossed Albumin-Cre $RragA^{GTP/flox}$ mice with $Tsc1^{flox/flox}$ mice. For the validation of absence of Alb-Cre mediated recombination in liver endothelial cells, we bred Albumin-Cre mice with mice carrying the Gt(ROSA) 26Sor$^{tm4(ACTB-tdTomato,-EGFP)Luo}$ (mTmG) allele[58] (JAX stock #007676).

## Animal procedures

Mice were housed under specific pathogen free conditions at 22 °C and with 12-h dark/light cycles (light cycle from 8:00 to 20:00). Mice were fed with a standard chow diet (Harlan Teklad #2018S/2018SC). For 24 h fasting experiments, mice were placed in a clean cage without access to food from 8 am to 8 am next day. In each cage a maximum of two animals were placed. For survival experiments, mice were observed weekly by trained personnel until they presented signs of morbidity, time at which mice started to be inspected daily until application of humane end-point criteria (https://grants.nih.gov/grants/olaw/guide-for-the-care-and-use-of-laboratory-animals.pdf) in consultation with veterinary staff blinded to the mouse genotype or procedure. Sex was considered in the study design and analysis of metabolic experiments. Neonates were obtained by caesarean section at E19.5[59] and placed on ice for sacrificing. To prevent early delivery, pregnant mums were injected with 2 mg of progesterone (Sigma P0130) in PBS once a day at E17.5 and 18.5. Rapamycin was administered since weaning in food, encapsulated in chow diet at 42 ppm (Rapamycin Holdings and Purina Lab Diet). For acute rapamycin treatment, IP administration of 2 mg/kg rapamycin was performed for three days.

## Hemograms and quantification of metabolites from blood

Blood was collected from the sub-mandibular vein of alive mice in EDTA tubes to measure alkaline phosphatase (ALP), alanine aminotransferase (ALT), bile acids, bilirubin and cholesterol using VetScan mammalian liver profile rotors (Abaxis #500-0040-12), and for white and red blood count using a blood cell counter (CVM LaserCell). During euthanasia, blood was sampled with a syringe from the heart.

## Glucose tolerance test (GTT)

Mice were fasted from 4 pm to 8 am. Following fasting, D-Glucose 2 g/Kg (Sigma #G7528) in PBS was injected intraperitoneally and tail blood glucose levels were measured at 15, 30, 45, 60, 90 and 120 min after injection. Glucose was monitored with Accu-check Aviva glucometer.

## Insulin Tolerance test (ITT)

Mice were fasted from 9 am to 3 pm. Following fasting, mice were injected intraperitoneally with 0.375 U/Kg of insulin (Humulin R U-100 Lilly) in PBS and tail blood glucose levels were measured at 15, 30, 45, 60, 90, and 120 min after injection. Glucose was monitored with Accu-check Aviva glucometer. In case of the appearance of seizures due to extremely low levels of glucose, mice were injected with 30% glucose (Sigma #G7528) in PBS.

## Immunohistochemical staining

For hematoxylin and eosin (H&E) and Sirius red staining as well as immunohistochemistry staining against CD45, Glutamine Synthetase and E-cadherin tissue samples were fixed in 10% neutral buffered formalin (4% formaldehyde in solution), and samples were then embedded in paraffin and cut at 3 μm, mounted in super frost plus slides and dried overnight. Slides were deparaffinized in xylene and rehydrated through a series of graded concentrations of ethanol in water, then paraffin sections were stained with H&E or Sirius red for fibrosis, or immunohistochemistry was performed against CD45 (CST #70257) dilution: 1/200, Glutamine synthetase (SIGMA #G2781) dilution: 1/500 or E-cadherin (BD Biosciences #610182) dilution: 1/1000 in an automated immunostaining platform (Ventana Discovery XT). Antigen retrieval was first performed with the appropriate pH buffer (CC1m, Ventana, Roche) and endogenous peroxidase was blocked (peroxide hydrogen at 3%). Subsequently, slides were incubated with the appropriate primary antibody and then with the visualization systems (Omni Map anti-Rabbit, Ventana, Roche) conjugated with horseradish peroxidase. Immunohistochemical reaction was developed using 3,30-diaminobenzidine tetrahydrochloride (DAB) (Chromo Map DAB, Ventana, Roche; DAB Dako) and nuclei were counterstained with Carazzi's hematoxylin. Finally, slides were dehydrated, cleared and mounted with mounting medium for evaluation at the microscope. Positive control sections were included for each staining run.

For immunofluorescence staining against Phospho-S240/244-S6, Glutamine Synthetase, E-cadherin, Ornithine aminotransferase, Phosphoenolpyruvate Carboxykinase 1 or Wheat Germ Agglutinin, sections were fixed 10 min with 4% PFA (Electron Microscopy Sciences #15710) in PBS at room temperature and then wash three times 5 min with PBS. Sections were blocked/permeabilized 1 h at room temperature in PBS with 10% Donkey Serum (Jackson ImmunoResearch #017-000-121), 10% FBS (Hyclone #SV30160.03) and 0.5% Triton (Sigma #T9284). Primary antibodies were diluted in blocking/permeabilization buffer and incubated overnight at 4 °C. This step was followed by three washes in PBS of 5 min each and incubation for 2 h with conjugated secondary antibodies (1:300) and DAPI (Sigma #D9542) at room temperature. After three washes of 5 min in PBS, cells were mounted with Prolong Gold antifade reagent (CST #9071). The following primary antibodies were used: Phospho-S240/244-S6 (CST #5364) dilution:1/100, Glutamine Synthetase (Sigma #G2781) dilution: 1/200, E-cadherin (BD Biosciences #610181) dilution: 1/100, Ornithine aminotransferase (GeneTex #GTX50004) dilution: 1/400, Phosphoenolpyruvate Carboxykinase 1 (Ptglab #16754-1-AP) dilution: 1/200 and Wheat Germ Agglutinin, Alexa Fluor 488 conjugate (Thermo Fisher #W11261) dilution: 1/500. The following secondary antibodies were used: Alexa fluor 555 goat anti-rabbit (Life Technologies #A-21428) dilution: 1/300, Alexa fluor 488 chicken anti-mouse (Life Technologies # A-21200) dilution: 1/300.

For immunostaining of organ sections, samples were fixed for 16 h in a solution of PFA 4% in PBS at 4 °C. After washing the tissue in PBS twice, organs were incubated for 2 h in 10% sucrose (Sigma #9378) in PBS. Then, organs were incubated for 2 h in 20% sucrose (Sigma #9378) in PBS. After that, organs were incubated for 2 h in 30% sucrose (Sigma #9378) in PBS. Then, organs were embedded in OCT (Sakura Tissue-Tek #4586) and frozen at −80 °C. Cryosections of organs (3 μm) were cut on a cryostat (Leica). Immunostaining was performed as described above. To detect membrane-TdTomato or membrane-EGFP, sections were blocked/permeabilized as described above, incubated with DAPI and mounted.

## Assessment and quantification of histological liver damage

Liver necrosis was diagnosed by the observation of single-cell necrosis or small groups of hepatocytes presenting an increased and pale cytoplasm, with a small or fragmented hyperchromatic nucleus (pyknosis), sometimes surrounded by inflammatory cells.

Quantification of necrotic area was performed in Zen 2.3 Blue edition Software (Zeiss). A region of interest (ROI) was drawn in every necrotic focus. All necrotic foci were summed and then relativized to the total hepatic area to calculate necrotic area. Liver fibrosis was determined by the observation of excess fibrous connective tissue between contiguous periportal spaces in the hematoxylin and eosin (H&E) staining. Fibrosis was corroborated, when needed, by the identification of collagen fibers in red color in the Sirius red staining specifically distributed surrounding the hepatocytes located in the periportal space or in the middle of the hepatic parenchyma. Quantification of liver fibrosis was performed in the Sirius red staining with ImageJ. Fibrotic areas were detected as ROI and represented relative to the total area of the tissue. The diagnose of hepatic inflammation was performed by the observation of mononucleated or polymorphonucleated inflammatory cells including, lymphocytes, plasmocytes, macrophages, neutrophils, or eosinophils in multi-focal areas of the liver. In these cases, it could or could not be accompanied by degeneration or necrosis of the hepatocytes, with or without fibrosis. Quantification of inflammation was performed in immunohistochemistry slides against CD45 using Zen 2.3 Blue edition Software (Zeiss). Positive areas of staining were selected as ROIs and compared with total tissue area. For all the quantifications, whole slides were acquired with a slide scanner (AxioScan Z1, Zeiss). Hepatocellular carcinomas were diagnosed based on the observation of a disorganized proliferation of hepatocytes with different degrees of cellular atypia forming nodular lesions in which the normal structure of the liver was lost and portal spaces and central veins were no longer observed. Granulomas were diagnosed based on the identification of chronic inflammation, predominantly mononuclear, with macrophage-like epithelioid cells and occasional giant cells associated with areas of hepatocyte necrosis. Adenomas were diagnosed based on the observation of a nodular area with loss of normal lobular architecture, clearly demarcated from the surrounding liver parenchyma with irregular growth pattern, consisting of hepatocytes with cellular atypia[60]. In some cases, other malignant neoplasms were diagnosed, predominantly lymphomas recognizable by the identification of a diffuse proliferation of rounded and large cells infiltrating the liver, the lung or the kidney.

**Immunofluorescence analysis and quantification**
Raw image files were quantified in ImageJ Software. In each image analyzed, five sections were defined in the radius of a lobule (the distance between the edge of a portal vein and the central vein), and ten ROIs were defined in ten hepatocytes of central zone 3, midlobular zone 2 and portal zone 1, while five ROIs were defined in five hepatocytes in the intermediate regions of each picture. Three pictures were captured per mouse and the livers of minimum three mice were stained per genotype and condition. Mean area of the ROIs of each section were plotted as hepatocyte area in pixels. For the quantification of P-S6 intensity, the total signal (integrated density) in each ROI was measured as signal per pixel area. For the quantification of GS intensity, mean intensity of each ROI was measured as signal per pixel area.

**Immunoblotting**
For cell experiments, cells were rinsed twice with ice-cold PBS and lysed in ice-cold lysis buffer (50 mM HEPES (pH 7.4), 40 mM NaCl, 2 mM EDTA, 1.5 mM sodium orthovanadate, 50 mM NaF, 10 mM pyrophosphate, 10 mM glycerophosphate and 1% Triton X-100 and one tablet of EDTA-free complete protease inhibitors (Roche) per 25 ml). Cell lysates were cleared by centrifugation at maximum speed for 10 min. For tissue extraction, a small piece of previously snap-frozen tissue was homogenized in ice-cold lysis buffer using the suggested homogenization protocols of the FastPrep machine (FastPrep-24™ 5 G). Tissue lysates were cleared by centrifugation at maximum speed

for 10 min. Protein content was measured with BCA (Bicinchoninic Acid) Protein Assay (Thermo Scientific™ #23222), protein was denatured by the addition of sample buffer, boiled for 5 min, resolved by SDS−PAGE and analyzed by immunoblotting. Western blot analyses were performed according to standard procedures. Antibodies were visualized by using Odyssey Infrared Imaging System (Application software version 3.0.30) LI-COR Biosciences. Antibodies from Cell Signaling Technology were used for detection of P-T389-S6K1 (#9234) dilution: 1/500, S6K1 (#2708) dilution: 1/500, P-S240/244-S6 (#5364) dilution 1/1000, P-S235/236-S6 (#2211) dilution: 1/1000, S6 (#2217) dilution: 1/1000, P-T37/46-4EBP1 (#2855) dilution: 1/500, 4EBP1 (#9644) dilution: 1/1000, RagA (#4357) dilution: 1/500, TSC1 (#6935) dilution 1/1000. Anti-vinculin (#V9131) dilution: 1/5000, was obtained from Sigma.

**Sample preparation for proteomic analysis**
Procedure of bulk liver proteomics was performed as described in ref. 17, in which we reported the results obtained in wild-type, Li-RagA^GTP and Li-TSC1^KO mice. In this study, we report the changes observed in Li-TSC1^KORagA^GTP mice. Briefly, liver samples were lysed and then sonicated to shear DNA. Protein concentration was determined using micro-BCA, and 50 µg of each sample were digested by means of the Protifi™ S-Trap™ Mini Spin Column Digestion Protocol. In this protocol, proteins were reduced and alkylated, SDS was removed, proteins were then digested with trypsin and elution from S-Trap columns was performed followed by speed-vac dry. Finally, the resulting peptides were re-dissolved in HEPES. Subsequently, 50 µg per sample were labeled using Thermo Scientific TMT11plex™ Isobaric Label Reagent Set, samples were mixed in 1:1 ratio and total peptide amount was determined from an aliquot by comparing overall signal intensities on a regular LC−MS/MS run. The final mixture was desalted using a Sep-Pak C18 cartridge and dried. Pre-fractioning of peptides was performed trough high pH reverse phase chromatography using an Ultimate 3000 HPLC system equipped with a sample collector. Based on the UV absorbance at 280 nm, fractions were selected for LC-MS/MS analysis. LC-MS/MS was done by coupling an UltiMate 3000 RSLCnano LC system to a Q Exactive HF mass spectrometer. 5 µl of peptides were loaded into a trap column and then transferred to an EASY-Spray PepMap RSLC C18 column and separated using a buffer gradient. The mass spectrometer was operated in a data-dependent mode, with an automatic switch between MS and MS/MS scans using a top 12 method.

**Proteomic data analysis**
Raw files were processed with MaxQuant (v 1.6.10.43) using the standard settings against a mouse protein database (UniProtKB/TrEMBL, 53,449 sequences) supplemented with contaminants. Carbamidomethylation of cysteines was set as a fixed modification whereas oxidation of methionines, protein N-term acetylation, and N/Q deamidation as variable modifications. Minimal peptide length was set to 7 amino acids and a maximum of two tryptic missed/cleavages were allowed. Results were filtered at 0.01 FDR (peptide and protein level). Afterward, the "proteinGroups.txt" file was loaded in Prostar (v1.18)[61] using the intensity values for further statistical analysis. Differential analysis was performed using the empirical Bayes statistics Limma 3.46.0. Proteins with a $p < 0.05$ and a $\log_2$ ratio >0.3 or < −0.3 were defined as regulated. The FDR was estimated to be below 5% by Benjamini−Hochberg. GSEA Pre-ranked (Broad institute) was used to perform gene-set enrichment analysis of the described gene signatures on a pre-ranked gene list, setting 1000 gene-set permutations. GSEA Pre-ranked calculates an enrichment score for each gene set using the Kolmogorov−Smirnov test. The nominal P-value estimates the statistical significance of the enrichment score for a single gene set and it was corrected for gene-set size and multiple hypothesis testing. Expression patterns were displayed in heatmaps using pheatmap

package (RRID:SCR_016418) in R software, with expression data, normalized by $z$-score. Hierarchical clustering was performed using euclidean correlation distance. The mass spectrometry proteomics data have been deposited to the ProteomeXchange Consortium with the identifier PXD041439.

## Mouse bulk liver and endothelial cells transcriptomics

Total RNA samples (250 ng) were processed with the QuantSeq 3' mRNA-Seq Library Prep Kit (FWD) for Illumina (Lexogen, Cat.No. 015) by following manufacturer instructions. RNA quality scores were 7.2 on average (range 5.9–7.8) when assayed on a PerkinElmer LabChip analyzer. Library generation is initiated by reverse transcription with oligodT priming, and a second strand synthesis is performed from random primers. Primers from both steps contain Illumina-compatible sequences. Libraries are completed by PCR: this kit generates directional libraries stranded in the sense orientation, the read1 (the only read in single read format) has the sense orientation. cDNA libraries are purified, applied to an Illumina flow cell for cluster generation and sequenced on an Illumina NextSeq 550 (with v2.5 reagent kits) by following manufacturer's protocols. Eightyfive-base-pair single-end sequenced reads followed adapter and polyA tail removal as indicated by Lexogen. The resulting reads were analysed with the nextpresso[62] pipeline as follows: sequencing quality was checked with FastQC v0.11.0 (https://www.bioinformatics.babraham.ac.uk/projects/fastqc/). Reads were aligned to the mouse genome (GRCm38) with TopHat2[63] using Bowtie[64] and Samtools[65], allowing 3 mismatches and 20 multihits. The GENCODE[66] vM20.GRCm38.Ensembl95 gene annotation was used. Read counts were obtained with HTSeq[67]. Differential expression and normalization was performed with DESeq2[68], keeping only those genes where the normalized count value was higher than 2 in at least 10% of the samples. Finally, those genes that had an adjusted $p$-value below 0.05 FDR were selected. GSEA Pre-ranked[23] was used to perform gene set enrichment analysis for the selected gene signatures on a pre-ranked gene list, setting 1000 gene set permutations. Only those gene sets with significant enrichment levels (FDR $q < 0.25$) were considered. Expression patterns were displayed in heatmaps using pheatmap package (RRID:SCR_016418) in R software, with expression data, normalized by $z$-score. Hierarchical clustering was performed using euclidean correlation distance. The transcriptomics data generated in this study have been deposited to the GEO database under the following accession codes: bulk liver from Li-TSC1$^{KO}$RagA$^{GTP}$mice (GSE225265) and endothelial cells isolated from wild-type and Li-TSC1$^{KO}$RagA$^{GTP}$ livers (GSE241848).

## Pig bulk liver transcriptomics

Total RNA samples (500 ng; RQS/RIN average 7.4, range 6.9–7.8) were processed with the QuantSeq 3' mRNA-Seq Library Prep Kit (FWD) for Illumina (Lexogen, Cat.No. 015) by following manufacturer instructions. Library generation is initiated by reverse transcription with oligodT priming, followed by a random-primed second strand synthesis. Primers from both steps contain Illumina-compatible sequences. Libraries are completed by PCR. This kit generates directional libraries stranded in the sense orientation, the read1 (the only read in single read format) has the sense orientation. cDNA libraries are purified, applied to an Illumina flow cell for cluster generation and sequenced on an Illumina NextSeq 550 (with v2.5 reagent kits) by following manufacturer's protocols. Eightyfive-base-pair single-end sequenced reads followed adapter and polyA tail removal as indicated by Lexogen. The resulting reads were analysed with the nextpresso pipeline as follows: sequencing quality was checked with FastQC v0.11.0. Reads were aligned to the pig genome (Sus_scrofa_11.fa from the UCSC Genome Browser) with TopHat2 using Bowtie and Samtools, allowing 3 mismatches and 20 multihits. The REFseq gene annotation for *Sus scrofa* (susScr11.refGene.gtf) was used. Read counts were obtained with HTSeq. Differential expression and normalization was performed

with DESeq2, keeping only those genes where the normalized count value was higher than 10 in at least 25% of the samples. Finally, those genes that had an adjusted $p$-value below 0.05 FDR were selected. GSEAPreranked was used to perform gene set enrichment analysis for the selected gene signatures on a pre-ranked gene list, setting 1000 gene set permutations. Only those gene sets with significant enrichment levels (FDR $q < 0.25$) were considered. The transcriptomics data generated in this study have been deposited to the GEO database under the following accession codes: bulk liver from control and TPN-fed neonatal pigs (GSE225266).

## Mouse liver scRNA-seq

Mouse liver cells purification protocol was adapted from refs. 31,32. Briefly, liver cells were isolated following a two-step protocol of Pronase/Collagenase digestion. Firstly, the liver was perfused with HBSS (Sigma #H6648) containing 0.2 mg/mL EDTA. Secondly, a perfusion with 0.4 mg/mL Pronase (Sigma #P5147) and 2 mg/mL Collagenase Type II (Worthington #LS004196) in HBSS was performed. Finally, the liver was perfused with HBSS containing 0.4 mg/mL Pronase, 2 mg/mL Collagenase Type II and 0.1 mg/mL DNase I (Roche #R104159001). The liver was minced and further digested with HBSS containing 0.4 mg/mL Pronase, 2 mg/mL Collagenase Type II and 0.1 mg/mL Dnase I for 25 min with shaking at 37 °C. To stop digestion, DMEM (Thermo Fisher #31966047) was added. The resulting liver cell suspension was filtered and washed three times through centrifugation at 300 g for 4 min in 2% FBS-PBS. The suspension was then subjected to density gradient centrifugation using 20% Percoll (GE Healthcare #17-0891-01) to remove dead cells. Resuspension of the cells in ACK lysis buffer (Thermo Fisher #A1049201) allowed the lysis of red blood cells. Cell suspension was then centrifuged and resuspended in 2% FBS-PBS to proceed with scRNA-seq analysis using 10X Genomics Chromium Single-Cell 3' according to the manufacturer's instructions.

## scRNA sequencing data analysis

Cell sample was loaded onto a 10x Chromium Single Cell GEM chip (10x Genomics) as described in the manufacturer's protocol (Chromium Next GEM Single Cell 3' GEM, Library & Gel Bead Kit v3.1, PN-1000121). Intended targeted cell recovery: ~10,000 cells. Generation of gel beads in emulsion (GEMs), barcoding, GEM-RT clean-up, cDNA amplification and library construction were all performed as recommended by the manufacturer. scRNA-seq libraries were sequenced with an Illumina NextSeq 550 (using v2.5 reagent kits) in paired-end fashion (28 bp + 56 bp bases). The bollito[69] pipeline was used to perform read analysis, as follows: Sequencing quality was checked with FastQC (http://www.bioinformatics.babraham.ac.uk/projects/fastqc/). Reads were aligned to the mouse reference genome (GRCm38, vM25 gene annotation from GENCODE[66] with STARsolo (STAR 2.7.3a[70]. Seurat 3.2.2[71] was used to check the quality of sequenced cells, explore and quantify single-cell data, obtain cell clusters and specific gene markers. Differential expression data were visualized in volcano plots using the package EnhancedVolcano (RRID:SCR_018931) from Bioconductor in R software. The transcriptomics data generated in this study have been deposited to the GEO database under the following accession codes: scRNAseq from Li-TSC1$^{KO}$RagA$^{GTP}$ and wild-type livers (GSE229830).

## Quantitative PCR

Total RNA was extracted from tissues or cells using Trizol (Invitrogen #15596026) according to manufacturer's instructions. For RNA extraction from tissue, RNeasy Tissue Kit (Qiagen #74106) was used. For RNA extraction from cells, Direct-zol RNA Minipreps Kits (ZYMO RESEARCH #R2050) were used. Samples were retro-transcribed using SuperScript IV VILO Master Mix (Invitrogen #11756500) following manufacturer's instructions. Quantitative real-time PCR was run in triplicates using GoTaq qPCR Master Mix (Promega #A6001) with a

reaction volume of 10 µL in a QuantStudio 6 Flex Real-Time PCR System thermocycler (Applied Biosystems). Data were analyzed by the change-in-threshold ($2^{-\Delta\Delta CT}$) method, using β-actin as "housekeeping" reference gene for mouse samples and Atp5f1 for pig samples. Results are represented as fold changes relative to the mean expression levels of the control condition. Primer sequences are listed in Supplementary Data 2.

### Detection of Wnt2 and Fzd8 in mouse tissue by RNA in situ hybridization

Paraffin-embedded tissue from mouse livers was sectioned and RNA-scope 2.5 VS probes for Mm-Wnt2 (313609, ACDbio) and Mm-Fzd8 (404949, ACDbio) was assayed on the Roche Ventana Discovery XT using the standardized automated protocols at the CNIO Histopathology Core Facility.

### Primary hepatocytes isolation and signaling

Primary hepatocytes were isolated following liver perfusion[72]. For mRNA expression assessment, cells were immediately frozen in Trizol (Invitrogen #15596026) for RNA extraction without plating them. For signaling experiments, primary hepatocytes from WT, Li-RagA$^{GTP}$ and Li-TSC1$^{KO}$ mice were cultured at a confluence of 250.000 cells per well and primary hepatocytes from Li-TSC1$^{KO}$ RagA$^{GTP}$ mice were cultured at a confluence of 500.000 cells per well due to lower viability of cells. In all the cases, cells were plated in collagen-coated (Sigma #C3867) multi six-well plates. After 6 h the media was changed, and hepatocytes were cultured overnight before any experimental procedure. To perform amino acid starvation, the next day, cells were rinsed three times and placed in DMEM:F12 without amino acids (US Biological Life Science #D9807-10) supplemented with 6 mM NaHCO₃ (Sigma #1063291000), 18 mM Hepes pH 7.4 (Lonza #17-737E), 25 mM Glucose (Sigma #G8769) and 10% dialyzed FBS during 1 h. Re-stimulation with amino acids was performed during 10 min when indicated. To perform overnight serum withdrawal, the next day, cells were rinsed in hepatocyte media without FBS and placed in FBS withdrawal medium for 16 h. Then, re-stimulation with 100 nM Insulin was performed during 15 min when indicated. Recombinant Wnt2 (R&D #11117-WN-010) was supplied to primary hepatocytes at 400 ng/µl for 16 h. Recombinant Rspo1 (R&D #3474-RS) was supplied to primary hepatocytes at 400 ng/µl for 16 h.

### Primary liver endothelial cells isolation

Selection of liver-derived endothelial cells was adapted from ref. 73. Briefly, mouse livers were digested with 2.25 mg/mL Dispase II (Roche #04942078001) and 0.8 mg/mL Collagenase A (Roche #10103586001) in calcium and magnesium-free HBSS (Gibco #14170112) for 1 h at 37 °C. Liver digestion was followed by positive selection of EC with magnetic beads (Dynabeads sheep anti-rat IgG, Thermo Fisher #11035) coated with anti-mouse CD144 antibody (BD Biosciences #555289). Once the isolation was finished, cells were immediately frozen in Trizol (Invitrogen #15596026) for RNA extraction to assess mRNA expression.

### Total parenteral nutrition in piglets

Surgery and catheter placement in piglets was performed[43]. Two-week-old piglets (*Sus scrofa/domestica*, strain White Yorkshire x Landrace pigs) were anesthetized using isoflurane and jugular catheters were placed into the jugular vein and carotid artery. During the initial 24 h postoperatively, all pigs received TPN at 50% of full intake providing (in g·kg$^{-1}$·day$^{-1}$) 12.5 glucose, 6.5 L-amino acids, 2.5 lipid, and 412 kJ·kg$^{-1}$·day$^{-1}$ at a volume of 120 ml·kg$^{-1}$·day$^{-1}$. Thereafter, intakes were increased to 100% within 48 h. Pre- and postoperatively on each day, piglets received enrofloxacin (2.5 mg/kg; Bayer, Shawnee Mission, KS). Postoperatively, each piglet received one dose of analgesic (0.1 mg/kg butorphenol tartrate; Fort Dodge Laboratories, Fort Dodge, IA).

### Statistical analysis

The n depicts the total number of animals per group and is indicated in each Figure legend. Statistical analysis was carried out with Prism 9 (GraphPad). Unpaired *t*-test was used for single comparisons, Log-rank (Mantel–Cox) test was used for survival curves, 1-way ANOVA (Dunnett's multiple compassion test) was used when more than one genotype was analyzed. All experiments that include a second variable were analyzed using 2-way ANOVA with Tukey's or Sidák's multiple comparison post-test. Where appropriate, the area under the curve (AUC) was calculated. Chi-squared test was used when comparing contingency data. All error bars in all panels depict Standard deviation (SD).

### Reporting summary

Further information on research design is available in the Nature Portfolio Reporting Summary linked to this article.

## Data availability

The transcriptomics data generated in this study have been deposited to the GEO database under the following accession codes: bulk liver from Li-TSC1$^{KO}$RagA$^{GTP}$mice (GSE225265), bulk liver from control and TPN-fed neonatal pigs (GSE225266), scRNAseq from Li-TSC1$^{KO}$RagA$^{GTP}$ and wild-type livers (GSE229830) and endothelial cells isolated from wild-type and Li-TSC1$^{KO}$RagA$^{GTP}$ livers (GSE241848). The mass spectrometry proteomics data have been deposited to the ProteomeXchange Consortium with the identifier PXD041439. The authors declare that other source data supporting the findings of this study within the article and its Supplementary information files are available upon reasonable request to the corresponding author. Source data are provided with this paper.

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

## Acknowledgements

We thank CNIO Histopathology, Animal Facility and Genomics Core Units for excellent technical support. The RETOS projects Program of Spanish Ministry of Science, Innovation and Universities, Spanish State Research Agency, cofunded by the European Regional Development Fund (grants PID2019-104012RB-I00 and PID2022-136413OB-I00), EU-H2020 Program (ERC-2014-STG-638891), Excellence Network Grant from MICIU/AEI (SAF2016-81975-REDT), Spanish Association Against Cancer Research Scientific Foundation Laboratory Grant, Beca de Investigación en Oncología Olivia Roddom, FERO Grant for Research in Oncology and funding from LaCaixa Banking Foundation (LCF/PR/HR21/0046) (to A.E.). The work in the lab of J.M. is supported by European Proteomics Infrastructure Consortium providing access (EPIC-XS): Project number 823839. A.B.P.G., L.d.P.R., N.D.S. and C.D.C.A. are recipients of Ayudas de contratos predoctorales para la formación de doctores from MICIU/AEI (BES-2017 – 081381, PRE-2019-090891, BES-2016-077410, BES-2015-073776). A.E. is an EMBO Young Investigator.

## Author contributions

ABPG performed most experiments, contributed to experimental design, data analysis, and writing of the manuscript. LdPR provided extensive experimental support, AS, NDS, CDCA, CS, and LLV, provided help with experimentation and analysis. FG and JM performed proteomics analyses, OGC and EPY helped with transcriptomics analyses. EC diagnosed the histology and pathology. JK and AJ contributed piglet experimentation and samples. GS contributed primary hepatocytes cultures and expertise. AE conceived and supervised the study, analyzed the data, wrote the manuscript and secured funding. All authors read and commented on the manuscript and figures.

## Competing interests

The authors declare no competing interests.
