## [Peer Review File · Nature Communications]

Hepatic nutrient and hormone signaling to mTORC1 instructs the postnatal metabolic zonation of the liverEditorial Note: Parts of this Peer Review File have been redacted as indicated to maintain the confidentiality of unpublished data.

REVIEWER COMMENTS

Reviewer #1 (Remarks to the Author):

This submitted study uses mouse models to understand how nutrient and growth factor signaling are integrated to control mTORC1 activity and liver metabolism. Three mutant mouse strains were generated: Li-RagAGTP mice (with constitutive Rag GTPase signaling in hepatocytes), Li-Tsc1KO mice (with constitutive insulin signaling in hepatocytes), and compound Li-RagAGTP; Li-Tsc1KO mice (Li-TKO;RGTP). The major focus of the study relates to the phenotype of Li-TKO;RGTP mice which is more severe than that in the single mutant mice. Li-RagAGTP; Li-Tsc1KO mice show reduced lifespan and present with a severe ensuing liver pathology involving impaired glucose homeostasis, liver dysfunction, inflammation, fibrosis, and hepatocellular carcinoma (HCC). An almost identical liver pathology was recently reported by another group (Cho CS et al., Cell Discov. 2019) in a mouse model in which nutrient and growth factor signaling to mTORC1 were simultaneously activated in hepatocytes. However, one major difference between the 2 studies is the discovery by Plata-Gomez et al. that metabolic zonation is disrupted in the Li-TKO;RGTP liver. The authors show gradual loss of the pericentral (Zone 3) marker GS, and expanded expression of the periportal (Zone 1) marker E-cadherin in Li-TKO;RGTP livers after birth, and this alteration is accompanied by declining expression of WNT2 in hepatic endothelial cells. Published studies have demonstrated that WNT secretion from central vein endothelial cells and hepatic sinusoidal endothelial cells is necessary to establish and maintaining Zone 3 hepatocyte gene expression. Whether loss of WNT2 expression precedes the loss of pericentral hepatocyte gene expression in Li-TKO;RGTP livers is not convincingly addressed in the manuscript.

The proposal that postnatal intermittent activity of mTORC1 signaling is necessary to establish correct metabolic zonation in the mammalian liver is provocative and very novel. However, as presented the data in support looks insufficient and the liver zonation phenotype requires a more thorough analysis using appropriate markers, better imaging and focusing on the most relevant stages. Another major deficiency is that no mechanism is offered to explain how intermittent mTORC1 signaling expands Zone 1 in the Li-TKO;RGTP liver. Furthermore, most results were obtained in adult mice after the liver pathology has ensued and this precludes distinguishing the primary and secondary effects of intermittent mTORC1 signaling. Therefore, although the study is novel and interesting it requires substantial modifications to substantiate the most important conclusions.

MAJOR ISSUES:

1. Some results in the paper indicate changes in metabolic zonation without properly showing the expression of specific zonation markers (e.g., Figs. 3D, 6A and 7C). Double immunostaining using antibodies for GS (Zone3) and Ecadherin (Zone 1), and Cyp2e1 (Zone 2/3) and Ecadherin (Zone 1) should be used to conclusively establish how metabolic zonation is affected. Also, expression of the Zone 1 marker APC should be evaluated since this is a known inhibitor of Wnt/beta-catenin signaling and its expansion could explain why Zone 3 is lost in the postnatal Li-TKO;RGTP liver. The investigators could use antibodies reported in other studies of liver zonation for the suggested immunostaining analysis (Benhamouche S. et al., Dev Cell 2006; Burke Z. et al., Sci. Reports, 2018; Ma R. et al., Elife, 2020; Planas-Paz et al., Nat. Cell Biol. 2015).
2. The results in Figure 3D compare the intensity of phospho-S6 immunostaining amongst different liver specimens. However, these results should be normalized since the intensity of immunofluorescence signals varies in individual experiments. Also, the image of the wildtype liver shows higher phospho-S6 expression in Zone 1 compared to Zone 3 under fasting conditions, and comparable phospho-S6 levels in Zone 1 and Zone 3 under refed conditions. This experiment should be repeated using anti P-S6 antibodies in combination with antibodies for GS (Zone 3), Cyp2e1 (Zone 2/3) and Ecadherin (Zone 1) to convincingly demonstrate that mTORC1 activity changes in the metabolic zones under fasting and refed conditions. The intensity of P-S6 and the size of Zone 1 and

Zone 3 should be quantified in both conditions and in each mutant strain. Investigating the expression of P-S6 after birth in the wildtype liver is also necessary since the study anticipates that P-S6 levels decrease in zone 3 as the zonation patterns are established.

3. The results in Figure 6 are the most relevant findings in the study but the analysis is only partial, and the quality of the images is suboptimal. The immunostaining experiments need to be repeated using a combination of antibodies as previously indicated (P-S6, GS, Cyp2e1, Ecadherin and APC). Also, the expression of WNT2 should be evaluated using in situ hybridization. These analyses should be performed in control and Li-TKO;RGTP livers isolated at E19.5 – P28, not in adult stages. Markers of apoptosis and proliferation should be included to rule out that the loss of pericentral GS+ hepatocytes is the result of increased cell death and/or reduced proliferation. Quantifying proliferation in Zone 1 is also necessary as this area is expanded in postnatal Li-TKO;RGTP livers.

4. Performing a complementary experiment of mTORC1 inactivation with Alb-cre or rapamycin administration to postnatal mice should strengthen the analysis as one would predict that blocking mTORC1 signaling reduces Zone 1 and expands Zone 3.

5. The Cho et al. Cell Discovery paper (2019) describes a mouse model with constitutive mTORC1 signaling in hepatocytes showing a phenotype very similar to that of Li-TKO;RGTP livers. In that paper, the authors conclude that excessive oxidative stress is a major driver of the liver pathology. One of the most significantly upregulated pathways in the adult Li-TKO;RGTP liver is ROS signaling (Fig. 4A) but the Plata Gomez study does not pursue or discuss this finding. The possibility that oxidative stress is increased in the postnatal Li-TKO;RGTP liver (P5 – P28) needs to be investigated.

6. The finding that pericentral endothelial cells (ECs) express very low or no WNT2 in Li-TKO;RGTP livers study led to the conclusion that constitutive hyperactivation of mTORC1 in hepatocytes affects the adjacent endothelium in a non-cell-autonomous manner. This intriguing alteration is analyzed only superficially, and the results do not offer any mechanistic explanation. Comparing gene expression profiles between ECs isolated from control and mutant livers could identify potentially affected signaling pathways. This experiment should be performed at a stage when WNT2 expression starts to decay in Li-TKO;RGTP livers (P10 – P14, Fig. 6D) and not in adult stages when a complex pathology has ensued. Immunostaining/in situ hybridization and qRT-PCR analyses could be used to characterize the central vein endothelium and the liver sinusoidal endothelium in pericentral areas of Li-TKO;RGTP livers as described in other studies (e.g., Halpern et al., Nat. Biotech. 2018).

7. The “Methods” section indicates that Li-Ind-TKO;RGTP mice were produced but the manuscript does not present any results using those mice. If that strain is still available, the investigators could use it to investigate if intermittent mTORC1 signaling impairs established metabolic zonation (i.e., after weaning and in the adult liver).

8. The parenteral experiments in Fig. 7 are interesting but comparing the findings in piglets to those in Li-Ind-TKO;RGTP mice is difficult, since there are no results (or references) showing how liver zonation is established in pigs after birth. Moreover, the study does not indicate if the experiments were performed prior or after the establishment of metabolic hepatic zonation.

OTHER ISSUES:

- The quality of the immunostaining images is poor, and the immunofluorescence signals look weak. Higher magnification pictures of P-S6 should be more convincing.
- SOX9 is highly expressed in cholangiocytes in the bile ducts, and at low levels in some periportal hepatocytes. Immunostaining experiments with SOX9 antibodies in combination with hepatocyte or cholangiocyte markers should clarify which cells are responsible for the elevated expression of Sox9 in Li-Ind-TKO;RGTP livers (Fig. 5).

Reviewer #2 (Remarks to the Author):

In their manuscript entitled "Regulation of nutrient and hormone signaling to mTORC1 in hepatocytes instructs the postnatal metabolic zonation of the liver" Plata-Gómez et al. characterize the impact of simultaneous activation of nutrient and hormone signaling on liver function via mTORC1. Therefore, the authors generated a mouse line harboring hepatocytes that express the active form of RagA and lack Tsc1. Interestingly, this did not only lead to an inflammatory phenotype and accelerated HCC development, but also to the loss of metabolic portal and central zonal hepatocyte identities, as indicated by their proteomic and transcriptomic analyses. Pharmacological inhibition of mTORC1 with rapamycin was sufficient to revert the phenotype. Further, the authors suggest that morphogenic Wnt ligands secreted by liver endothelial cells are involved in modulating mTORC1 activity. In a model of total parenteral nutrition (TPN) in neonatal pigs, Plata-Gómez and colleagues recapitulated the inability to establish metabolic zonation after birth dependent on postnatal constitutively high mTORC1 activity.

Overall, the authors highlight an interesting concept of how postnatal fluctuations in mTORC1 signaling in hepatocytes may lead to the establishment of spatially segregated liver functions. Convincing data are provided showing that hepatocytes sense nutrients and hormones (independently) which impacts metabolic gene expression. The paper is very well written and interesting to read. However, the proposed EC-hepatocyte crosstalk linking mTORC1 and Wnt signaling to regulate metabolic zonation required additional experimental work. Thus, the following concerns should be addressed:

Major comments

1. The authors claim they found different levels of mTORC1 activation by phospho-S6 staining in Z3 in the WT after refeeding, and in the double mutants. However, it is unclear to the reviewer how the three zones were defined: did the authors perform co-stainings? Or was this selection done "manually"? If done manually, can the authors validate this with a co-staining with a zoned marker? How did the authors ensure exclusive quantification of hepatocyte mTORC1 activity? Please also add magnified insets to the images in Figure 3D. Moreover, can the authors elaborate more on the finding that Z3 hepatocytes seem to be most sensitive to manipulation of mTORC1 regulators? For example, are there any hints when looking at zonal expression profiles (from public available data)?
2. The authors have performed RNA-seq on Li-TSC1KORagAGTP hepatocytes and compared it to the Li-TSC1KO and the WT. Why did they not include the RagGTP mice? It would be interesting to see whether they show comparable signatures to the TSC1KO and elaborate further, if the effects on mTORC1 seen in the double mutant are rather additive or synergistic on a gene level. Additionally, the WT+rapamycin control condition is missing. The authors see around 3000 DEGs in the double mutant compared to WT, however, only 621 genes are "rescued" by rapamycin, meaning inhibition of mTORC1. What are these non-rescued genes, is this solely an effect of long-term activation of mTORC1 or are other pathways affected as well in the double mutants? What happens on the gene level if the authors perform long-term treatment with rapamycin in the double mutants?
3. When the authors perform GSEA for Z1 and Z3 on the DEGs in the double mutants, they find a partial loss of Z3 and an enrichment of Z1 signatures. On the other hand, many metabolic functions including Z1 metabolism are decreased, rather indicating a general decline in hepatocyte function than a strictly zoned effect. In line with that, some of their histological stainings indicate vast morphological changes implying rather general disruptions of the liver than specific zonal effects. How do the authors explain this discrepancy?
4. Although the authors have provided some evidence on the EC:Wnt2-Hep:mTORC1 signaling axis, the data remains superficial and direct link between Wnt and mTORC1 signaling is missing. How does

the hypothesis of EC-derived Wnt2 modulating metabolic zonation fit with previous published data including EC-specific knockout of Wnt2 (PMID: 36220068) showing only a mild effect on Wnt-regulated metabolic enzymes in zone 3. Did the authors also see gender-specific effects as the Monga lab did? The authors should also check RSPO ligands and components of the RSPO module that play an important role in metabolic zonation (PMID: 35006616, PMID: 27088858). E.g. in their in vitro experiment they conclude that "a reduction in Wnt pathway targets with similar levels of Wnt ligands in hepatocytes was consistent with the scRNAseq data supporting reduced production and secretion of Wnt2 ligand from EC." But what if it is RSPO that is missing? They authors hypothesize that a hepatocyte-EC crosstalk is responsible for the impaired metabolic zonation rather than a hepatocyte-specific mechanisms but what about a direct connection of the Wnt pathway with mTORC1 as shown in different settings (PMID: 30297426, PMID: 3071311).

5. The authors characterize zonation in embryos and adults of the WT and double mutants. However, it remains still unclear if the inflammatory phenotype can be observed at the embryonic stage already. Can the authors elaborate on the impact of inflammation on zonation?

6. In Figure 7d the authors perform qPCR on bulk liver and check for zonation markers. Overall, only few genes seem to be affected. In general, it would be more meaningful to provide this data rather on Z1 and Z3 hepatocytes than on bulk (e.g. spatial sorting).

Minor comments

7. The authors should be consistent with their nomenclature and their visualization. E.g. using abbreviations vs full word ("ctrl" vs "control" or "rapa" vs "rapamycin") in the same Figure (here Fig2), and differences in line thicknesses even within the same plot (eg Fig2E). Moreover, some of the figures are blurry (eg. Fig3D).

8. Figure 1C would benefit from magnified insets.

9. Figure 2A: The authors indicate mTORC1 activity in this WB. How do they explain that RagA is expressed at "normal" levels in the double mutant comparable to the WT, but not as in the single RagA mutant?

10. Figure 4F: The authors claim an expansion of Z1 by staining for E-Cadherin. However, the representative picture shown doesn't display this phenotype. A co-staining showing the same area would help the reader to better estimate locations of the periportal versus pericentral zones in each genotype. Can this be confirmed with a second marker gene? The double mutant seems to exhibit huge morphological changes which imply rather general disruptions of the liver than specific zonal effects. In general, E-Cadherin is not metabolic zonal marker but rather a member of the Wnt signaling pathway and thus an indirect marker. Maybe staining for zone1 genes like Pck1, Hal etc gives more clear results.

11. The authors performed SiC RNAseq analysis of parenchymal and non-parenchymal cells. However, not all clusters were annotated. Can the authors annotate them?

12. In Figure 6d and 6e the authors only look at one Wnt ligand and target gene (Wnt2, Axin2). The authors should expand this panel for other Wnt signaling genes.

13. Typo in line 363: "we" instead of "me".

14. Figure 7H: Instead of showing signatures including only few genes for UP/DOWN zonation genes in double mutants the authors should apply the general GSEA for Z1 and Z3.

15. Figure 7L: The authors performed P-S6 staining on TPN and milk-fed piglets. When is the staining

done for the milk-fed condition (fed vs starved)?

16. Instead of, or addition to, a nearly 30-year-old review on metabolic zonation in the introduction section I would suggest citing a more recent one (e.g. PMID: 36693201)

REVIEWERS' COMMENTS:

(The Reviewers' comments are in black font and Authors' responses are in blue)

Reviewer #1 (Remarks to the Author):

This submitted study uses mouse models to understand how nutrient and growth factor signaling are integrated to control mTORC1 activity and liver metabolism. Three mutant mouse strains were generated: Li-RagAGTP mice (with constitutive Rag GTPase signaling in hepatocytes), Li-Tsc1KO mice (with constitutive insulin signaling in hepatocytes), and compound Li-RagAGTP; Li-Tsc1KO mice (Li-TKO;RGTP). The major focus of the study relates to the phenotype of Li-TKO;RGTP mice which is more severe than that in the single mutant mice. Li-RagAGTP; Li-Tsc1KO mice show reduced lifespan and present with a severe ensuing liver pathology involving impaired glucose homeostasis, liver dysfunction, inflammation, fibrosis, and hepatocellular carcinoma (HCC). An almost identical liver pathology was recently reported by another group (Cho CS et al., Cell Discov. 2019) in a mouse model in which nutrient and growth factor signaling to mTORC1 were simultaneously activated in hepatocytes. However, one major difference between the 2 studies is the discovery by Plata-Gomez et al. that metabolic zonation is disrupted in the Li-TKO;RGTP liver. The authors show gradual loss of the pericentral (Zone 3) marker GS, and expanded expression of the periportal (Zone 1) marker E-cadherin in Li-TKO;RGTP livers after birth, and this alteration is accompanied by declining expression of WNT2 in hepatic endothelial cells. Published studies have demonstrated that WNT secretion from central vein endothelial cells and hepatic sinusoidal endothelial cells is necessary to establish and maintaining Zone 3 hepatocyte gene expression. Whether loss of WNT2 expression precedes the loss of pericentral hepatocyte gene expression in Li-TKO;RGTP livers is not convincingly addressed in the manuscript.

The proposal that postnatal intermittent activity of mTORC1 signaling is necessary to establish correct metabolic zonation in the mammalian liver is provocative and very novel. However, as presented the data in support looks insufficient and the liver zonation phenotype requires a more thorough analysis using appropriate markers, better imaging and focusing on the most relevant stages. Another major deficiency is that no mechanism is offered to explain how intermittent mTORC1 signaling expands Zone 1 in the Li-TKO;RGTP liver. Furthermore, most results were obtained in adult mice after the liver pathology has ensued and this precludes distinguishing the primary and secondary effects of intermittent mTORC1 signaling. Therefore, although the study is novel and interesting it requires substantial modifications to substantiate the most important conclusions.

We thank reviewer for valuing our MS as “provocative and very novel”, and “novel and interesting”. We believe we now provide a more thorough support for the conclusions with deeper and complementary analyses, and with important clarifications on some of the data already presented.

MAJOR ISSUES:

1. Some results in the paper indicate changes in metabolic zonation without properly showing the expression of specific zonation markers (e.g., Figs. 3D, 6A and 7C). Double immunostaining using antibodies for GS (Zone3) and Ecadherin (Zone 1), and Cyp2e1 (Zone 2/3) and Ecadherin (Zone 1) should be used to conclusively establish how metabolic zonation is affected. Also, expression of the Zone 1 marker APC should be evaluated since this is a known inhibitor of Wnt/beta-catenin signaling and its expansion could explain why Zone 3 is lost in the postnatal Li-TKO;RGTP liver. The investigators could use antibodies reported in other studies of liver zonation for the suggested immunostaining analysis (Benhamouche S. et al., Dev Cell 2006; Burke Z. et al., Sci. Reports, 2018; Ma R. et al., Elife, 2020; Planas-Paz et al., Nat. Cell Biol. 2015).

We have now extended the battery of zonation markers as indicated by the reviewer. In our hands the antibody against Cyp2e1 (OTI5F11 Thermofisher #MA5-25001) did not yield trustworthy staining, and we used OAT as an alternative for Z3 (GeneTex #GTX50004). As a Z1 marker, we used PCK1 (Ptglab #16754-1-AP), as indicated by Reviewer 2. We have performed double immunostaining against GS (Z3) and E-cad (Z1) as well as OAT (Z2/3) and Pck1 (Z1) to show zonation in adult mice of the different genetic strains used (Figure 4f and Supp Fig 4h, o), and also in the early postnatal stages (Fig 6a and Supp 6c). As expected from our previous histology and immunohistochemistry/IF, proteomics and transcriptomics analyses, the conclusions of a diffused zonated pattern (with decreased Z3 and expanded Z1) are reinforced with these new complementary analyses, which are included in the MS (Fig 4,6). We thank the reviewer for this improvement.

Moreover, we tested several antibodies against APC (EP701Y to APC Abcam #ab40778-Anti-APC (Ab-7,) CC-1 Calbiochem Merck #OP80, Novus #NB100-91662, and APC (F-3) Santa Cruz Biotechnology #c-9998), but in our hands, with our IF protocol and not exceeding a 1:100 concentration, neither of them worked robustly. Thus, we cannot fully exclude a role for APC in the defected zonation pattern of Li-TSC1^{KO}RagA^{GTP} mice, but the lack of differential expression (shown below, Table 1 for Reviewer) from our proteomics and transcriptomics analyses do not support such a role. Instead, new compelling data (see comment #4 from Reviewer 2) support a hepatocyte-intrinsic downregulation of Fzd, acting together with decreased EC-derived production of Wnt ligands and the RSPO module.

Li-TSC1KO RagAGTP vs wild-type (24h fasting)				
Transcriptomics (bulk liver)	Gene name	log2FC	pvalue	padj
	Apc	0,191563878	0,314885239	0,549188544
Proteomics (bulk liver)		logFC	pvalue	
	Apc	-0,002	0,983	

Table 1 for Reviewer. Ratio of the expression of APC from the transcriptomics and proteomics data presented in the MS. No significant differences were observed between mutant and wild type samples.

2. The results in Figure 3D compare the intensity of phospho-S6 immunostaining amongst different liver specimens. However, these results should be normalized since the intensity of immunofluorescence signals varies in individual experiments. Also, the image of the wildtype liver shows higher phospho-S6 expression in Zone 1 compared to Zone 3 under fasting conditions, and comparable phospho-S6 levels in Zone 1 and Zone 3 under refed conditions. This experiment should be repeated using anti P-S6 antibodies in combination with antibodies for GS (Zone 3), Cyp2e1 (Zone 2/3) and Ecadherin (Zone 1) to convincingly demonstrate that mTORC1 activity changes in the metabolic zones under fasting and refed conditions. The intensity of P-S6 and the size of Zone 1 and Zone 3 should be quantified in both conditions and in each mutant strain. Investigating the expression of P-S6 after birth in the wildtype liver is also necessary since the study anticipates that P-S6 levels decrease in zone 3 as the zonation patterns are established.

The reviewer raises an important point on normalization. We thought about this aspect during the experimental design, and so we processed all samples simultaneously. Thus, the experiment on Figure 3 is, empirically, a multi-sample, single-batch IF experiment, so inter-experimental normalization does not apply.

In order to accurately define portal and central zones we had originally included a zonation marker (GS) in all the slides where the quantifications were performed (Fig 3d). Our double immunostaining conducted against p-S6 and GS show zonal changes in mTORC1 activity under fasting/refed in the different genetic models. We have replaced the original images to now show both channels instead of only the channel of pS6 (see Figure 3d). Because portal and central zones are very clear with this double IF, and even with a DAPI counter-stain for a trained eye, we have not repeated this series of quantifications with E-Cadherin as it would have required literally hundreds of hours or work that would not change the validity or robustness of the data already presented.

To investigate potential incipient spatial differences in the regulation of mTORC1 during the establishment of metabolic zonation in livers from wild type mice, we fasted p14 mice for 8 hours and compared them with ad libitum fed, to E19.5 mice and to the spatial differences in the adult stage (Figure 3 on first submission). As this Reviewer accurately anticipated, the phosphorylation of rpS6 in p14 mice already decreases in Zone 3 upon fasting, with less sensitivity to fasting in Zone 1 hepatocytes. These new images are now included in Supp Fig 6k.

3. The results in Figure 6 are the most relevant findings in the study but the analysis is only partial, and the quality of the images is suboptimal. The immunostaining experiments need to be repeated using a combination of antibodies as previously indicated (P-S6, GS, Cyp2e1, Ecadherin and APC). Also, the expression of WNT2 should be evaluated using in situ hybridization. These analyses should be performed in control and Li-TKO;RGTP livers isolated at E19.5 – P28, not in adult stages. Markers of apoptosis and proliferation should be included to rule out that the loss of pericentral GS+ hepatocytes is the result of increased cell death and/or reduced proliferation. Quantifying proliferation in Zone 1 is also necessary as this area is expanded in postnatal Li-TKO;RGTP livers.

We apologize for the suboptimal definition and quality of the images, most likely a consequence of the file compression for reviewing processes. We hope now the quality of images has not been compromised. For the early postnatal days/weeks (original Figure 6), we have now extended the analysis to the same combination of antibodies used now for adult animals (GS+Ecad and OAT+Pck1). In new Figure 6a and Supp Fig 6c, it can be appreciated that all mice, regardless of the genotype, are born with identical lack of molecular zonation. However, while GS and OAT start to be confined to a few layers of Z3 already at p5, and Z1 markers PCK1 and E-Cadherin show an enrichment in Z1 that starts at p10-p14, this zonation pattern is only partially defined in Li-TSC1^{KO}RagA^{GTP} pups, with scattered expression of Z3 markers in the central layers of hepatocytes, and extended expression of both Z1 markers even at p28.

Regarding Wnt2, the levels of this ligand are shown with *in situ* hybridization (commercial RNAscope probe) in the different postnatal stages (Supp Fig 6i). The pattern of positivity in wild type mice follows the expected preferential expression in Z3 cells, with a location consistent with endothelial cells, and a marked reduction of expression in livers from Li-TSC1^{KO}RagA^{GTP} mice. This result is in syntony with the proteomics, transcriptomics, qPCR, WB from bulk livers and from isolated hepatocytes and EC cells and their *in vitro* functional analyses.

To evaluate caspase-dependent apoptosis we have performed immunohistochemistry against cleaved caspase-3 in p14 and 20-week-old mice but observed no detectable increase in apoptosis in samples from Li-TSC1^{KO}RagA^{GTP} mice (shown in Figure 1 for Reviewer). To evaluate proliferation, we have performed staining against Ki67 followed by quantification (shown in Figure 1 for Reviewer) in the same early postnatal and adult mice. Remarkably, all p14 livers show proliferation, while samples from adult Li-TSC1^{KO}RagA^{GTP} exhibit an increase in proliferation in the three hepatic zones, and not restricted to Z1. These data do not support a selective, early expansion of Z1 or the diffusion of molecular zonation as a consequence of early programmed cell death in Z3 or the exclusive proliferation of Z1.

Figure 1 for Reviewer. Immunohistochemical staining for cleaved-caspase 3 and Ki67 at p14 (**A**) and 20-week-old (**B**) samples from wild type and Li-TSC1^{KO}RagA^{GTP} livers. Apoptosis is not appreciable, and quantification of Ki67 shows high proliferation in all three zones of Li-TSC1^{KO}RagA^{GTP} and wild type livers at p14. In samples from adult mice, increased proliferation is observed in all zones from Li-TSC1^{KO}RagA^{GTP} as compared to adult wild type mice, thus collectively not supporting an early Z1-specific proliferation.

4. Performing a complementary experiment of mTORC1 inactivation with Alb-cre or rapamycin administration to postnatal mice should strengthen the analysis as one would predict that blocking mTORC1 signaling reduces Zone 1 and expands Zone 3.

We have now performed a complementary experiment to that of the administration of rapamycin at weaning showed in Figure 2 and Supp Fig 2 on the previously submitted version. In this new experiment, we started rapamycin administration at p1 and through a 2-week period, when the mice were euthanized. We have evaluated again the same markers of zonation by RT-qPCR and IHC as well as Wnt target genes. As seen in Supp Fig 6p-s, inhibition of mTORC1 signaling during the early postnatal stage also partially rescues the impaired zonation of Li-TSC1^{KO}RagA^{GTP} livers, leading to an expansion of Z3 (increased Glul, Cyp2e1) and a concomitant reduction in Z1 (decreased Cdh1). Remarkably, this pharmacological intervention also partially restored the levels of Wnt target genes and improved GS and E-Cadherin zoned patterns in livers from Li-TSC1^{KO}RagA^{GTP} mice. An effect of rapamycin in influencing the maturation in postnatal wt mice was not seen. This may be related to an incomplete or intermittent effect of rapamycin dosage on blocking physiological (not overactive) levels of mTORC1. It is also possible that, at the molecular level, blocking mTORC1 activity does not necessarily mirror molecularly the exact opposite of constitutive mTORC1 activity. Nevertheless, this experiment does prove that early postnatal high and constitutive mTORC1 activity mediates impaired zonal maturation of hepatocytes.

5. The Cho et al. Cell Discovery paper (2019) describes a mouse model with constitutive mTORC1 signaling in hepatocytes showing a phenotype very similar to that of Li-TKO;RGTP livers. In that paper, the authors conclude that excessive oxidative stress is a major driver of the liver pathology. One of the most significantly upregulated pathways in the adult Li-TKO;RGTP liver is ROS signaling (Fig. 4A) but the Plata Gomez study does not pursue or discuss this finding. The possibility that oxidative stress is increased in the postnatal Li-TKO;RGTP liver (P5 – P28) needs to be investigated.

In their seminal discovery, Cho et al. reported increased oxidative stress and we have confirmed their observations in Li-TSC1^{KO}RagA^{GTP} livers. To evaluate whether this phenomenon was also taking place in the early postnatal Li-TSC1^{KO}RagA^{GTP} livers, we have quantified the expression of the three oxidative stress genes described by Cho and colleagues. The results show that the levels of Srxn1 do not change, and Gsta1 and Gstm3 are mildly increased in the postnatal day p14 in Li-TSC1^{KO}RagA^{GTP} livers (Supp Fig 6l-o). The magnitude of the change in the mRNAs of these last two genes at p14 is very mild compared to the massive increase observed during the adult stage (almost 1000- and 70- fold, respectively). Thus, we believe that oxidative stress is a very relevant phenomenon during a chronic phase of liver damage, and contributing to liver function decline and death, as shown previously by Cho; but probably less so in the early postnatal stages. That said, it is possible that oxidative signaling, downstream of increased nutrient and growth factor signaling, is also involved in the early phenotypes of Li-TSC1^{KO}RagA^{GTP} mice in addition to the molecular mediators of the zonation phenotype (Wnt2 and the RSPO module, plus the early decrease in Fzd receptor in mutant hepatocytes). We thank the reviewer for bringing this point and have added this data and its interpretation to the Results and Discussion sections.

6. The finding that pericentral endothelial cells (ECs) express very low or no WNT2 in Li-TKO;RGTP livers study led to the conclusion that constitutive hyperactivation of mTORC1 in hepatocytes affects the adjacent endothelium in a non-cell-autonomous manner. This intriguing alteration is analyzed only superficially, and the results do not offer any mechanistic explanation. Comparing gene expression profiles between ECs isolated from control and mutant livers could identify potentially affected signaling pathways. This experiment should be performed at a stage when WNT2 expression starts to decay in Li-TKO;RGTP livers (P10 – P14, Fig. 6D) and not in adult stages when a complex pathology has ensued. Immunostaining/in situ hybridization and qRT-PCR analyses could be used to characterize the central vein endothelium and the liver sinusoidal endothelium in pericentral areas of Li-TKO;RGTP livers as described in other studies (e.g., Halpern et al., Nat. Biotech. 2018).

We agree with this reviewer with the intriguing hepatocyte-EC communication and the value of such phenomenon. To gather mechanistic bases on this cross-talk, we have:

1. Isolated EC from wild type and Li-TSC1^{KO}RagA^{GTP} livers at the suggested p14 stage and conducted bulk RNAseq analysis to identify potentially affected signaling pathways in EC.

2. We have also quantified *Wnt* targets and *Wnt2* expression levels at the p14 postnatal stage.
 3. We have quantitated sinusoidal (*Lyve1*) versus central (*Cd36*) markers by qPCR at p14 and in adult stage, and have complemented this data with IHC against sinusoidal marker *Lyve1* as well as with heatmaps of liver sinusoidal and central vein EC genes from the bulk transcriptomics performed on purified EC.

The bulk RNAseq from EC at p14 is now presented in Fig 6f-h, and samples cluster as a function of the genotype, suggesting differences between the two genotypes already at p14. Interestingly, morphogenic pathways (*Wnt*, *Hedgehog*) appear downregulated, together with some metabolic pathways (fatty acids, peroxisome, glycolysis) and signaling (*mTORC1*, *myc*, *KRas*) enriched in and *Li-TSC1^{KO}RagA^{GTP}* EC. We have validated the decreased expression of *Wnt*-related genes, and more importantly, the decreased expression of *Wnt2* ligand by qPCR (in consistence with the decreased positivity of *Wnt2* in Z3 EC detected with RNAScope, Supp Fig 6i and j). Moreover, expression of both *Lyve1* (liver sinusoidal EC marker) and *Cd36* (central EC marker) was similar in EC from wild type and *Li-TSC1^{KO}RagA^{GTP}* mice (shown below by mRNA expression and validated by IHC, Figure 2 for Reviewer). In contrast, the expression of *Cd36* is decreased in purified EC from adult and *Li-TSC1^{KO}RagA^{GTP}* mice. Collectively, these analyses support an abnormal zonation of EC from and *Li-*

TSC1^{KO}RagA^{GTP} livers at early stages.

Figure 2 for Reviewer. qPCRs (Top) and heatmaps (Middle) of liver sinusoidal EC (LSEC) and central vein EC (CVEC) from bulk transcriptomics from EC isolated from p14 wild type and *Li-TSC1^{KO}RagA^{GTP}* EC from liver. Bottom. Immunohistochemical staining for *Lyve-1* fails to show robust differences in the pattern of *Lyve-1* expression in wild type and mutant livers.

[redacted]

- The quality of the immunostaining images is poor, and the immunofluorescence signals look weak. Higher magnification pictures of P-S6 should be more convincing.

We believe that the compression during the handling of the MS has limited the quality of all figures. Nevertheless, we have now included insets in some panels for enhanced clarity.

- SOX9 is highly expressed in cholangiocytes in the bile ducts, and at low levels in some periportal hepatocytes. Immunostaining experiments with SOX9 antibodies in combination with hepatocyte or cholangiocyte markers should clarify which cells are responsible for the elevated expression of Sox9 in Li-Ind-TKO;RGTP livers (Fig. 5).

We have performed this double IHC for Sox9 and the cholangiocyte marker Keratin-19, and the results are shown below (Fig 4 for Reviewer). The images point to Sox9 expression both in cholangiocytes and hepatocytes from Li-TSC1^{KO}RagA^{GTP} mice.

Figure 4 for Reviewer. Immunohistochemical staining of Sox9 and Keratin-19 shows that Sox9-positive cells include both hepatocytes and cholangiocytes.

Reviewer #2 (Remarks to the Author):

In their manuscript entitled “Regulation of nutrient and hormone signaling to mTORC1 in hepatocytes instructs the postnatal metabolic zonation of the liver” Plata-Gómez et al. characterize the impact of simultaneous activation of nutrient and hormone signaling on liver function via mTORC1. Therefore, the authors generated a mouse line harboring hepatocytes that express the active form of RagA and lack Tsc1. Interestingly, this did not only lead to an inflammatory phenotype and accelerated HCC development, but also to the loss of metabolic portal and central zonal hepatocyte identities, as indicated by their proteomic and transcriptomic analyses. Pharmacological inhibition of mTORC1 with rapamycin was sufficient to revert the phenotype. Further, the authors suggest that morphogenic Wnt ligands secreted by liver endothelial cells are involved in modulating mTORC1 activity. In a model of total parenteral nutrition (TPN) in neonatal pigs, Plata-Gómez and colleagues recapitulated the inability to establish metabolic zonation after birth dependent on postnatal constitutively high mTORC1 activity. Overall, the authors highlight an interesting concept of how postnatal fluctuations in mTORC1 signaling in hepatocytes may lead to the establishment of spatially segregated liver functions. Convincing data are provided showing that hepatocytes sense nutrients and hormones (independently) which impacts metabolic gene expression. The paper is very well written and interesting to read. However, the proposed EC-hepatocyte crosstalk linking mTORC1 and Wnt signaling to regulate metabolic zonation required additional experimental work. Thus, the following concerns should be addressed:

We thank the reviewer for a thorough analysis of our MS and for finding our work interesting in concept and to read, with convincing data.

Major comments

1. The authors claim they found different levels of mTORC1 activation by phospho-S6 staining in Z3 in the WT after refeeding, and in the double mutants. However, it is unclear to the reviewer how the three zones were defined: did the authors perform co-stainings? Or was this selection done “manually”? If

done manually, can the authors validate this with a co-staining with a zoned marker? How did the authors ensure exclusive quantification of hepatocyte mTORC1 activity? Please also add magnified insets to the images in Figure 3D. Moreover, can the authors elaborate more on the finding that Z3 hepatocytes seem to be most sensitive to manipulation of mTORC1 regulators? For example, are there any hints when looking at zonal expression profiles (from public available data)?

This comment is shared with Reviewer 1 (see response to comment #1 from Reviewer 1, above). All IF staining for quantification within a panel were conducted the same day as a single batch of slides, and including a co-stain for GS to determine the central and portal zones with accuracy. We have now added the omitted GS channel to the IF panel. We have manually defined the region of interest (ROI) as each hepatocyte, and cells were easily identified because p-S6 is cytoplasmic, because other cell types are morphologically different and because of the nuclear shape and size of the DAPI counterstain. For each hepatic lobule, 10 ROIs were manually defined in each central Zone 3, mid-lobular Zone 2 and portal Zone 1, plus 5 ROIs were defined in the intermediate regions of each picture. Three pictures were captured per mouse and the liver of at least three mice was stained per genotype and condition. We have edited the methods section of the manuscript to clarify how quantification was performed.

To look into zonal expression profiles, we obtained the primary expression data on the seminal Halpern et al 2017 paper¹ (shown in their Supplementary Table S3) and compiled the expression gradient of components and regulators of the mTORC1 pathway. For these mTORC1-related genes, we have represented the average expression of Layers 1, 2 and 3 and defined it as Zone 3, and the average expression of Layers 7, 8 and 9 and defined it as Zone 1 (see Supp Fig 3d). Among the components involved in mTORC1 signaling pathway, the ones that are significantly spatially segregated, according to Halpern et al. criteria (q value <0.2 Kruskal-Wallis test) in the liver show a central pattern of expression. From 52 genes, 18 show a zoned pattern of expression, with 15 of them being enriched in the central Z3. This central pattern of expression for many genes related to the mTORC1 pathway is a potential explanation for the increased sensitivity of Z3 hepatocytes to changes in mTORC1 activity (Supp Fig 3d).

2. The authors have performed RNA-seq on Li-TSC1KORagAGTP hepatocytes and compared it to the Li-TSC1KO and the WT. Why did they not include the RagGTP mice? It would be interesting to see whether they show comparable signatures to the TSC1KO and elaborate further, if the effects on mTORC1 seen in the double mutant are rather additive or synergistic on a gene level. Additionally, the WT+rapamycin control condition is missing. The authors see around 3000 DEGs in the double mutant compared to WT, however, only 621 genes are “rescued” by rapamycin, meaning inhibition of mTORC1. What are these non-rescued genes, is this solely an effect of long-term activation of mTORC1 or are other pathways affected as well in the double mutants? What happens on the gene level if the authors perform long-term treatment with rapamycin in the double mutants?

We apologize for the lack of clarity on the samples used in this analysis. The comparison of wild type mice, and Li-RagA^{GTP} mice was conducted and reported in our Nature Communications 2021 paper². In brief, Li-RagA^{GTP} mice and Li-TSC1^{KO} mice shared a large fraction of the transcriptional changes, but those of Li-TSC1^{KO} samples were larger in magnitude, and the number was also bigger. In this MS, our proteomics data include comparisons of single- and double-mutant mice, and the support for synergism is large (both in magnitude of change and in the number of significant changes). To support the notion of overlap between single mutants and the synergistic changes occurring in the Li-TSC1^{KO}RagA^{GTP} liver, we have now added a Venn diagram (Figure 5 for Reviewer and Supp Fig 4d in the MS) that visually conveys this message. In brief, most changes in single mutants are contained in Li-TSC1^{KO}RagA^{GTP} samples, and a large cluster of additional changes is present in Li-TSC1^{KO}RagA^{GTP} samples.

Figure 5 for Reviewer. Venn diagram of differentially-expressed genes (DEG) in livers from Li-RagA^{GTP} (L-R), Li-TSC1^{KO} (L-T) and Li-TSC1^{KO}RagA^{GTP} individually compared to wild type samples. Note the overlap of changes and the large increase in changes in DEG in Li-TSC1^{KO}RagA^{GTP} samples.

Indeed, as this Reviewer mentions, the rapamycin treatment of wild type mice was omitted from the experimental conditions in this already large transcriptomic analysis. The reasons were to obtain a cost-efficient set of data, and because the changes rescued by rapamycin were to be obtained from those observed in untreated Li-TSC1^{KO}RagA^{GTP} mice. As pointed out by reviewer, acute (3-day) rapamycin treatment does not correct the entire set of differentially-expressed genes of Li-TSC1^{KO}RagA^{GTP} livers, and presumably only those directly controlled by deregulated mTORC1 signaling (Supp Fig 4c). The genes that are not rescued include (among other processes) signatures related to inflammation, fibrosis and liver damage, (shown below in Figure 6 for Reviewer) processes that are unlikely to be mitigated by acute rapamycin treatment, and additional metabolic signatures. Nevertheless, rapamycin largely corrects the phenotype when administered for extended periods (Figure 2 and Supplementary Figure 2). It is reasonable to conceive that a profound correction of gene expression (both direct and indirect genes) parallels the drastic phenotypic normalization by long-term pharmacological inhibition of abnormally high mTORC1 activity.

Figure 6 for Reviewer. Gene set enrichment analysis for differentially-expressed genes from rapamycin-treated Li-TSC1^{KO}RagA^{GTP} and wild type mice.

3. When the authors perform GSEA for Z1 and Z3 on the DEGs in the double mutants, they find a partial loss of Z3 and an enrichment of Z1 signatures. On the other hand, many metabolic functions including Z1 metabolism are decreased, rather indicating a general decline in hepatocyte function than a strictly zoned effect. In line with that, some of their histological stainings indicate vast morphological changes implying rather general disruptions of the liver than specific zonal effects. How do the authors explain this discrepancy?

The Reviewer brings an important point. We should emphasize (and have done so in the Discussion) that the fact that zonation is impaired does not preclude nor negates the development of additional phenotypic alterations. Indeed, this is very clear in adult mice, where liver damage, HCC development and metabolic alterations (fasting metabolism, glucose homeostasis, lipid metabolism) are affected in Li-TSC1^{KO}RagA^{GTP} mice. The (lack of) triggering of zonation is a very early event, and the

accompanying later phenotypes may be more directly related to this phenomenon or, alternatively, independent changes operating by chronically deregulated mTORC1 activity. In other words, adult dysfunctional hepatocytes and the early control of zonation within the first weeks of life may be independent processes. The new data pinpointing molecular mediators of the phenotype (see the response to the next comment) and the in vitro rescue of Wnt signaling by exogenous supply of ligands support this independence, rather than a ‘discrepancy’. Again, we do not think or claim that zonation is the one and only phenotype these livers have, neither that these changes must be functionally linked.

4. Although the authors have provided some evidence on the EC:Wnt2-Hep:mTORC1 signaling axis, the data remains superficial and direct link between Wnt and mTORC1 signaling is missing. How does the hypothesis of EC-derived Wnt2 modulating metabolic zonation fit with previous published data including EC-specific knockout of Wnt2 (PMID: 36220068) showing only a mild effect on Wnt-regulated metabolic enzymes in zone 3. Did the authors also see gender-specific effects as the Monga lab did? The authors should also check RSPO ligands and components of the RSPO module that play an important role in metabolic zonation (PMID: 35006616, PMID: 27088858). E.g. in their in vitro experiment they conclude that “a reduction in Wnt pathway targets with similar levels of Wnt ligands in hepatocytes was consistent with the scRNAseq data supporting reduced production and secretion of Wnt2 ligand from EC.” But what if it is RSPO that is missing? They authors hypothesize that a hepatocyte-EC crosstalk is responsible for the impaired metabolic zonation rather than a hepatocyte-specific mechanisms but what about a direct connection of the Wnt pathway with mTORC1 as shown in different settings (PMID: 30297426, PMID: 3071311).

Our work states that Wnt2 is an important player in the EC-hepatocyte morphogenic cross-talk that results in the establishment of metabolic zonation. Consistently with this notion, the Monga lab work on Wnt2-KO EC shows an incomplete, but quite prominent, loss of central cell identity (this is very clear in Figure 2a and 2b on that work³). Nevertheless, we do not intend to claim that Wnt2 is the sole mediator of the phenotype of Li-TSC1^{KO}RagA^{GTP} mice. The fact that the phenotype is “genetically initiated” by activation of mTORC1 in hepatocytes through the expression of Albumin-Cre points to additional molecular mediators, at least from the hepatocyte-to-EC direction. Indeed, new data generated in response to this Reviewer’s comment point to hepatocyte-intrinsic defects in processing Wnt signaling (decreased levels of Frizzled receptors) and to changes in the Rspo axis (see below).

Regarding potential gender-specific effects, the data on Figure 5b and 5c were from female mice for practical reasons, but the phenotypes in males and females are similar (this is also shown in the survival curves on Figure 1). Expression data on male mice is now presented in Supp Fig 5c,d, and show similar deregulation of Wnt2 ligand and Wnt targets, with the exception of gender-specific differences in Ccnd1 (down in females, up in males). Data on pups (p0 – p28) has been generated without separating sexes for technical reasons and due to the limited influence of sex hormones before puberty.

Following on the Reviewer’s request on analyzing the potential involvement of the Rspo-Lgr5-Rnf43 module, we have now quantified Rspo3 ligand levels in LECs by qPCR (Figure 5m), in addition to the levels of Lgr5 and Rnf43, which had been already quantified by qPCR and shown in Figure 5b). Moreover, Figure 5l now shows the quantification of Lgr4 and Znf3 obtained from the RNAseq data from bulk liver. Importantly, expression of Rspo3 is significantly decreased in EC, in addition to Wnt2 being the top differentially-expressed gene in the endothelial compartment in scRNAseq. To ascertain a potential involvement of decreased Rspo in the impaired Wnt pathway activation and transcriptional output in hepatocytes, we tested the effect of the supplementation of recombinant Wnt2, Rspo1 and both in combination in their ability to transactivate Wnt targets in primary wild type and Li-TSC1^{KO}RagA^{GTP} hepatocytes (Supp Fig 5k-q). Strikingly, supplementation of Rspo1 induces an approximate 5-fold transactivation of the Wnt/β-catenin pathway in mutant primary hepatocytes, with a minimal effect on wild type counterparts. The effect of Wnt2 supplementation is around 200x, comparable to the effect of concomitant addition of Wnt2 and Rspo1. Thus, we conclude that both Wnt2 and Rspo are decreased in EC from Li-TSC1^{KO}RagA^{GTP} livers and, if exogenously supplemented to isolated hepatocytes, can boost the decreased Wnt pathway activation of Li-TSC1^{KO}RagA^{GTP} hepatocytes. We thank the Reviewer for providing additional insight, and we have now included this data in Supp Figure 5k-q.

Finally, we sought to address the potential hepatocyte-intrinsic impairment in transducing Wnt signaling in Li-TSC1^{KO}RagA^{GTP} hepatocytes. It has been described that mTORC1 suppresses Wnt/β-catenin

signaling through the regulation of Fzd protein levels⁴. Fzd1, Fzd7 and Fzd8 are high in mouse hepatocytes (according to the single-cell sequencing repository Tabula muris). We quantified the levels of Fzd1, 7 and 8 by qPCR during the different postnatal maturation, as well as in isolated hepatocytes (Figure 6i-l). All livers, regardless of the genotype, express equal levels of all three receptors at p0. Strikingly, and coincidentally in time with the maturation of zonation in mice, the levels of Fzd1 and Fzd8 peak in wild type mice around p10, but this increase fails to occur in Li-TSC1^{KO}RagA^{GTP} samples. Expression in isolated Li-TSC1^{KO}RagA^{GTP} hepatocytes is also decreased, further supporting a hepatocyte-intrinsic inability to transduce the signals from Wnt ligands. Collectively, these new results extend the alterations in Wnt signaling in livers from Li-TSC1^{KO}RagA^{GTP} mice to intrinsic signal-transduction based on postnatal decrease in Fzd receptors and decreased synthesis and secretion of EC-derived Wnt2 ligand and Rspo. This multi-layered postnatal deregulation is coherent with the positive reinforcement of morphogenic cues and provides one hepatocyte-intrinsic factor. We thank the reviewer for improving the mechanistic insight on our MS.

5. The authors characterize zonation in embryos and adults of the WT and double mutants. However, it remains still unclear if the inflammatory phenotype can be observed at the embryonic stage already. Can the authors elaborate on the impact of inflammation on zonation?

We attempted to ascertain inflammatory cells in the early postnatal livers from Li-TSC1^{KO}RagA^{GTP} mice. CD45 staining of early postnatal liver sections, which is high in adult Li-TSC1^{KO}RagA^{GTP} mice, is confounded by the profuse extramedullary hematopoiesis that takes place in the embryonic and early neonatal liver, thus compromising our ability to conclusively address this point. While assessing inflammation was not straightforward, we observed a transient increase in oxidative damage in postnatal Li-TSC1^{KO}RagA^{GTP} livers, albeit minimal compared to that of adult Li-TSC1^{KO}RagA^{GTP} mice (see also response to comment #5 from Reviewer 1).

Figure 7 for Reviewer. Representative pictures of immunohistochemistry staining against CD45 in the liver of wild type and Li-TSC1^{KO}RagA^{GTP} mice in embryonic E19.5 and in the different postnatal stages.

6. In Figure 7d the authors perform qPCR on bulk liver and check for zonation markers. Overall, only few genes seem to be affected. In general, it would be more meaningful to provide this data rather on Z1 and Z3 hepatocytes than on bulk (e.g. spatial sorting).

We agree with the Reviewer on the partial effects on piglet zonation, but unfortunately, performing spatial sorting is not a feasible experimental setup with piglets. In addition, the number of experimentally available piglets is smaller compared to that typically used in mouse experimentation, and the data dispersion in individuals that are not genetically identical is typically larger. We have attempted to conduct complementary IHC or IF for alternative means to obtain spatial resolution, but this approach depends on antibodies with specificity for pigs. We have tested different antibodies for immunofluorescence, but these have not worked in this species:

- Purified mouse anti E-Cadherin, Beckton Bickinson # 610181.
- Mouse monoclonal E-cadherin, BD Biosciences # 610182.
- Rabbit monoclonal EP701Y to APC, Abcam #ab40778.
- Anti-APC (Ab-7) mouse mAb (CC-1), Calbiochem. Merck #OP80.
- APC antibody, BSA-free. Novus #NB100-91662.

-APC (F-3). Santa Cruz Biotechnology #sc-9998.
 -CYP2E1 monoclonal antibody OTI5F11, Thermofisher #MA5-25001.
 -PCK1 Polyclonal antibody-Proteintech, Ptglab #16754-1-AP.
 -OAT antibody [AT23A2] 100, Genetex #GTX50004.

That said, we believe we have provided extensive data about zonation in piglets samples through IF against GS, RT-qPCR in bulk liver, RNAseq on bulk liver including GSEAs analyses, already included in main figures. Furthermore, we have now found in the bulk RNA seq from bulk livers a significant downregulation of Rspo3 in piglets on the TPN regime (Table 2 for Reviewer and Supp Fig 7g). While Wnt2 was not detected in this transcriptomic analysis, the levels of Wnt2 were found reduced, as assessed by qPCR (Figure 7j).

Transcriptomics TPN VS MILK						
Name	baseMean	log2FoldChan	lfcSE	stat	pvalue	padj
RSPO3	52,8255423	-1,4138034	0,22878521	-6,1796099	6,426E-10	5,5792E-08

Table 2 for Reviewer. Differential expression of RSPO3 in liver samples from orally-fed versus TPN-fed piglets.

Minor comments

7. The authors should be consistent with their nomenclature and their visualization. E.g. using abbreviations vs full word (“ctrl” vs “control” or “rapa” vs “rapamycin”) in the same Figure (here Fig2), and differences in line thicknesses even within the same plot (eg Fig2E). Moreover, some of the figures are blurry (eg. Fig3D).

We have amended the inconsistencies. We believe that the blurry images are the consequence of data compression, but have included insets on some of the panels for improved visualization.

8. Figure 1C would benefit from magnified insets.

We have included insets.

9. Figure 2A: The authors indicate mTORC1 activity in this WB. How do they explain that RagA is expressed at “normal” levels in the double mutant comparable to the WT, but not as in the single RagA mutant?

We are as intrigued as the Reviewer and have no molecular explanation for this. We have observed these changes in levels before^{2,5}, and we know is not based on mRNA levels, but our previous in vitro attempts to control degradation of RagA have been unsuccessful. The Reviewer probably appreciated that the levels of TSC1 protein also increase in RagA^{GTP} cells, and again, the molecular explanation is not evident.

10. Figure 4F: The authors claim an expansion of Z1 by staining for E-Cadherin. However, the representative picture shown doesn’t display this phenotype. A co-staining showing the same area would help the reader to better estimate locations of the periportal versus pericentral zones in each genotype. Can this be confirmed with a second marker gene? The double mutant seems to exhibit huge morphological changes which imply rather general disruptions of the liver than specific zonal effects. In general, E-Cadherin is not metabolic zonal marker but rather a member of the Wnt signaling pathway and thus an indirect marker. Maybe staining for zone1 genes like Pck1, Hal etc gives more clear results.

This comment also overlaps with comment #1 of Reviewer 1. We have now added Pck1 staining as an independent marker of Z1 (see also our response above).

11. The authors performed SiC RNAseq analysis of parenchymal and non-parenchymal cells. However, not all clusters were annotated. Can the authors annotate them?

We apologize for this and have now annotated all clusters in the Extended Data file.

12. In Figure 6d and 6e the authors only look at one Wnt ligand and target gene (Wnt2, Axin2). The authors should expand this panel for other Wnt signaling genes.

We have now included Rnf43, Ccnd1 and Lgr5 together with Axin2 as additional Wnt signaling genes in the experimental supplementation of primary hepatocytes with recombinant proteins rWnt2 and rRspo1. This was shown in the initial version of the MS in Figure 6d and 6e, with rWnt3a only, instead of rRspo1 and rWnt2. In this new extended panel, all Wnt target genes analyzed increased their expression after rWnt2 or rRspo1 supplementation, as described in the Comment 4 to Reviewer 2 and shown in Supp Fig 5k-q.

13. Typo in line 363: “we” instead of “me”.

We have amended the mistake.

14. Figure 7H: Instead of showing signatures including only few genes for UP/DOWN zonation genes in double mutants the authors should apply the general GSEA for Z1 and Z3.

We have added this analysis to Supp Fig 7d-f in addition to that present on first submission.

15. Figure 7L: The authors performed P-S6 staining on TPN and milk-fed piglets. When is the staining done for the milk-fed condition (fed vs starved)?

Piglets were fasted 4-6 hours prior to sacrifice. We have now included this info in the Methods Section.

16. Instead of, or addition to, a nearly 30-year-old review on metabolic zonation in the introduction section I would suggest citing a more recent one (e.g. PMID: 36693201).

We have updated the reference and thank the reviewer.

References in this rebuttal:

1. Halpern, K. B. *et al.* Single-cell spatial reconstruction reveals global division of labour in the mammalian liver. *Nature* (2017) doi:10.1038/nature21065.
2. de la Calle Arregui, C. *et al.* Limited survival and impaired hepatic fasting metabolism in mice with constitutive Rag GTPase signaling. *Nat Commun* (2021) doi:10.1038/s41467-021-23857-8.
3. Hu, S. *et al.* Single-cell spatial transcriptomics reveals a dynamic control of metabolic zonation and liver regeneration by endothelial cell Wnt2 and Wnt9b. *Cell Rep Med* **3**, (2022).
4. Zeng, H. *et al.* MTORC1 signaling suppresses Wnt/ β -catenin signaling through DVL-dependent regulation of Wnt receptor FZD level. *Proc Natl Acad Sci U S A* **115**, (2018).
5. Efeyan, A. *et al.* Regulation of mTORC1 by the Rag GTPases is necessary for neonatal autophagy and survival. *Nature* **493**, 679–83 (2013).

REVIEWER COMMENTS

Reviewer #1 (Remarks to the Author):

I appreciate the effort made by the investigators to address my previous concerns and their inclusion of new data in this revised manuscript. The new double-immunofluorescence images facilitate the interpretation of changes in metabolic zonation due to increased mTORC1 activity. Also, the new results from the analysis of postnatal livers are relevant as they show specific changes in Fzd and Wnt2 expression that precede the loss of Zone 3 identity and could be responsible for the Li-TSC1KORagAGTP zonation phenotype. Notwithstanding those improvements, while the new data clearly expands and strengthens some aspects of the analysis, the study continues to lack deep mechanistic insight. The proposal that excessive mTORC1 activity in hepatocytes impairs Wnt production in perivenous ECs is largely based on results of qPCR and in situ hybridization experiments, and the presumptive mechanism behind the hepatocyte/EC crosstalk was not investigated or validated. Those and other major issues remain which are indicated in my comments below.

REMAINING MAJOR ISSUES:

1. The discovery that persistent mTORC1 activity causes downregulation of Wnt target genes in Zone 3 hepatocytes is not novel. A published study by Zeng et al (PNAS 2018) showed that increased mTORC1 activation (through Tsc1 genetic ablation) decreases the expression of Wnt target genes characteristic of Zone 3 hepatocytes in the murine liver (including Axin2, Cyp1a2, Cyp2e1, Glul and Lgr5). While the liver phenotype was not thoroughly analyzed in the Zeng et al study, the authors reported similar alterations in the intestinal crypts of Tsc1-null mice. Upon performing experiments in cell cultures, organoids and mice, the investigators proposed a model in which increased mTORC1 activity reduces Wnt signaling by affecting DVL-dependent FZD protein turnover. This resubmitted study expands on those published results by analyzing the liver phenotype of Li-TSC1KORagAGTP mice (which is more prominent than that in Tsc1-null mice). It is surprising that the Zeng et al findings are only vaguely mentioned in the manuscript and this omission is important. A discussion of how the new data in this study compares to that in the Zeng et al paper is necessary. Also, the authors should explain why they did not investigate (or considered) if altered DVL-dependent FZD protein turnover could be a mechanism behind the loss of Zone 3 hepatocyte identity.

2. Some important results in the revised manuscript appear contradictory and should be discussed. For instance, the results in new Figure 3D show very low GS expression in Zone 3 hepatocytes and almost negligible P-S6 expression in the WT liver under fasting conditions. In contrast, GS expression is high in Zone 3 hepatocytes under refeed conditions in which P-S6 expression is prominently upregulated throughout the WT liver. How is this finding reconciled with the results in Li-TSC1KORagAGTP livers showing that high mTORC1 activity almost entirely abrogates GS pericentral expression? Those apparent discrepancies should be clarified and discussed. Furthermore, whether fasting/refeeding changes Zones 2 and 1 metabolic expression should be investigated in the WT adult liver as well. Other issues found with the results in Figure 3 are: 1) the mice used for these experiments were fasted for 24 hours. This is an extreme condition and published studies have concluded that mice should not be fasted for more than 12 hours (PMID: 32110077). Given this concern, it would be important to determine if similar changes in P-S6 and GS expression occur in mice fasted overnight. 2) New Supplementary Figure 3D shows data extracted from a published study of scRNA sequencing in the adult WT mouse liver. The results show increased expression of various upstream regulators of mTORC1 in Zone 3 hepatocytes compared to Zone 1 hepatocytes, and the authors argue that this could explain why Zone 3 hepatocytes have increased sensitivity to mTORC signaling. However, Rheb expression shows an opposite trend (lower in Z3 than in Z1) and this could also influence the responsiveness of Zone 3 hepatocytes to mTORC1 activation. Those discrepancies are not discussed in this resubmission. 3) The immunofluorescence results in panel D show almost no P-S6 expression in the fasted WT liver, and very abundant expression in the refeed WT liver, whereas the results in panel E show only minor quantitative differences (< 2-fold) in P-S6 intensity between

WT livers under fasting and re-fed conditions. What is the explanation for those inconsistencies since the quantitative results were obtained from the immunofluorescence images? 4) Figure 3 should include high magnification images showing separately the green (GS) and red channels (P-S6). The insets do not clearly show that P-S6 is expressed in Zone 3 hepatocytes (except in the double-mutant liver).

3. The main conclusion in this resubmitted study is that nutrient and hormone signaling via mTORC1 is a major determinant of postnatal metabolic liver zonation. This notion is largely based on compelling results in mice and pigs demonstrating that experimental conditions that cause excessive mTORC1 signaling in hepatocytes impair postnatal metabolic zonation. While this finding is novel and important, the analysis remains largely descriptive, and the underlying mechanism is partially explored and lacks rigorous validation. The hypothesis that excessive mTORC1 activity abrogates a spike in Fzd expression in the P10 murine liver is based on qPCR results and no further experiments were conducted to demonstrate how increased mTORC1 activity affects Fzd transcription in hepatocytes. Also, the location of those hepatocytes in which Fzd1 and Fzd8 are upregulated in P10 livers was not investigated. More importantly, a major question that remains unanswered is whether mTORC1 signaling plays a major role in the establishment of metabolic zonation in the postnatal liver under physiological conditions. Loss-of-function experiments could be used to validate the authors' proposal that mTORC1 signaling contributes to segregate metabolic programs in postnatal hepatocytes, and this analysis could be complemented with experiments of rapamycin administration to newborn mice. While this revised manuscript does not include data addressing whether lack of mTORC1 signaling impairs postnatal metabolic zonation, in their responses to Reviewer 2 the investigators mention the results of experiments showing that "An effect of rapamycin in influencing the maturation in postnatal wt mice was not seen." This result contradicts the hypothesis that mTORC1 is a key regulator of postnatal zonation and is unclear why it has been omitted in the manuscript.

4. The finding that both Wnt2 transcripts and Wnt2 proteins are decreased in the Li-TSC1KORagAGTP adult liver led the authors to propose that excessive mTORC1 signaling in hepatocytes decreases Wnt2 expression in central vein ECs non-cell autonomously. This premise is supported by the new results showing that Wnt2 expression is significantly decreased in the Li-TSC1KORagAGTP liver at P10, a stage when Glul and Axin2 expression is no different between WT and Li-TSC1KORagAGTP livers (new Fig. 6C-E). Also, new data in supplementary Figure 5 show that Li-TSC1KORagAGTP hepatocytes have similar response to Wnt2 and Wnt2+Rspo1 stimulation as WT hepatocytes (these results are important and should be moved to the main figures section). Insufficient Wnt2 production in ECs could thus be the main reason behind the gradual loss of Zone 3 identity in pericentral hepatocytes under excessive mTORC1 activity, as those cells require high levels of WNT stimulation to maintain their phenotype. While this finding is significant, the study falls short of disclosing the potential underlying mechanism and the molecular basis of the presumptive hepatocyte-EC crosstalk. This lack of mechanistic insight continues to be a major limitation in the study.

5. The experiments in primary hepatocytes and ECs (Figure 5) should include rigorous validation of the enriched cell populations (and similar validation should be performed for the ECs isolated from P14 livers in Figure 6F). QRT-PCR experiments should compare the expression of bona-fide endothelial cell genes (e.g., Lyve1, Cdh5, Stab2) and hepatocyte genes (e.g., Alb, Hnf4) in the isolated cell preparations to confirm their purity. This is important because the results in Figure 5I show expression of transcripts characteristic of pericentral hepatocytes (Lgr5, Axin2), cholangiocytes (Sox9) and immune cells (Cd44) in the isolated ECs. Another paradoxical result shown in Figure 5G is the upregulation of Sox9 in Li-TSC1KORagAGTP primary hepatocytes since a figure provided to this reviewer shows that Sox9 expression is comparable in periportal hepatocytes of WT and Li-TSC1KORagAGTP livers. The investigators need to explain why was Sox9 expression analyzed and demonstrate where is Sox9 expression upregulated in the Li-TSC1KORagAGTP liver.

6. The authors indicate that metabolic zonation starts to become evident only days or weeks after birth in mammals (page 8, line 352) and this statement is not entirely correct. Two published studies

(PMIDs: 29426940 and 32154783) clearly showed that GS expression is already restricted to hepatocytes in Zone 3 shortly before birth, and Cyp2e1 expression is largely restricted to Zone 2 hepatocytes at P2 in the murine liver. These studies need to be properly cited and discussed in the context of the results in this manuscript showing almost no GS expression in the E19.5 WT liver (new supplemental Figure 6C).

Reviewer #2 (Remarks to the Author):

I thank the authors for their detailed responses and new data that significantly improved the manuscript. I have no further comments.

REVIEWERS' COMMENTS:

(The Reviewers' comments are in black font and Authors' responses are in blue)

Reviewer #1 (Remarks to the Author):

I appreciate the effort made by the investigators to address my previous concerns and their inclusion of new data in this revised manuscript. The new double-immunofluorescence images facilitate the interpretation of changes in metabolic zonation due to increased mTORC1 activity. Also, the new results from the analysis of postnatal livers are relevant as they show specific changes in Fzd and Wnt2 expression that precede the loss of Zone 3 identity and could be responsible for the Li-TSC1KORagAGTP zonation phenotype. Notwithstanding those improvements, while the new data clearly expands and strengthens some aspects of the analysis, the study continues to lack deep mechanistic insight. The proposal that excessive mTORC1 activity in hepatocytes impairs Wnt production in perivenous ECs is largely based on results of qPCR and in situ hybridization experiments, and the presumptive mechanism behind the hepatocyte/EC crosstalk was not investigated or validated. Those and other major issues remain which are indicated in my comments below.

We thank the Reviewer for valuing the improvement of the revised version of our MS, and the strengthening it provides to the work. We agree with some elusive basis of the molecular hepatocyte-to-EC communication, albeit both hepatocyte-intrinsic (Fzd receptors, see also new data as requested) and paracrine (Rspo and Wnt2) mediators of the EC-to-hepatocyte crosstalk have been dissected in the MS. We believe that the metabolic dissection of the mTORC1 pathway as a determinant of the maturation of liver zonation after birth, with genetic and nutrition-based models in two mammalian species, and multilayered -omics approaches and deep cell biology and signal transduction analyses, constitute a significant step forward in the field, and would be valued by the readership of Nature Communications.

REMAINING MAJOR ISSUES:

1. The discovery that persistent mTORC1 activity causes downregulation of Wnt target genes in Zone 3 hepatocytes is not novel. A published study by Zeng et al (PNAS 2018) showed that increased mTORC1 activation (through Tsc1 genetic ablation) decreases the expression of Wnt target genes characteristic of Zone 3 hepatocytes in the murine liver (including Axin2, Cyp1a2, Cyp2e1, Glul and Lgr5). While the liver phenotype was not thoroughly analyzed in the Zeng et al study, the authors reported similar alterations in the intestinal crypts of Tsc1-null mice. Upon performing experiments in cell cultures, organoids and mice, the investigators proposed a model in which increased mTORC1 activity reduces Wnt signaling by affecting DVL-dependent FZD protein turnover. This resubmitted study expands on those published results by analyzing the liver phenotype of Li-TSC1KORagAGTP mice (which is more prominent than that in Tsc1-null mice). It is surprising that the Zeng et al findings are only vaguely mentioned in the manuscript and this omission is important. A discussion of how the new data in this study compares to that in the Zeng et al paper is necessary. Also, the authors should explain why they did not investigate (or considered) if altered DVL-dependent FZD protein turnover could be a mechanism behind the loss of Zone 3 hepatocyte identity.

We apologize for not having commented and contextualized appropriately the results by Zeng et al. We have now explicitly mentioned the relevance of this work in the Discussion, and extended the mention in the Results section before the experiments on the transient wave of Fzd expression on wt, but not mutant, livers. We believe our work complements the nice mentioned work of Cong and colleagues, but assesses a liver maturation phase not specifically investigated in their work. We have extended the discussion in this direction. Regarding the potential investigation on Dvl as a mediator of liver maturation, we observe minimal changes in the levels of mRNAs and protein for Dvl1, 2, 3 in our transcriptomics and proteomics data, respectively (shown below in Table 1 for Reviewer). But more importantly, we would not have pursued this direction, as the novelty of a Dvl-based molecular connection has been nicely assessed in the 2018 PNAS paper. The Dvl-Fzd turnover axis may certainly be at work in postnatal liver maturation, and surely may act together with the transient raise in expression that we describe (concomitantly with Wnt ligand secretion and the Rspo axis), and we have now explicitly discussed this.

		Li-TSC1KORagAGTP vs wild type (24h fasting)			
		Gene name	log2FC	pvalue	padj
Transcriptomics (bulk liver)	Dvl1	-0,121300057	0,554383803	0,752346278	
	Dvl2	0,148116156	0,686121983	0,839854339	
	Dvl3	-0,084303733	0,785715994	0,895109715	
Proteomics (bulk liver)	Dvl1	-0,043	2,58E-01		
	Dvl2	NaN	NaN		
	Dvl3	0,115	7,95E-04		

Table 1 for Reviewer. Ratio of the expression of the *Dvl* family from the transcriptomics and proteomics data from this MS. No significant differences between mutant and wild type samples.

2. Some important results in the revised manuscript appear contradictory and should be discussed. For instance, the results in new Figure 3D show very low GS expression in Zone 3 hepatocytes and almost negligible P-S6 expression in the WT liver under fasting conditions. In contrast, GS expression is high in Zone 3 hepatocytes under refeed conditions in which P-S6 expression is prominently upregulated throughout the WT liver. How is this finding reconciled with the results in Li-TSC1KORagAGTP livers showing that high mTORC1 activity almost entirely abrogates GS pericentral expression? Those apparent discrepancies should be clarified and discussed.

The Reviewer is correct and this representative image of a wt fasted liver shows rather low levels of GS. We did not perceive this apparent (and misleading) decrease in GS, as GS was merely used to unequivocally identify Z3 hepatocytes, as instructed in the first submission, and not for quantitation purposes. Thus, the image is not as representative for GS as it is for phospho-S6. To conclusively ascertain whether GS expression is affected by 24h fasting in wt mice, we have mined our transcriptomic data, and observed no significant differences at the mRNA level of the Glul gene in wild-type livers under fasting vs. refeeding (Table 2 for reviewer). Thus, we have proceeded to substitute the “representative” picture included the MS to avoid misleading conclusions. Regardless of the changes, a critical point is whether non-Zone 3 cells express Glul and whether Zone 3 cells do not express Glul. It is important to mention that the anomalous expression of GS in the Li-Tsc1^{KO}RagA^{GTP} livers is evident both at the quantification of expression and at the pattern of expression, with even some layer 1 Zone 3 hepatocytes devoid of GS signal. Thus, we have replaced the “representative” image of a wt-fasted livers for a more representative one of both phospho-S6 and GS.

Wild-type 2h refeed vs wild-type 24h fasting			
Name	log2FC	pvalue	padj
Glul	0,280686155	0,027001444	0,145721243

Table 2 for Reviewer. Ratio of the expression of Glul from the transcriptomics data presented in the MS. No significant differences were observed between fasted and refeed wild-type samples.

Furthermore, whether fasting/refeeding changes Zones 2 and 1 metabolic expression should be investigated in the WT adult liver as well. Other issues found with the results in Figure 3 are: 1) the mice used for these experiments were fasted for 24 hours. This is an extreme condition and published studies have concluded that mice should not be fasted for more than 12 hours (PMID: 32110077). Given this concern, it would be important to determine if similar changes in P-S6 and GS expression occur in mice fasted overnight.

We investigated changes in the expression of all metabolic genes and observe changes in wt livers as a function of feeding status (Figure 1 for Reviewer). These changes are minor if compared to Li-Tsc1^{KO}RagA^{GTP} samples. While acute fasting/refeeding is expected to modulate the expression of metabolic genes (including zoned genes), it would be extremely surprising if such modulation results in changes in molecular zonal identity.

Regarding the duration of the fasting period, with all due respect, the work cited by the Reviewer is focused on measurements of metabolic (glucose) tolerance and does not elaborate on other protocols, including the regulation of mTORC1 activity. Indeed, our glucose tolerance tests present in the MS were performed with the standard overnight fasting, and not 24h fasting. Moreover, insulin tolerance tests were performed with a 6-hour (9am-to-3pm) fasting. That said, we agree with this Reviewer on the fact that 24h exceeds a typical fasting period in humans, and also overnight fasting does if comparing basal

Figure 1 for Reviewer. Hierarchical clustering Heatmap diagram representing mRNA levels of zonated genes in wild-type animals in 24h fasting (n=3), wild-type animals after 2h refeeding, Li-TSC1^{KO}RagA^{GTP} in 24h fasting (n=3) and Li-TSC1^{KO}RagA^{GTP} after 2h refeeding (n=3) normalized by Z-score.

metabolic rates of mice and humans. Nevertheless, in both fasting regimes that overall period of suppressed intake is one night, (as this Reviewer knows) the active phase of the mice.

We have previously conducted both overnight fasting and 24h fasting regimes (while euthanasia being done always in the morning to avoid circadian metabolic effects) and observed no substantial differences in terms of the inhibition of mTORC1 between these two periods (de la Calle Arregui et al. 2021). Others, including experts in the mTOR field, using Li-Tsc1^{KO} mice (Cornu et al. 2014; Sengupta et al. 2010), have also used overnight and 24h fasting periods without reporting substantial differences. To illustrate this, we have now included herein side-by-side representative images of immunohistochemical images of phospho-S6 from overnight and 24h fasted wild-type livers (Figure 2 for Reviewer), where inhibition of mTORC1 activity in overnight- and 24-hour-fasted wild-type livers is alike.

Figure 2 for Reviewer. Immunohistochemical stain for phospho-S6 in livers from wild-type mice fasted for 16h (left) and fasted for 24h (middle), and Li-TSC1^{KO}RagA^{GTP} mouse fasted for 24h (right).

2) New Supplementary Figure 3D shows data extracted from a published study of scRNA sequencing in the adult WT mouse liver. The results show increased expression of various upstream regulators of mTORC1 in Zone 3 hepatocytes compared to Zone 1 hepatocytes, and the authors argue that this could explain why Zone 3 hepatocytes have increased sensitivity to mTORC signaling. However, Rheb expression shows an opposite trend (lower in Z3 than in Z1) and this could also influence the

responsiveness of Zone 3 hepatocytes to mTORC1 activation. Those discrepancies are not discussed in this resubmission.

We thank the Reviewer for pointing out this exception. From a total of 52 mTOR-related genes analyzed, 18 exhibited significant zoned differences, with 15 out of 18 being significantly higher in Z3, and only 3 being significantly lower in Z3, including Rheb. This asymmetry satisfied Reviewer 2, who asked on first submission whether we could provide a potential explanation to the increased response of Zone 3 to nutrition and genetic perturbations. On second submission, we mentioned that not all regulators exhibit a Z3-to-Z1 dissipation of expression, and actually other regulators show no change. While the Rheb GTPase is a critical positive regulator of mTORC1, the guanosine nucleotide to which Rheb binds in a TSC-dependent manner may define its activity more than the changes in mRNA levels. All in all, it would be surprising if all regulators (both positive and negative) of mTORC1 would have the same pattern of zoned expression. We have now explicitly mentioned in the legend that Rheb, and two other genes, show the opposite trend to most zoned mTOR-related genes.

3) The immunofluorescence results in panel D show almost no P-S6 expression in the fasted WT liver, and very abundant expression in the refed WT liver, whereas the results in panel E show only minor quantitative differences (< 2-fold) in P-S6 intensity between WT livers under fasting and refed conditions. What is the explanation for those inconsistencies since the quantitative results were obtained from the immunofluorescence images?

We performed the quantifications specifically to have non-subjective impressions of quantity of signal in each condition, genotype and spatial location. The quantification has been also done in the western-blots from Supplementary Figure 2a to avoid the perception of inconsistencies; because of the previous comment on the appreciable changes in GS, we have replaced the original representative image.

4) Figure 3 should include high magnification images showing separately the green (GS) and red channels (P-S6). The insets do not clearly show that P-S6 is expressed in Zone 3 hepatocytes (except in the double-mutant liver).

We have now included separate channels in the inset for clarity. Thanks for the suggestion.

3. The main conclusion in this resubmitted study is that nutrient and hormone signaling via mTORC1 is a major determinant of postnatal metabolic liver zonation. This notion is largely based on compelling results in mice and pigs demonstrating that experimental conditions that cause excessive mTORC1 signaling in hepatocytes impair postnatal metabolic zonation. While this finding is novel and important, the analysis remains largely descriptive, and the underlying mechanism is partially explored and lacks rigorous validation. The hypothesis that excessive mTORC1 activity abrogates a spike in Fzd expression in the P10 murine liver is based on qPCR results and no further experiments were conducted to demonstrate how increased mTORC1 activity affects Fzd transcription in hepatocytes. Also, the location of those hepatocytes in which Fzd1 and Fzd8 are upregulated in P10 livers was not investigated. More importantly, a major question that remains unanswered is whether mTORC1 signaling plays a major role in the establishment of metabolic zonation in the postnatal liver under physiological conditions.

With all due respect, Figure 7, consisting on neonatal pigs subjected to total parenteral nutrition and submitted already on the first version, were obtained specifically to mitigate this potential concern: to what extent does the mouse data obtained with genetic approaches mirrors a physiological phenomenon. And precisely the peak in Fzd1 and Fzd8 expression during the first two weeks after birth in wild-type mice, but not in mice with high mTORC1 activity provides additional new support for the “under physiological conditions” point. Hepatocyte-specific expression of Fzd was already shown in primary hepatocytes (Figure 6l). Following this Reviewer’s comments, we aimed at assessing the dynamic expression of Fzd receptors in postnatal liver slides.

We first undertook an IF-based approach but unfortunately, the antibody from Novus Biologicals NB100-2439 resulted in widespread cytoplasmic signal most likely not reflecting real Fzd8 expression levels (not shown). Thus, in order to address this comment, we switched to a commercial RNAscope probe

for *Fzd8*, and the results are now shown in Supplementary Figure 6x and y. Consistently with the qPCR data now present in Figure 6k, postnatal wild-type livers have increased signal for *Fzd8* in hepatocytes of all regions (quantification in Supplementary Figure 6y includes all zones and Zone 3 only hepatocytes) as compared to *Li-Tsc1^{KO}RagA^{GTP}* samples. And importantly, the signal intensity from the wild-type livers dissipates in adult mice, with minimal differences compared to adult *Li-Tsc1^{KO}RagA^{GTP}* samples. A widespread (non-zonal restriction of the) expression of a receptor for a morphogen is not unprecedented, and in the liver has also been shown for glucagon receptor, for example (Cheng et al. 2018).

Loss-of-function experiments could be used to validate the authors' proposal that mTORC1 signaling contributes to segregate metabolic programs in postnatal hepatocytes, and this analysis could be complemented with experiments of rapamycin administration to newborn mice. While this revised manuscript does not include data addressing whether lack of mTORC1 signaling impairs postnatal metabolic zonation, in their responses to Reviewer 2 the investigators mention the results of experiments showing that "An effect of rapamycin in influencing the maturation in postnatal wt mice was not seen." This result contradicts the hypothesis that mTORC1 is a key regulator of postnatal zonation and is unclear why it has been omitted in the manuscript.

This experiment was performed for the second submission, and included as a Figure for Reviewer. Pharmacological inhibition of mTORC1 early after birth partially and significantly rescues the liver maturation phenotype of Li-Tsc1^{KO}RagA^{GTP} at early postnatal stages, fully supporting deregulated mTORC1 activity as the driver of the deficient zonal maturation of Li-Tsc1^{KO}RagA^{GTP} livers. As the Reviewer comments, rapamycin does not influence significantly the maturation of wild-type livers (see below Figure 3 for Reviewer). From a technical point of view, it is reasonable to consider that rapamycin may correct excessive mTORC1 activity of Li-Tsc1^{KO}RagA^{GTP} mice, but may not abrogate all functions of mTORC1 equally and constantly, neither may it suffice to "improve" a normal maturation process in wild-type livers. More importantly, constitutively increased mTORC1 activity (and the inability of mutant hepatocytes to undergo transient phases of mTORC1 inhibition) may not be the exact opposite mirror to constantly inhibiting the mTORC1 pathway. According to our hypothesis, hepatocytes have high mTORC1 activity until birth thanks to steady transplacental supply that supports embryonic growth; and only when feeding intermittency starts mTORC1 activity undergoes on-off transitions. Constitutively low mTORC1 activity after birth, just like fluctuating on-off cycles, may suffice to allow the execution of a Zone 3 - Wnt program. So, the prediction of this Reviewer is one potential outcome, but neither the only potential outcome nor necessarily a contradiction to the model.

We believe this is an interesting discussion but may exceed the Results section and scope of this MS, but worth being included in the Discussion Section. If this Reviewer considers this result in Figure 3 for Reviewer important for the message of the MS, we would be happy to include in the Supplementary Figures file.

Figure 3 for Reviewer. mRNA levels of zoned genes in bulk liver of wild-type pups treated with rapamycin.

4. The finding that both *Wnt2* transcripts and *Wnt2* proteins are decreased in the *Li-TSC1^{KO}RagA^{GTP}* adult liver led the authors to propose that excessive mTORC1 signaling in hepatocytes decreases *Wnt2*

expression in central vein ECs non-cell autonomously. This premise is supported by the new results showing that Wnt2 expression is significantly decreased in the Li-TSC1KORagAGTP liver at P10, a stage when Glul and Axin2 expression is no different between WT and Li-TSC1KORagAGTP livers (new Fig. 6C-E). Also, new data in supplementary Figure 5 show that Li-TSC1KORagAGTP hepatocytes have similar response to Wnt2 and Wnt2+Rspo1 stimulation as WT hepatocytes (these results are important and should be moved to the main figures section). Insufficient Wnt2 production in ECs could thus be the main reason behind the gradual loss of Zone 3 identity in pericentral hepatocytes under excessive mTORC1 activity, as those cells require high levels of WNT stimulation to maintain their phenotype. While this finding is significant, the study falls short of disclosing the potential underlying mechanism and the molecular basis of the presumptive hepatocyte-EC crosstalk. This lack of mechanistic insight continues to be a major limitation in the study.

The identity of hepatocyte-to-EC molecular message, assuming that is one protein or metabolite, remains unknown. In our opinion, the important point here is whether the MS delivers a relevant message and constitutes an impactful step forward in our understanding of the zonation process and the postnatal liver maturation phase. With the enormous amount of data already present, including bulk and single-cell transcriptomics, proteomics, and experiments in mice and piglets, we believe that the MS does contribute an important lesson to the physiological control of metabolism and postnatal development, more than interesting for the wide readership of Nature Communications. We agree with the importance of the Wnt2 and Rspo1 results and have moved them to main Figure 5.

5. The experiments in primary hepatocytes and ECs (Figure 5) should include rigorous validation of the enriched cell populations (and similar validation should be performed for the ECs isolated from P14 livers in Figure 6F). QRT-PCR experiments should compare the expression of bona-fide endothelial cell genes (e.g., Lyve1, Cdh5, Stab2) and hepatocyte genes (e.g., Alb, Hnf4) in the isolated cell preparations to confirm their purity. This is important because the results in Figure 5I show expression of transcripts characteristic of pericentral hepatocytes (Lgr5, Axin2), cholangiocytes (Sox9) and immune cells (Cd44) in the isolated ECs.

We thank the Reviewer for pointing this important technical omission. We have previously reported a protocol for the isolation of primary hepatocytes from mice (Plata-Gómez et al. 2021), so we are confident that purified cells are hepatocytes. Moreover, scRNA sequencing data, present on first submission, showed the decrease in the expression of the mRNA for the Wnt2 ligand, which we then validated in purified EC. Indeed, the expression of Wnt2 ligand in purified EC is at least 100x higher than its expression in primary hepatocytes, already contributing to validate the data and the purity. We have now quantified the mRNA levels of the endothelial marker Pecam1 and the hepatocyte marker Albumin in purified EC and purified hepatocytes. The remarkable differences in the expression (Pecam1 is almost 50x higher in EC as compared to hepatocytes, and Albumin is 1000x higher in hepatocytes as compared to EC) is now shown in Figure 4 for Reviewer.

Another paradoxical result shown in Figure 5G is the upregulation of Sox9 in Li-TSC1KORagAGTP primary hepatocytes since a figure provided to this reviewer shows that Sox9 expression is comparable in periportal hepatocytes of WT and Li-TSC1KORagAGTP livers. The investigators need to explain why was Sox9 expression analyzed and demonstrate where is Sox9 expression upregulated in the Li-TSC1KORagAGTP liver.

Sox9 was included originally because it is a bona fide Wnt target, at least in small intestine (Blache et al. 2004). However, as this Reviewer pointed to in the first revision of our MS, it also marks additional

populations in the liver, such as cholangiocytes. Our earlier Figure 4 for Reviewer showed that Sox9 is present in cholangiocytes and, as mentioned, in some hepatocytes. Presumably, the increase in Sox9 likely reflects the increase in the number of cholangiocytes that Li-Tsc1^{KO}RagA^{GTP} livers show. Furthermore, the expression of Sox9 is not comparable in periportal hepatocytes of wild-type and Li-Tsc1^{KO}RagA^{GTP} livers. As shown again in Figure 5 for Reviewer, Sox9 expression is increased in periportal hepatocytes from mutant livers in comparison to wild-type. Collectively, our data shows that Sox9 is increased in Li-Tsc1^{KO}RagA^{GTP} livers, but it is still unclear to us why this Reviewer believes on the importance of this demonstration in the context of our MS.

Figure 5 for Reviewer – from the previous resubmission. Immunohistochemical staining of Sox9 and Keratin-19 shows that Sox9-positive cells include both hepatocytes and cholangiocytes.

6. The authors indicate that metabolic zonation starts to become evident only days or weeks after birth in mammals (page 8, line 352) and this statement is not entirely correct. Two published studies (PMIDs: 29426940 and 32154783) clearly showed that GS expression is already restricted to hepatocytes in Zone 3 shortly before birth, and Cyp2e1 expression is largely restricted to Zone 2 hepatocytes at P2 in the murine liver. These studies need to be properly cited and discussed in the context of the results in this manuscript showing almost no GS expression in the E19.5 WT liver (new supplemental Figure 6C).

The Reviewer is correct, and we believe we have now modulated the sentence to provide a more accurate statement. Indeed, these two studies report an incipient, albeit very mild late embryonic definition of some markers of zonation. For example, while GS is detectable earlier (mid-term development), a complete pattern of zonation is only found weeks after birth, as others and us have shown. Because the in-utero morphogenesis of the liver is instructed by a Wnt/βcat gradient, spatial differences in the embryonic livers do occur. It is not the morphogenesis, but the maturation and full definition of the metabolic zonation that is only achieved days/weeks after birth. In (Burke et al. 2018), GS begins to be expressed in few hepatoblasts at mid-term development, and gradually increases in number thereafter. Restriction of GS expression occurs at E18 and becomes increasingly limited to the terminal perivenous hepatocytes postnatally. (Ma et al. 2020) detect GS expression in E18.5. Consistently, we do observe minimal, but detectable levels of GS and OAT at E19.5 (Figure 6a and Supplementary Figure 6c). Only after birth does the expression raise and is progressively confined to Zone 3. We have included a citation to these two studies and modified our statement as mentioned.

Reviewer #2 (Remarks to the Author):

I thank the authors for their detailed responses and new data that significantly improved the manuscript. I have no further comments.

We thank the Reviewer for her/his constructive review of our MS.

References:

- Blache P, Van De Wetering M, Duluc I, Domon C, Berta P, et al. 2004. SOX9 is an intestine crypt transcription factor, is regulated by the Wnt pathway, and represses the CDX2 and MUC2 genes. *Journal of Cell Biology*. 166(1):
- Burke ZD, Reed KR, Yeh SW, Meniel V, Sansom OJ, et al. 2018. Spatiotemporal regulation of liver development by the Wnt/ β -catenin pathway. *Sci Rep*. 8(1):
- Cheng X, Kim SY, Okamoto H, Xin Y, Yancopoulos GD, et al. 2018. Glucagon contributes to liver zonation. *Proc Natl Acad Sci U S A*
- Cornu M, Oppliger W, Albert V, Robitaille AM, Trapani F, et al. 2014. Hepatic mTORC1 controls locomotor activity, body temperature, and lipid metabolism through FGF21. *Proc Natl Acad Sci U S A*
- de la Calle Arregui C, Plata-Gómez AB, Deleyto-Seldas N, García F, Ortega-Molina A, et al. 2021. Limited survival and impaired hepatic fasting metabolism in mice with constitutive Rag GTPase signaling. *Nat Commun*
- Ma R, Martínez-Ramírez AS, Borders TL, Gao F, Sosa-Pineda B. 2020. Metabolic and non-metabolic liver zonation is established non-synchronously and requires sinusoidal Wnts. *Elife*. 9:
- Plata-Gómez AB, Crespo M, de la Calle Arregui C, de Prado-Rivas L, Sabio G, Efeyan A. 2021. Protocol for the assessment of mTOR activity in mouse primary hepatocytes. *STAR Protoc*
- Sengupta S, Peterson TR, Laplante M, Oh S, Sabatini DM. 2010. mTORC1 controls fasting-induced ketogenesis and its modulation by ageing. *Nature*. 468(7327):1100–1104

REVIEWERS' COMMENTS

Reviewer #1 (Remarks to the Author):

I thank the authors for their responses to my points raised in the last revision. All the major issues have been satisfactorily addressed, and I find the new version of the manuscript highly improved and comprehensive.

Only a few minor issues remain related to Figure 5, which in my opinion require some adjustment. First, the magnitude of Sox9 upregulation (>10-fold) in the mutant primary hepatocytes justifies clarifying that Sox9 expression increases in periportal but not pericentral hepatocytes in the d-KO liver. Second, the relevance of CD44 upregulation in the mutant primary hepatocytes is not discussed and it is probably better to eliminate this result. Third, the qPCR results from isolated ECs ("Wnt2" Figure 5J) could include validation of the enrichment of this preparation (Figure 4 for this reviewer). Fourth, the qPCR results in Figure 5I show that the Wnt targets Rnf43, Axin2 and Lgr5 are decreased in primary ECs from the mutant liver. This result is confusing (and not properly discussed) because those Wnt targets are expressed in pericentral hepatocytes and not in liver endothelial cells (perhaps this result reflects minimal contamination with pericentral hepatocytes?). Since the most relevant result here is the reduced expression of Wnt2 and Rspo3 in ECs of the mutant liver, I would suggest eliminating the entire Figure 5I.